# Testing Noise Assumptions of Learning Algorithms

**Surbhi Goel**
surbhig@cis.upenn.edu
University of Pennsylvania

**Adam R. Klivans**
klivans@cs.utexas.edu
UT Austin

**Konstantinos Stavropoulos**
kstavrop@cs.utexas.edu
UT Austin

**Arsen Vasilyan**
arsenvasilyan@gmail.com
UT Austin

## Abstract

We pose a fundamental question in computational learning theory: *can we efficiently test whether a training set satisfies the assumptions of a given noise model?* This question has remained unaddressed despite decades of research on learning in the presence of noise. In this work, we show that this task is tractable and present the first efficient algorithm to test various noise assumptions on the training data. To model this question, we extend the recently proposed testable learning framework of Rubinfeld and Vasilyan [RV23] and require a learner with an associated test to satisfy the following two conditions: (1) whenever the test accepts, the learner outputs a classifier along with a *certificate of optimality*, and (2) the test must pass for any dataset drawn according to a specified modeling assumption on both the marginal distribution and the noise model. We then consider the problem of learning halfspaces over Gaussian marginals with Massart noise (where each label can be flipped with probability less than $1/2$ depending on the input features), and give a fully-polynomial time testable learning algorithm. We also show a separation between the classical setting of learning in the presence of structured noise and testable learning. In fact, for the simple case of random classification noise (where each label is flipped with fixed probability $\eta = 1/2$), we show that testable learning requires super-polynomial time while classical learning is trivial.

## 1 Introduction

Developing efficient algorithms for learning in the presence of noise is one of the most fundamental problems in machine learning with a long line of celebrated research. Assumptions on the noise model itself vary greatly. For example, the well-studied random classification noise model (RCN) assumes that the label corruption process is independent across examples, whereas malicious noise models allow a fraction of the (joint) data-generating distribution to be changed adversarially. Understanding the computational landscape of learning with respect to different noise models remains a challenging open problem, serving as the central focus of numerous works in the theory of supervised learning [BFKV98, ABHU15, ABHZ16, YZ17, ZLC17, MV19, DKTZ20a, DKTZ20a, DKK+22, DKS18, BEK02, DDK+24] and unsupervised learning [DKK+24, CKMY22, CGR18, PMJS14, MPW16, BB20, MS16, DDNS22, BKS23, KLL+23, CG18, BBKS24].

In this paper, we address for the first time whether it is possible to efficiently test if the assumptions of a specific noise model hold for a given training set. There are two key reasons for developing such a test. First, without verifying the assumptions of the noise model, we cannot guarantee that our resulting hypothesis achieves the optimal error rate. Second, it is essential to select the learning algorithm best suited to the noise properties of the training set. Specifically, highly structured noise models often admit faster algorithms, and we should choose these algorithms whenever possible.

39th Conference on Neural Information Processing Systems (NeurIPS 2025) Workshop: Reliable ML.

We use the recently introduced *testable learning* [RV23] framework to model these questions. In this framework, a learner first runs a test on the training set. Whenever the test accepts, the learner outputs a classifier along with a proof that the classifier has near-optimal error. Furthermore, the test must accept with high probability whenever the training set is drawn from a distribution satisfying some specified set of modeling assumptions. If the test rejects, the learner recognizes that one of the modeling assumptions has failed and will therefore refrain from outputting a classifier. Here, our modeling assumptions will include both the structure of the noise model and the structure of the marginal distribution from which the data is generated.

More concretely, we will consider the problem of learning halfspaces under Gaussian marginals with respect to *Massart* noise, an extensively studied problem where an adversary flips binary labels independently with probability at most $1/2$ (the probability of flipping can vary across instances). The goal is to find a halfspace $\mathrm{sign}(\mathbf{v} \cdot \mathbf{x})$ with near-optimal misclassification error rate opt $+ \epsilon$, where opt is the best misclassification error rate achievable by a halfspace. For this problem, a long line of work [ABHU15, ABHZ16, YZ17, ZLC17, MV19, DKTZ20a] resulted in the algorithm of Diakonikolas et al. [DKTZ20a] that runs in time $\mathrm{poly}(d/\epsilon)$ and achieves the optimal error rate. In contrast, the worst-case-noise version of this problem (i.e. agnostic learning or, equivalently, learning with adversarial label noise) is believed to require exponential time in the accuracy parameter, even with respect to Gaussian marginals [KKMS08, DKK$^+$21, GGK20].

In this work, we give a testable learning algorithm for halfspaces that runs in time $\mathrm{poly}(d/\epsilon)$ and *certifies the optimality* of its output hypothesis whenever it accepts. Additionally, the algorithm is guaranteed to accept (with high probability) and output a classifier if the marginal distribution is Gaussian and the noise satisfies the Massart condition.

## 1.1 Our Results

**Noise Model.** We focus on the class of i.i.d. oracles where the marginal distribution on $\mathbb{R}^d$ is the standard Gaussian and the labels are generated by an origin-centered halfspace with Massart noise, as defined below.

**Definition 1.1** (Massart Noise Oracle). Let $f : \mathbb{R}^d \to \{\pm 1\}$ be a concept, let $\eta : \mathbb{R}^d \to [0, 1/2]$ and let $\mathcal{D}$ be a distribution over $\mathbb{R}^d$. The oracle $\mathsf{EX}^{\mathsf{Massart}}_{\mathcal{D}, f, \eta}$ receives $m \in \mathbb{N}$ and returns $m$ i.i.d. examples of the form $(\mathbf{x}, y) \in \mathbb{R}^d \times \{\pm 1\}$, where $\mathbf{x} \sim \mathcal{D}$ and $y = \xi \cdot f(\mathbf{x})$, with $\xi = 1$ w.p. $1 - \eta(\mathbf{x})$ and $\xi = -1$ w.p. $\eta(\mathbf{x})$. The quantity $\sup_{\mathbf{x} \in \mathbb{R}^d} \eta(\mathbf{x}) \in [0, 1/2]$ is called the noise rate.

Formally, we consider the oracle class $\mathsf{EX}^{\mathsf{Massart}}_{\mathcal{N}, \mathcal{H}_{\mathsf{hs}}, \eta_0} = \{\mathsf{EX}^{\mathsf{Massart}}_{\mathcal{N}, f, \eta} : f \in \mathcal{H}_{\mathsf{hs}}, \sup_{\mathbf{x} \in \mathbb{R}^d} \eta(\mathbf{x}) \leq \eta_0\}$, where $\mathcal{N}$ is the standard Gaussian distribution in $d$ dimensions and $\mathcal{H}_{\mathsf{hs}}$ is the class of origin-centered halfspaces over $\mathbb{R}^d$, which is formally defined as follows.

**Definition 1.2** (Origin-Centered Halfspaces). We denote with $\mathcal{H}_{\mathsf{hs}}$ the class of origin-centered halfspaces over $\mathbb{R}^d$, i.e., the class of functions $f : \mathbb{R}^d \to \{\pm 1\}$ of the form $f(\mathbf{x}) = \mathrm{sign}(\mathbf{v} \cdot \mathbf{x})$ for some $\mathbf{v} \in \mathbb{S}^{d-1}$, where $\mathrm{sign}(t) = 1$ if $t \geq 0$ and otherwise $\mathrm{sign}(t) = -1$.

**Learning Setting.** Our results work in the following extension of testable learning [RV23].

**Definition 1.3** (Testable Learning, extension of Definition 4 in [RV23]). Let $\mathcal{H} \subseteq \{\mathbb{R}^d \to \{\pm 1\}\}$ be a concept class, $\mathcal{O}$ a class of (randomized) example oracles and $m : (0, 1) \times (0, 1) \to \mathbb{N}$. The tester-learner receives $\epsilon, \delta \in (0, 1)$ and a dataset $\bar{S}$ consisting of i.i.d. points from some distribution $\mathcal{D}_{\mathbf{x}, y}$ over $\mathbb{R}^d \times \{\pm 1\}$ and then either outputs Reject or $(\mathsf{Accept}, h)$ for some $h : \mathbb{R}^d \to \{\pm 1\}$, satisfying the following.

1. (Soundness). If the algorithm accepts, then $h$ satisfies the following with probability $1 - \delta$.
$$\mathbb{P}_{(\mathbf{x}, y) \sim \mathcal{D}_{\mathbf{x}, y}}[y \neq h(\mathbf{x})] \leq \mathsf{opt} + \epsilon, \text{ where } \mathsf{opt} = \min_{f \in \mathcal{H}} \mathbb{P}_{(\mathbf{x}, y) \sim \mathcal{D}_{\mathbf{x}, y}}[y \neq f(\mathbf{x})]$$

2. (Completeness). If $\bar{S}$ is generated by $\mathsf{EX}(m')$, for some i.i.d. oracle $\mathsf{EX} \in \mathcal{O}$ and $m' \geq m(\epsilon, \delta)$, then the algorithm accepts with probability at least $1 - \delta$.

The difference between Definition 1.3 and the definition of [RV23] is that the completeness criterion does not only concern the marginal distribution on $\mathbb{R}^d$, but the joint distribution over $\mathbb{R}^d \times \{\pm 1\}$. The choice of the oracle class $\mathcal{O}$ encapsulates all of the modeling assumptions under which our

algorithm should accept (both on the marginal distribution on $\mathbb{R}^d$, as well as on the labels). Note that the probability of success can be amplified through repetition (see [RV23]), so it suffices to solve the problem for $\delta = 1/3$. Our main results and their relation to prior work are summarized in Table 1.

| Noise Model | | Classical Setting | Testable Setting |
|---|---|---|---|
| Massart $(\eta_0 = 1/2 - c)$ | | $\text{poly}(d, 1/\epsilon)$ [DKTZ20a] | $\text{poly}(d, 1/\epsilon)$ [Thm. B.1] |
| Strong Massart $(\eta_0 = \frac{1}{2})$ | (Upper) | $d^{O(\log \frac{1}{\epsilon})} 2^{\text{poly}(\frac{1}{\epsilon})}$ [DKK$^+$22] | $d^{\tilde{O}(1/\epsilon^2)}$ [RV23, GKK23] |
| | (Lower) | $d^{\Omega(\log(1/\epsilon))}$ [DKK$^+$22] | $d^{\Omega(1/\epsilon^2)}$ [Thm. C.6] |
| Adversarial | (Upper) | $d^{\tilde{O}(1/\epsilon^2)}$ [KKMS08] | $d^{\tilde{O}(1/\epsilon^2)}$ [RV23, GKK23] |
| | (Lower) | $d^{\Omega(1/\epsilon^2)}$ [DKPZ21] | $d^{\Omega(1/\epsilon^2)}$ (implied) |

Table 1: Runtime upper and lower bounds (in the Statistical Query model) for learning the class of origin-centered halfspaces $\mathcal{H}_{\mathsf{hs}}$ over the standard Gaussian distribution with respect to different noise assumptions.

**Upper Bound.** In Theorem B.1, we show that there is a polynomial-time tester-learner for the class $\mathcal{H}_{\mathsf{hs}}$ with respect to $\mathsf{EX}_{\mathcal{N}, \mathcal{H}_{\mathsf{hs}}, \eta_0}^{\mathsf{Massart}}$ for any $\eta_0 \leq 1/2 - c$, where $c$ is any positive constant. Moreover, whenever our algorithm accepts, it is guaranteed to output the optimal halfspace with respect to the input dataset $\bar{S}$, even if $\bar{S}$ is not generated from i.i.d. examples and can, therefore, be completely arbitrary. Given the upper bounds of Table 1 our algorithm can be used as a first step before applying the more powerful (but also more expensive) tester-learner of [RV23, GKK23]. If our algorithm accepts, then we do not need to run the more expensive algorithm. In other words, our results highlight that testable learning can be used for algorithm selection for problems where different assumptions motivate different algorithmic approaches.

**Lower Bounds.** Our upper bound holds when the noise rate is bounded away below $1/2$. We show that this is necessary: in the high-noise regime ($\eta_0 = 1/2$), the best known lower bounds for learning under adversarial label noise also hold in the testable setting, with respect to random classification noise of rate $1/2$ (Definition 2.1), which is a special case of Massart noise. We give both cryptographic lower bounds (Theorem C.4) assuming subexponential hardness on the problem of learning with errors (LWE), as well as statistical query lower bounds (Theorem C.6). Our lower bounds are inherited from lower bounds from the literature of agnostic learning [DKPZ21, Tie23, DKR23] (combined with Observation C.1). Our testable learning model highlights an underappreciated aspect of these agnostic learning lower bounds, namely, that the hard instances are in fact indistinguishable from completely random instances (i.e., random classification noise of rate $1/2$).

Our results imply a separation between the classical and testable settings in the high-noise regime ($\eta_0 = 1/2$, see second row of Table 1), demonstrating that the complexity of testable learning displays a sharper transition with respect to varying noise models compared to classical learning. For the of RCN at noise rate $1/2$ case, the separation is even stronger, since learning is trivial in the classical setting.

## 1.2 Our Techniques

The techniques we employ in this work are significantly more sophisticated than recently developed tools from testable learning. In fact, it is not even clear that techniques from testable learning should apply, as assumptions on the marginal distribution are quite different from assumptions on the noise model. Concretely, we depart from prior work in testable learning [GKSV24, GKSV23] where the testers are designed to certify specific properties of a particular learning algorithm. Instead, here we obtain a "black-box" result that can take *any* learner that is guaranteed to output a near-optimal halfspace in the Massart setting and certify optimality properties of the learner's output hypothesis. To do this, we decompose the error of the candidate output in terms of quantities for which we can provide certifiable bounds by developing appropriate testers (see Section 2 for more details on the decomposition). In particular, we provide a disagreement tester with significantly sharper guarantees compared to the one developed for standard testable learning [GKSV23], and a spectral tester that combines and expands ideas from [GKSV23] as well as recent work on tolerant testable learning

[GSSV24]. The main technical tool we develop to provide these improved guarantees is a notion of families of sandwiching approximators with respect to partitions of $\mathbb{R}^d$.

An outline of the proof of our main result (Theorem B.1) is provided in Appendix B. As a warm-up, in Section 2, we consider the special case of random classification noise, whose analysis is simpler. We complete the proof sketch of our main result in Appendix B. In the following, we give an overview of the disagreement and the spectral testers.

**Disagreement tester and sandwiching polynomials.** Let $\mathbf{v}$ be a unit vector and $S$ be a dataset of size $\text{poly}(d/\epsilon)$. If $S$ consists of Gaussian data-points, then for every unit vector $\mathbf{v}'$ (w.h.p. over $S$)

$$\mathbb{P}_{\mathbf{x} \in S}[\text{sign}(\mathbf{v} \cdot \mathbf{x}) \neq \text{sign}(\mathbf{v}' \cdot \mathbf{x})] = \angle(\mathbf{v}, \mathbf{v}')/\pi \pm \epsilon.$$

Suppose, given $S$ and $\mathbf{v}$, one would like to certify that this property approximately holds for every $\mathbf{v}'$. The method of exhaustive search - i.e. checking this property for different candidate vectors $\mathbf{v}'$ - can be shown to require at least $2^{\Omega(d)}$ time. Using a moment-based approach, we show how to improve this run-time exponentially. In particular, in time $\text{poly}(d, 1/\epsilon)$, we can certify that for all $\mathbf{v}' \in \mathbb{S}^{d-1}$

$$\mathbb{P}_{\mathbf{x} \in S}[\text{sign}(\mathbf{v} \cdot \mathbf{x}) \neq \text{sign}(\mathbf{v}' \cdot \mathbf{x})] = (1 \pm 0.01)\angle(\mathbf{v}, \mathbf{v}')/\pi \pm \epsilon. \tag{1.1}$$

Moreover, whenever $S$ is Gaussian, our tests are guaranteed to pass (Theorem 2.4). Note that [GKSV24, GKSV23, DKK+23] provided disagreement testers that certified one-sided bounds and suffered constant multiplicative error factors[1], while here our disagreement testers certify both upper and lower bounds on the disagreement probability, with a small and controllable multiplicative error factor.

As mentioned earlier, directly checking the disagreement for each candidate vector $\mathbf{v}'$ in a Euclidean cover of the sphere $\mathbb{S}^{d-1}$ does not work, since their number is exponential to the dimension $d$. Instead, our tester discretizes $\mathbb{R}^d$ into buckets corresponding to $\mathbf{v} \cdot \mathbf{x} \in [i\epsilon, (i+1)\epsilon]$ for varying $i$ (Figure 2) and checks for any constant-degree monomial $m$ that

$$\mathbb{E}_{\mathbf{x} \sim S}[m(\mathbf{x}) \cdot \mathbb{1}_{i\epsilon \leq \mathbf{x} \cdot \mathbf{v} \leq (i+1)\epsilon}] \approx \mathbb{E}_{\mathbf{x} \sim \mathcal{N}(0, I_d)}[m(\mathbf{x}) \cdot \mathbb{1}_{i\epsilon \leq \mathbf{x} \cdot \mathbf{v} \leq (i+1)\epsilon}]. \tag{1.2}$$

We show that passing this test for constant-degree $m$ is sufficient for our purposes (Lemma 2.5). The previous work [GKSV24, GKSV23, DKK+23] considered only tests involving monomials $m$ of degree at most 4, and (as explained earlier) achieved bounds far weaker than Equation (1.1). A key ingredient to our improved testers is extending the notion of sandwiching polynomials of [GKK23] to much more general piecewise-polynomial functions (see Appendix D).

**Spectral tester and monotonicity under removal.** The disagreement tester is only guaranteed to accept when the input $S$ is drawn i.i.d. from the standard Gaussian distribution. However, in our analysis it is important to have a tester that will accept even if given a set $S'$ which is a subset of a Gaussian sample. We call this property *monotonicity under removal* and its importance is related to the fact that in the Massart noise model, the labels are not flipped independently of the corresponding features, but the probability of receiving a flipped label can adversarially depend on $\mathbf{x}$. Note that tester in Equation (1.2) is not monotone under removal.

To obtain a tester for the disagreement region that is monotone under removal (Theorem B.3), we augment our Disagreement Tester using ideas from the recent work by [GSSV24] on tolerant testable learning (see Appendix E). In particular, instead of checking Equation (1.2), our Spectral Tester checks that for every constant-degree polynomial $p$ we have

$$\mathbb{E}_{\mathbf{x} \sim S}[p(\mathbf{x})^2 \cdot \mathbb{1}_{i\epsilon \leq \mathbf{x} \cdot \mathbf{v} \leq (i+1)\epsilon}] \lesssim \mathbb{E}_{\mathbf{x} \sim \mathcal{N}(0, I_d)}[p(\mathbf{x})^2 \cdot \mathbb{1}_{i\epsilon \leq \mathbf{x} \cdot \mathbf{v} \leq (i+1)\epsilon}], \tag{1.3}$$

which can be verified efficiently by computing the spectrum of an appropriate matrix. The main difference between our spectral tester and the one in [GSSV24] is that ours partitions $\mathbb{R}^d$ into a number of strips and performs a test for each of them, while the one by [GSSV24] runs the same test on the whole $\mathbb{R}^d$ iteratively, each time removing a number of points from the input set. See Algorithm 2 for the full algorithm description. As in the case of the Disagreement Tester, the analysis again leverages the method of piecewise-polynomial sandwiching functions introduced in this work.

---

[1]i.e. certified that $\mathbb{P}_{\mathbf{x} \in S}[\text{sign}(\mathbf{v} \cdot \mathbf{x}) \neq \text{sign}(\mathbf{v}' \cdot \mathbf{x})] \leq O(\angle(\mathbf{v}, \mathbf{v}')) + \epsilon.$

## 1.3 Related Work

**Learning with label noise.** Learning of halfspaces under label noise has been the topic of a large number of works. Perhaps the most well-studied noise model is the framework of *agnostic learning* which corresponds to adversarial (i.e. worst-case) labels. In case of halfspaces, the literature exhibits a tradeoff between run-time and the classification error achievable:

- In time $d^{\tilde{O}(1/\epsilon^2)}$ one can find a hypothesis with accuracy $\mathsf{opt} + \epsilon$ under Gaussian data distribution [KKMS08, DGJ$^+$10]. See also [DKK$^+$21] for a proper learning algorithm. An algorithm with a run-time $d^{O(1/\epsilon^{2-\Omega(1)})}$ (and let alone a polynomial run-time) is believed to be impossible due to statistical query lower bounds [GGK20, DKZ20, DKZ20], as well as recent cryptographic reductions from lattice problems [DKR23, Tie23]. This works utilize reductions to the continuous LWE problem [BRST21], shown in [GVV22] to be harder than the LWE problem widely used in lattice-based cryptography (a quantum reduction was given in [BRST21]).

- A worse error bound of $O(\mathsf{opt}) + \epsilon$ can be obtained in time $\mathrm{poly}(d/\epsilon)$ [ABL17]. Despite various refinements [Dan15, DKTZ20b, DKK$^+$21], the improvement of the error bound to $\mathsf{opt} + \epsilon$ is precluded by the aforementioned hardness results.

Overall, if one is not allowed to assume anything about data labels, one has to choose between a high run-time of $d^{\tilde{O}(1/\epsilon^2)}$ and or a higher error of $O(\mathsf{opt}) + \epsilon$.

In order to obtain an error bound of $\mathsf{opt} + \epsilon$ in time $\mathrm{poly}(d/\epsilon)$ a large body of works focused on moving beyond worst-case models of label noise. In the Random Classification Noise (RCN) model [AL88] the labels are flipped independently with probability $\eta$. It was shown in [BFKV98] that in the RCN model halfspaces can be learned up to error $\mathsf{opt} + \epsilon$ in time $\mathrm{poly}(d/\epsilon)$. See also [Coh97, DKT21, DTK23, DDK$^+$24].

The Massart noise model, introduced in [MN06], is more general than the RCN model and allows the noise rate $\eta(\mathbf{x})$ to differ across different points $\mathbf{x}$ in space $\mathbb{R}^d$, as long as it is at most some rate $\eta_0$. First studied in [ABHU15], learning halfspaces up to error $\mathsf{opt} + \epsilon$ under Massart noise model has been the focus of a long line of work [ABHZ16, MV19, YZ17, ZLC17, DKTZ20a, ZL21, ZSA20, DKK$^+$22].

We would like to note that intermediate steps in the algorithm [DKK$^+$22] work by finding what is referred in [DKK$^+$22] as sum-of-squares certificates of optimality for certain halfspaces. We would like to emphasize that certificates in the sense of [DKK$^+$22] have to be sound only *assuming* that the labels satisfy the Massart property. In contrast with this, certificates developed in this work satisfy soundness without making any assumptions on the label distribution (which is the central goal of this work).

There has also been work on distribution-free learning under Massart noise [DGT19, CKMY20], which achieves an error bound of $\eta_0 + \epsilon$, but as a result can lead to a much higher error than the information-theoretically optimal bound of $\mathsf{opt} + \epsilon$.

**Testable learning.** The framework of testable learning was introduced in [RV23] with a focus on developing algorithms in the agnostic learning setting that provide certificates of (approximate) optimality of the obtained hypotheses or detect that a distributional assumption does not hold. Many of the existing agnostic learning results have since been shown to have testable learning algorithms with matching run-times. This has been the case for agnostic learning algorithms with $\mathsf{opt} + \epsilon$ error guarantee [RV23, KSV24b, GSSV24, STW24], as well as $O(\mathsf{opt}) + \epsilon$ error bounds [GGKS24, GKSV23, DKK$^+$23, DKLZ24].

We note that [GGKS24, GKSV23] also give testable learning algorithms in the setting where data labels are *assumed* to be Massart (and the algorithm needs to either output a hypothesis with error $\mathsf{opt} + \epsilon$ or detect that data distribution is e.g. not Gaussian). We emphasize that the results of [GGKS24, GKSV23] do not satisfy soundness when the user is not promised that data labels satisfy the Massart noise condition (which is the central goal of this work).

**Testable Learning with Distribution Shift (TDS learning).** The recently introduced TDS framework [KSV24b, KSV24a, CKK$^+$24, GSSV24] considers a setting in which the learning algorithm is given a labeled training dataset and an unlabeled test dataset and aims to either (i) produce accurate labeling for the testing dataset (ii) detect that distribution shift has occurred and the test dataset is not produced

from the same data distribution as the training dataset. Although conceptually similar, the work in TDS learning addresses a different assumption made in learning theory. Nevertheless, the spectral testing technique introduced in [GSSV24] is a crucial technical tool for our results in this work.

## 2 Polynomial-Time Tester-Learners

We first focus on the simpler RCN noise model, and the *Disagreement Tester* we design to test the RCN noise model. We show how to obtain a tester-learner with respect to the challenging Massart noise model, and describe the *Spectral Tester* in Appendix B.

**Notation.** We denote with $\mathbb{R}, \mathbb{N}, \mathbb{Z}$ the sets of real, natural and integer numbers correspondingly. For simplicity, we denote the $d$-dimensional standard Gaussian distribution as $\mathcal{N}_d$ or $\mathcal{N}$ if $d$ is clear by context. For any set $S$, let $\mathrm{Unif}(S)$ denote the uniform distribution over $S$. We may also use the notation $\mathbf{x} \sim S$ in place of $\mathbf{x} \sim \mathrm{Unif}(S)$. For a set of points in $\mathbb{R}^d$, we denote with $\bar{S}$ the corresponding labeled set over $\mathbb{R}^d \times \{\pm 1\}$ where the corresponding labels are those in the input of the algorithm unless otherwise specified. For a vector $\mathbf{x} \in \mathbb{R}^d$, we denote with $x_i$ its $i$-th coordinate.

We formally define random classification noise as follows.

**Definition 2.1** (Random Classification Noise (RCN) Oracle). Let $f : \mathbb{R}^d \to \{\pm 1\}$ be a concept, let $\eta_0 \in [0, 1/2]$ and let $\mathcal{D}$ be a distribution over $\mathbb{R}^d$. The oracle $\mathsf{EX}_{\mathcal{D},f,\eta_0}^{\mathsf{RCN}}$ receives $m \in \mathbb{N}$ and returns $m$ i.i.d. examples of the form $(\mathbf{x}, y) \in \mathbb{R}^d \times \{\pm 1\}$, where $\mathbf{x} \sim \mathcal{D}$ and $y = \xi \cdot f(\mathbf{x})$, where $\xi = 1$ w.p. $1 - \eta_0$ and $\xi = -1$ w.p. $\eta_0$. In other words, $\mathsf{EX}_{\mathcal{D},f,\eta_0}^{\mathsf{RCN}} = \mathsf{EX}_{\mathcal{D},f,\eta}^{\mathsf{Massart}}$ where $\eta(\mathbf{x})$ is the constant function with value $\eta_0$. In the special case $\eta_0 = 1/2$, the function $f$ does not influence the output distribution and we denote the corresponding oracle with $\mathsf{EX}_{\mathcal{D},1/2}^{\mathsf{RCN}}$.

Informally, for some ground-truth halfspace $f$, the RCN oracle $\mathsf{EX}_{\mathcal{D},f,\eta_0}^{\mathsf{RCN}}$ outputs an example $\mathbf{x} \sim \mathcal{D}$ whose label is $f(\mathbf{x})$ with probability $1 - \eta_0$ and is flipped with probability $\eta_0$. We consider the case that $\mathcal{D} = \mathcal{N}_d$, and $\eta_0 \leq 1/2 - c$ for some constant $c > 0$ and $f \in \mathcal{H}_{\mathsf{hs}}$.

**Theorem 2.2** (Warm-up: RCN). *Let $c \in (0, 1/2)$ be any constant and $\eta_0 = 1/2 - c$. Then, there is an algorithm that testably learns the class $\mathcal{H}_{\mathsf{hs}}$ with respect to $\mathsf{EX}_{\mathcal{N},\mathcal{H}_{\mathsf{hs}},\eta_0}^{\mathsf{RCN}} = \{\mathsf{EX}_{\mathcal{N},f,\eta_0}^{\mathsf{RCN}} : f \in \mathcal{H}_{\mathsf{hs}}\}$ with time and sample complexity $\mathrm{poly}(d, 1/\epsilon) \log(1/\delta)$.*

Testing whether the noise is indeed RCN directly is impossible, since it requires estimating $\mathbb{E}[y|\mathbf{x}]$ for all $\mathbf{x} \in \mathbb{R}^d$, but we never see any example twice. Instead, we will need to design more specialized tests that only check the properties of the RCN model that are important for learning halfspaces. Specifically, we show that some key properties of the RCN noise can be certified using what we call the Disagreement Tester (Theorem 2.4). Suppose, first, that when the samples are generated by an oracle $\mathsf{EX}_{\mathcal{N},f^*,\eta_0}^{\mathsf{RCN}}$, for some $f^*(\mathbf{x}) = \mathsf{sign}(\mathbf{v}^* \cdot \mathbf{x})$, then we can *exactly* recover the ground-truth vector $\mathbf{v}^* \in \mathbb{S}^{d-1}$ by running some algorithm $\mathcal{A}$ (in reality, $\mathbf{v}^*$ can be recovered only approximately, and we will address this later).

**Relating the output error to optimum error.** Let $\bar{S}$ be the input set of labeled examples and let $\mathbf{v} \in \mathbb{S}^{d-1}$ be the output of $\mathcal{A}$ on input $\bar{S}$. Note that, since $\bar{S}$ is not necessarily generated by $\mathsf{EX}_{\mathcal{N},f^*,\eta_0}^{\mathsf{RCN}}$, we do not have any a priori guarantees on $\mathbf{v}$. We may relate the output error $\mathbb{P}_{(\mathbf{x},y)\sim\bar{S}}[y \neq \mathsf{sign}(\mathbf{v}\cdot\mathbf{x})]$ to the optimum error $\mathbb{P}_{(\mathbf{x},y)\sim\bar{S}}[y \neq \mathsf{sign}(\mathbf{v}^*\cdot\mathbf{x})]$, by accounting for the set $\bar{S}_g$ of points $(\mathbf{x}, y) \in \bar{S}$ that are labeled correctly by $\mathbf{v}$ but incorrectly by $\mathbf{v}^*$, as well as the set $\bar{S}_b$ of points in $\bar{S}$ that are labeled incorrectly by $\mathbf{v}$ but correctly by $\mathbf{v}^*$. Overall, we have the following

$$\mathbb{P}_{(\mathbf{x},y)\sim\bar{S}}[y \neq \mathsf{sign}(\mathbf{v}\cdot\mathbf{x})] = \mathbb{P}_{(\mathbf{x},y)\sim\bar{S}}[y \neq \mathsf{sign}(\mathbf{v}^*\cdot\mathbf{x})] + \frac{|\bar{S}_b|}{|\bar{S}|} - \frac{|\bar{S}_g|}{|\bar{S}|} \tag{2.1}$$

**Towards a testable bound.** We have assumed that if the noise was indeed RCN, then $\mathbf{v} = \mathbf{v}^*$. Therefore, in this case, $|\bar{S}_b| = |\bar{S}_g| = 0$. However, if the noise assumption is not guaranteed, given $\bar{S}$, we cannot directly compute the quantities $|\bar{S}_b|, |\bar{S}_g|$, as their definition involves the unknown vector

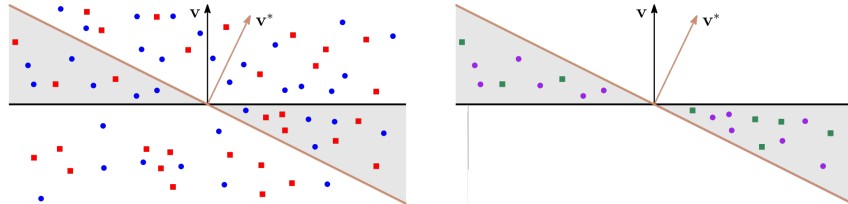

Figure 1: The shaded region is $\{\mathbf{x} \in \mathbb{R}^d : \mathsf{sign}(\mathbf{v} \cdot \mathbf{x}) \neq \mathsf{sign}(\mathbf{v}^* \cdot \mathbf{x})\}$. Left: red square points have label $+1$, blue round points have label $-1$. Right: green square points are in $\bar{S}_g$ and purple round points are in $\bar{S}_b$.

$\mathbf{v}^*$. Nevertheless, we show how to obtain a certificate that $|\bar{S}_b|/|\bar{S}| - |\bar{S}_g|/|\bar{S}|$ is at most $O(\epsilon)$. We first express the ratios above as

$$|\bar{S}_b|/|\bar{S}| = \mathop{\mathbb{P}}_{(\mathbf{x},y)\sim\bar{S}}[y \neq \mathsf{sign}(\mathbf{v} \cdot \mathbf{x}) \text{ and } \mathsf{sign}(\mathbf{v}^* \cdot \mathbf{x}) \neq \mathsf{sign}(\mathbf{v} \cdot \mathbf{x})], \tag{2.2}$$

$$|\bar{S}_g|/|\bar{S}| = \mathop{\mathbb{P}}_{(\mathbf{x},y)\sim\bar{S}}[\mathsf{sign}(\mathbf{v}^* \cdot \mathbf{x}) \neq \mathsf{sign}(\mathbf{v} \cdot \mathbf{x})] - |\bar{S}_b|/|\bar{S}|. \tag{2.3}$$

Combining equations (2.1), (2.2) and (2.3), defining $\bar{S}_{\mathsf{False}} = \{(\mathbf{x},y) \in \bar{S} : y \neq \mathsf{sign}(\mathbf{v} \cdot \mathbf{x})\}$ we obtain the following bound:

$$\frac{|\bar{S}_b|}{|\bar{S}|} - \frac{|\bar{S}_g|}{|\bar{S}|} \leq 2\frac{|\bar{S}_{\mathsf{False}}|}{|\bar{S}|} \mathop{\mathbb{P}}_{(\mathbf{x},y)\sim\bar{S}_{\mathsf{False}}}[\mathsf{sign}(\mathbf{v}^* \cdot \mathbf{x}) \neq \mathsf{sign}(\mathbf{v} \cdot \mathbf{x})] - \mathop{\mathbb{P}}_{(\mathbf{x},y)\sim\bar{S}}\left[\mathsf{sign}(\mathbf{v}^* \cdot \mathbf{x}) \neq \mathsf{sign}(\mathbf{v} \cdot \mathbf{x})\right]$$
$$\tag{2.4}$$

The term $|\bar{S}_{\mathsf{False}}|/|\bar{S}|$ can be explicitly computed, since we have $\mathbf{v}$ and $\bar{S}$ and we can verify whether its value is at most $1/2 - c$, as would be the case if the noise was RCN. Otherwise, we may safely reject. Now, we want to obtain certificates that the first term in Equation 2.4 can't be too large and the second term can't be too small.

**The disagreement tester and how it is applied.** Our goal is to certify that both probabilities in Equation 2.4 are approximately equal to $\angle(\mathbf{v}, \mathbf{v}^*)/\pi$, which is what we would expect if the example oracle were indeed in $\mathsf{EX}^{\mathsf{RCN}}_{\mathcal{N}, \mathcal{H}_{\mathsf{hs}}, \eta_0}$. This is because of the following fact, as well as the fact that even after conditioning on the event $y \neq \mathsf{sign}(\mathbf{v}^* \cdot \mathbf{x})$, $\mathbf{x}$ remains Gaussian.

**Fact 2.3.** *Let $\mathbf{x} \sim \mathcal{N}_d$ and $\mathbf{v}, \mathbf{v}^* \in \mathbb{S}^{d-1}$. Then $\mathbb{P}_{\mathbf{x}\sim\mathcal{N}_d}[\mathsf{sign}(\mathbf{v}^* \cdot \mathbf{x}) \neq \mathsf{sign}(\mathbf{v} \cdot \mathbf{x})] = \angle(\mathbf{v}, \mathbf{v}^*)/\pi$.*

Recall, however, that we do not make any assumptions on the input examples. Therefore, we would like to certify the guarantee of Fact 2.3. We show that this is possible by developing the following tester.

**Theorem 2.4** (Disagreement tester, see Theorem D.1). *Let $\mu \in (0, 1)$ be any constant. Algorithm 1 receives $\epsilon, \delta \in (0, 1)$, $\mathbf{v} \in \mathbb{S}^{d-1}$ and a set $S$ of points in $\mathbb{R}^d$, runs in time $\mathrm{poly}(d, 1/\epsilon, |S|)$ and then either outputs* Reject *or* Accept, *satisfying the following specifications.*

1. *(Soundness) If the algorithm accepts, then the following is true for any $\mathbf{v}' \in \mathbb{S}^{d-1}$*

$$(1 - \mu)\angle(\mathbf{v}, \mathbf{v}')/\pi - \epsilon \leq \mathop{\mathbb{P}}_{\mathbf{x}\sim S}[\mathsf{sign}(\mathbf{v} \cdot \mathbf{x}) \neq \mathsf{sign}(\mathbf{v}' \cdot \mathbf{x})] \leq (1 + \mu)\angle(\mathbf{v}, \mathbf{v}')/\pi + \epsilon$$

2. *(Completeness) If $S$ consists of at least $(\frac{Cd}{\epsilon\delta})^C$ i.i.d. examples from $\mathcal{N}_d$, where $C \geq 1$ is some sufficiently large constant depending on $\mu$, then the algorithm accepts with probability at least $1 - \delta$.*

We choose $\mu = c$ and run the tester above on the datapoints in $\bar{S}$ and $\bar{S}_{\mathsf{False}} = \{(\mathbf{x}, y) \in \bar{S} : y \neq \mathsf{sign}(\mathbf{v} \cdot \mathbf{x})\}$. If the tester accepts, then (using Equation 2.4) the excess error $|\bar{S}_b|/|\bar{S}| - |\bar{S}_g|/|\bar{S}|$ is at most $((1 - 2c)(1 + c) - (1 - c))\angle(\mathbf{v}, \mathbf{v}^*)/\pi + 2\epsilon$ which, in turn, is upper-bounded by $2\epsilon$, since $(1 - 2c)(1 + c) - (1 - c) = -2c^2 < 0$.

**Designing the disagreement tester.** Our disagreement tester builds on ideas from prior work on testable agnostic learning by [GKSV24]. In particular, [GKSV24] show that when the angle between

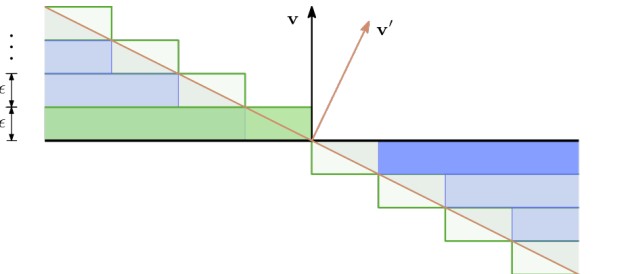

Figure 2: For vectors $\mathbf{v}, \mathbf{v}' \in \mathbb{S}^{d-1}$, the region $\{\mathbf{x} \in \mathbb{R}^d : \text{sign}(\mathbf{v} \cdot \mathbf{x}) \neq \text{sign}(\mathbf{v}' \cdot \mathbf{x})\}$ is contained in the union of green regions and it contains the union of blue regions. In the diagram we highlight one of the green regions (top left) and one of the blue regions (bottom right).

the input vector $\mathbf{v}$ and some unknown vector $\mathbf{v}'$ is $\epsilon$, then one can give a testable bound of $O(\epsilon)$ on the quantity $\mathbb{P}_{\mathbf{x} \sim S}[\text{sign}(\mathbf{v} \cdot \mathbf{x}) \neq \text{sign}(\mathbf{v}' \cdot \mathbf{x})]$ by running some efficient tester (Proposition D.1 in [GKSV24]). To achieve this, the region $\{\mathbf{x} \in \mathbb{R}^d : \text{sign}(\mathbf{v} \cdot \mathbf{x}) \neq \text{sign}(\mathbf{v}' \cdot \mathbf{x})\}$ is covered by a disjoint union of simple regions whose masses can be testably upper bounded. In order to bound the mass of the simple regions, it is crucial to use the fact that $\mathbf{v}$ is known.

Here, our approach needs to be more careful, since we (1) require both upper and lower bounds on the quantity $\mathbb{P}_{\mathbf{x} \sim S}[\text{sign}(\mathbf{v} \cdot \mathbf{x}) \neq \text{sign}(\mathbf{v}' \cdot \mathbf{x})]$, (2) we do not have a specific target threshold for the angle $\angle(\mathbf{v}, \mathbf{v}')$, but we need to provide testable bounds that involve $\angle(\mathbf{v}, \mathbf{v}')$ as a free parameter and (3) we can only tolerate a small constant multiplicative error factor $(1 \pm \mu)$. To obtain this improvement, we combine the approach of [GKSV24] with the notion of sandwiching polynomial approximators. Sandwiching polynomial approximators are also used to design testable learning algorithms (see [GKK23]), but we use them here in a more specialized way, by allowing the sandwiching function to be piecewise-polynomial.

In particular, we first observe that the region $\{\mathbf{x} \in \mathbb{R}^d : \text{sign}(\mathbf{v} \cdot \mathbf{x}) \neq \text{sign}(\mathbf{v}' \cdot \mathbf{x})\}$ can be approximated from above and from below by the disjoint union of a small number of simple regions (Figure 2). More precisely, if we let $\mathbf{v}_\perp$ be the unit vector in the direction $\mathbf{v}' - (\mathbf{v}' \cdot \mathbf{v})\mathbf{v}$, then we have

$$\mathbb{P}_{\mathbf{x} \sim S}\left[\mathbf{v} \cdot \mathbf{x} \geq 0 > \mathbf{v}' \cdot \mathbf{x}\right] \leq \sum_{i=0}^{\infty} \mathbb{P}_{\mathbf{x} \sim S}\left[\mathbf{v} \cdot \mathbf{x} \in [i\epsilon, (i+1)\epsilon], \mathbf{v}_\perp \cdot \mathbf{x} \leq -i\epsilon/\tan(\mathbf{v}, \mathbf{v}')\right] \quad (2.5)$$

$$\mathbb{P}_{\mathbf{x} \sim S}\left[\mathbf{v} \cdot \mathbf{x} \geq 0 > \mathbf{v}' \cdot \mathbf{x}\right] \geq \sum_{i=0}^{\infty} \mathbb{P}_{\mathbf{x} \sim S}\left[\mathbf{v} \cdot \mathbf{x} \in [i\epsilon, (i+1)\epsilon], \mathbf{v}_\perp \cdot \mathbf{x} \leq -(i+1)\epsilon/\tan(\mathbf{v}, \mathbf{v}')\right] \quad (2.6)$$

In fact, the number of interesting terms in the summations can be bounded by $O(\epsilon \log^{1/2}(1/\epsilon))$, because the remaining terms are testably negligible, due to Gaussian concentration and since we have access to $\mathbf{v}$.

Each term in the summations of (2.5), (2.6) is of the form $\mathbb{E}_{\mathbf{x} \sim S}[\mathcal{I}_i(\mathbf{x}) \cdot f_i(\mathbf{x})]$, where $f_i$ is some unknown halfspace and $\mathcal{I}_i(\mathbf{x}) = \mathbb{1}\{\mathbf{v} \cdot \mathbf{x} \in [i\epsilon, (i+1)\epsilon]\}$. Since we know $\mathbf{v}$, $\mathcal{I}_i(\mathbf{x})$ is a known quantity for all $\mathbf{x}$ in $S$. The quantities $f_i(\mathbf{x})$ are unknown, but we can effectively substitute them by polynomials, because, under the Gaussian distribution they admit low-degree sandwiching approximators. This allows us to provide a testable bound by matching the low degree Chow parameters of the functions $\mathcal{I}_i(\mathbf{x})$ under $\text{Unif}(S)$ to the corresponding Chow parameters under $\mathcal{N}_d$, due to the fact that polynomials are linear combinations of monomials and the number of low-degree monomials is small enough so that we can test them all.

Moment matching and Chow matching in particular are known to have applications in testable learning (see, e.g., [GKK23, RV23, GKSV24, KSV24b, CKK+24]), but here we have to use this tool in a careful way. We prove the following lemma (see Proposition D.6) based on a delicate argument that uses the sandwiching approximators of [DGJ+10, GOWZ10] (see Appendix D.3).

**Lemma 2.5** (Informal). *Let $C \geq 1$ be some sufficiently large constant. Suppose that for all $\alpha \in \mathbb{N}^d$ with $\|\alpha\|_1 \leq C/\mu^4$ and for all $i$:* $\left|\mathbb{E}_{\mathbf{x} \sim S}\left[\mathcal{I}_i(\mathbf{x}) \cdot \prod_{j \in [d]} x_j^{\alpha_j}\right] - \mathbb{E}_{\mathbf{x} \sim \mathcal{N}}\left[\mathcal{I}_i(\mathbf{x}) \cdot \prod_{j \in [d]} x_j^{\alpha_j}\right]\right| \leq \frac{\epsilon^2 \log(1/\epsilon)}{Cd^{C^2/\mu^4}}$. *Then, for all $\mathbf{v}'$,* $|\mathbb{P}_{\mathbf{x} \sim S}[\mathbf{v} \cdot \mathbf{x} \geq 0 > \mathbf{v}' \cdot \mathbf{x}] - \mathbb{P}_{\mathbf{x} \sim \mathcal{N}}[\mathbf{v} \cdot \mathbf{x} \geq 0 > \mathbf{v}' \cdot \mathbf{x}]| \leq \mu \frac{\angle(\mathbf{v}, \mathbf{v}')}{2\pi} + \epsilon$.

---

**Algorithm 1:** Disagreement tester

---

**Input:** $\epsilon, \delta, \mu \in (0,1)$, $\mathbf{v} \in \mathbb{S}^{d-1}$ and set $S$ of points in $\mathbb{R}^d$

Let $C \geq 1$ be a sufficiently large constant

Set $K = \frac{2}{\epsilon}\sqrt{\log(2/\epsilon)}$, $k = C/\mu^4$ and $\Delta = \frac{\epsilon}{CKd^{Ck}}$

**for** $\alpha \in \mathbb{N}^d$ *with* $\|\alpha\|_1 \leq k$ **do**

    **for** $i = -K, -K+1, \ldots, 0, 1, \ldots, K-1$ **do**

        Let $\mathcal{I}_i(\mathbf{x}) = \mathbb{1}\{i\epsilon \leq \mathbf{v} \cdot \mathbf{x} < (i+1)\epsilon\}$ for all $\mathbf{x} \in S$

        Let $\Delta_{i,\alpha} = \left| \mathbb{E}_{\mathbf{x}\sim S}[\prod_{j\in[d]} x_j^{\alpha_j} \mathcal{I}_i(\mathbf{x})] - \mathbb{E}_{\mathbf{x}\sim\mathcal{N}_d}[\prod_{j\in[d]} x_j^{\alpha_j} \mathcal{I}_i(\mathbf{x})] \right|$

    Let $\Delta_{\infty,\alpha} = \left| \mathbb{E}_{\mathbf{x}\sim S}[\prod_{j\in[d]} x_j^{\alpha_j} \mathbb{1}\{\mathbf{v}\cdot\mathbf{x} \geq K\epsilon\}] - \mathbb{E}_{\mathbf{x}\sim\mathcal{N}_d}[\prod_{j\in[d]} x_j^{\alpha_j} \mathbb{1}\{\mathbf{v}\cdot\mathbf{x} \geq K\epsilon\}] \right|$

    Let

    $\Delta_{-\infty,\alpha} = \left| \mathbb{E}_{\mathbf{x}\sim S}[\prod_{j\in[d]} x_j^{\alpha_j} \mathbb{1}\{\mathbf{v}\cdot\mathbf{x} \leq -K\epsilon\}] - \mathbb{E}_{\mathbf{x}\sim\mathcal{N}_d}[\prod_{j\in[d]} x_j^{\alpha_j} \mathbb{1}\{\mathbf{v}\cdot\mathbf{x} \leq -K\epsilon\}] \right|$

**if** for some $(i,\alpha)$ we have $\Delta_{i,\alpha} > \Delta$ **then** output Reject **else** output Accept

---

Based on the above lemma (and a symmetric argument for the case $\mathbf{v} \cdot \mathbf{x} < 0 \leq \mathbf{v}' \cdot \mathbf{x}$), the disagreement tester of Theorem 2.4 only needs to test quantities of the form $\mathbb{E}[\mathcal{I}_i \prod_j x_j^{\alpha_j}]$, which are known as constant-degree Chow parameters [Cho61, OS08] of the functions $\mathcal{I}_i(\mathbf{x})$, as described in Algorithm 1. Due to standard concentration arguments, if $S$ was i.i.d. from the Gaussian distribution, then the tests would pass.

**Approximate recovery of ground truth.** The final technical hurdle that remains unaddressed in the above derivation of the testable learning result for RCN is the fact that even under the target assumption, the ground-truth vector can be recovered only approximately. In particular, the following is true.

**Fact 2.6.** *For any $\epsilon', \delta \in (0,1)$ and $\eta_0 = 1/2 - c$, where $c > 0$ is any constant, there is an algorithm with time and sample complexity $\mathrm{poly}(d, 1/\epsilon')\log(1/\delta)$ that has access to an example oracle $\mathsf{EX}_{\mathcal{N}, f^*, \eta_0}^{\mathsf{RCN}}$ for some unknown $f^*(\mathbf{x}) = \mathsf{sign}(\mathbf{v}^* \cdot \mathbf{x})$, $\mathbf{v}^* \in \mathbb{S}^{d-1}$ and outputs $\mathbf{v} \in \mathbb{S}^{d-1}$ such that $\angle(\mathbf{v}^*, \mathbf{v}) \leq \epsilon'$, with probability at least $1 - \delta$.*

The place where we have to be more careful is when we argue that the distribution of $\mathbf{x}$ conditioned on $y \neq \mathsf{sign}(\mathbf{v} \cdot \mathbf{x})$ is Gaussian, under the target assumption. This is not true anymore, because $\mathbf{v}$ is not necessarily equal to $\mathbf{v}^*$ and, therefore, the event $y \neq \mathsf{sign}(\mathbf{v} \cdot \mathbf{x})$ does not coincide with the event $y \neq \mathsf{sign}(\mathbf{v}^* \cdot \mathbf{x})$, which, due to the definition of RCN noise, is independent from $\mathbf{x}$. However, the only case that these two events do not coincide is when $\angle(\mathbf{x}, \mathbf{v}) \leq O(\epsilon' d)$, due to the guarantee of Fact 2.6 that $\mathbf{v}$ and $\mathbf{v}^*$ are geometrically close. Since we have access to both $S$ and $\mathbf{v}$, we may directly test whether $\mathbb{P}_{\mathbf{x}\sim S}[\angle(\mathbf{x}, \mathbf{v}) \leq O(\epsilon' d)]$ is bounded by $O(\epsilon)$, as would be the case under the target assumption, if $\epsilon'$ is chosen to be $\mathrm{poly}(\epsilon/d)$. Therefore, this event is certifiably negligible and the initial argument goes through.

## Acknowledgments and Disclosure of Funding

Surbhi Goel was supported by OpenAI Superalignment Fast Grant and by NSF award CCF-2504016. Part of this work was conducted while the author was visiting the Simons Institute for the Theory of Computing. Adam Klivans was supported by NSF awards AF-1909204, CCF-2504016 and the NSF AI Institute for Foundations of Machine Learning (IFML). Konstantinos Stavropoulos was supported by the NSF AI Institute for Foundations of Machine Learning (IFML), by scholarships from Bodossaki Foundation and Leventis Foundation, and by the Apple Scholars in AI/ML PhD fellowship. Arsen Vasilyan was supported in part by NSF awards CCF-2006664, DMS-2022448, CCF-1565235, CCF-1955217, CCF-2310818, the NSF AI Institute for Foundations of Machine Learning (IFML) Big George Fellowship and Fintech@CSAIL. Part of this work was conducted while the author was visiting the Simons Institute for the Theory of Computing.

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

# A Preliminaries

## A.1 Some standard notation.

When we say $a = b \pm c$ we mean that $a$ in in the interval $[b - c, b + c]$. When we say a polynomial is "degree-$k$" we mean that the degree polynomial is *at most* $k$. For a vector $\mathbf{x}$ in $\mathbb{R}^d$, let $\mathbf{x}^{\otimes k}$ denote the $\binom{d+1}{k}$-dimensional vector whose elements are of the form $\mathbf{x}^\alpha = \prod_{j \in [d]} x_j^{\alpha_j}$ for $\alpha \in \mathbb{N}^d$ with $\|\alpha\|_1 = \sum_j \alpha_j \le k$. In other words, $\mathbf{x}^{\otimes k}$ is the vector one gets by evaluating all multidimensional monomials of degree at most $k$ on input $\mathbf{x}$. We also view degree-$k$ polynomials as corresponding to elements $\mathbb{R}^{\binom{d+1}{k}}$, i.e. their coefficient vectors. Using this notation we have

$$p(\mathbf{x}) = p \cdot \mathbf{x}^{\otimes k}.$$

Additionally, for a degree-$k$ polynomial $p$ over $\mathbb{R}^d$, say $p(\mathbf{x}) = \sum_{\alpha:\|\alpha\|_1 \le k} p_\alpha \mathbf{x}^\alpha$, we will use the notation $\|p\|_{\mathrm{coeff}}$ to denote the 2-norm of the coefficients of $p$, specifically

$$\|p\|_{\mathrm{coef}} := \left( \sum_{\alpha \in \mathbb{N}^d} p_\alpha^2 \right)^{1/2}$$

Note that if the largest in absolute value coefficient of polynomial $p$ has absolute value $B$, then we have

$$B \le \|p\|_{\mathrm{coeff}} \le B \, (d+1)^{k/2} . \tag{A.1}$$

In this work we will use the convention that $\mathrm{sign}(0) = 1$.

## A.2 Standard lemmas.

We will also need the following lemma:

**Lemma A.1.** *Let $\mathcal{H}$ be a collection of subsets of $\mathbb{R}^d$ of non-zero VC dimension $\Delta_{VC}$ and let $S$ be a collection of $N$ i.i.d. samples from $\mathcal{N}(0, I_d)$. Then, with probability at least $1 - \delta$ for every polynomial $p$ of degree at most $k$ with coefficients bounded by $B$ in absolute value and all sets $A$ in $\mathcal{H}$ we have*

$$\left| \mathop{\mathbb{E}}_{\mathbf{x} \sim S} [p(\mathbf{x}) \mathbb{1}_{\mathbf{x} \in A}] - \mathop{\mathbb{E}}_{\mathbf{x} \sim \mathcal{N}(0, I_d)} [p(\mathbf{x}) \mathbb{1}_{\mathbf{x} \in A}] \right| \le \frac{60(2k)^{k+2}(d+1)^k \Delta_{VC}}{\delta} \left( \frac{\log N}{N} \right)^{1/4},$$

$$\left| \mathop{\mathbb{E}}_{\mathbf{x} \sim S} \left[ (p(\mathbf{x}))^2 \mathbb{1}_{\mathbf{x} \in A} \right] - \mathop{\mathbb{E}}_{\mathbf{x} \sim \mathcal{N}(0, I_d)} \left[ (p(\mathbf{x}))^2 \mathbb{1}_{\mathbf{x} \in A} \right] \right| \le \frac{60 B^2 (4k)^{2k+2}(d+1)^{6k} \Delta_{VC}}{\delta} \left( \frac{\log N}{N} \right)^{1/4} .$$

*Proof.* Let $R$ be a positive real number, to be set later. For any monomial $m$ over $\mathbb{R}^d$ of degree at most $k$, we can decompose

$$m(\mathbf{x}) \mathbb{1}_{\mathbf{x} \in A} = m(\mathbf{x}) \mathbb{1}_{\mathbf{x} \in A \wedge |m(\mathbf{x})| \le R} \pm |m(\mathbf{x})| \mathbb{1}_{|m(\mathbf{x})| > R}$$

This allows us to bound the quantity in our lemma in the following way:

$$\left| \mathop{\mathbb{E}}_{\mathbf{x} \sim S} [m(\mathbf{x}) \mathbb{1}_{\mathbf{x} \in A}] - \mathop{\mathbb{E}}_{\mathbf{x} \sim \mathcal{N}(0, I_d)} [m(\mathbf{x}) \mathbb{1}_{\mathbf{x} \in A}] \right| =$$

$$\left| \mathop{\mathbb{E}}_{\mathbf{x} \sim S} \left[ |m(\mathbf{x})| \mathbb{1}_{\mathbf{x} \in A \wedge |m(\mathbf{x})| \le R \wedge m(\mathbf{x}) > 0} \right] - \mathop{\mathbb{E}}_{\mathbf{x} \sim S} \left[ |m(\mathbf{x})| \mathbb{1}_{\mathbf{x} \in A \wedge |m(\mathbf{x})| \le R \wedge m(\mathbf{x}) < 0} \right] \right|$$

$$\pm \left( \mathop{\mathbb{E}}_{\mathbf{x} \sim S} \left[ |m(\mathbf{x})| \mathbb{1}_{|m(\mathbf{x})| > R} \right] + \mathop{\mathbb{E}}_{\mathbf{x} \sim \mathcal{N}(0, I_d)} \left[ |m(\mathbf{x})| \mathbb{1}_{|m(\mathbf{x})| > R} \right] \right) \quad \text{(A.2)}$$

We start by considering the first term above:

$$\left| \mathop{\mathbb{E}}_{\mathbf{x}\sim S} \left[ m(\mathbf{x})\mathbb{1}_{\mathbf{x}\in A \wedge |m(\mathbf{x})|\leq R \wedge m(\mathbf{x})>0} \right] - \mathop{\mathbb{E}}_{\mathbf{x}\sim\mathcal{N}(0,I_d)} \left[ m(\mathbf{x})\mathbb{1}_{\mathbf{x}\in A \wedge |m(\mathbf{x})|\leq R \wedge m(\mathbf{x})>0} \right] \right| =$$

$$\left| \int_0^R \mathop{\mathbb{P}}_{\mathbf{x}\sim S} \left[ m(\mathbf{x})\mathbb{1}_{\mathbf{x}\in A \wedge |m(\mathbf{x})|\leq R} \geq z \right] dz - \int_0^R \mathop{\mathbb{P}}_{\mathbf{x}\sim\mathcal{N}(0,I_d)} \left[ m(\mathbf{x})\mathbb{1}_{\mathbf{x}\in A \wedge |m(\mathbf{x})|\leq R} \geq z \right] dz \right| \leq$$

$$\int_0^R \left| \mathop{\mathbb{P}}_{\mathbf{x}\sim S} \left[ m(\mathbf{x})\mathbb{1}_{\mathbf{x}\in A \wedge |m(\mathbf{x})|\leq R} \geq z \right] - \mathop{\mathbb{P}}_{\mathbf{x}\sim\mathcal{N}(0,I_d)} \left[ m(\mathbf{x})\mathbb{1}_{\mathbf{x}\in A \wedge |m(\mathbf{x})|\leq R} \geq z \right] \right| dz \leq$$

$$R \max_{z\in[0,R]} \left| \mathop{\mathbb{P}}_{\mathbf{x}\sim S} \left[ z \leq m(\mathbf{x}) \leq R \wedge \mathbf{x} \in A \right] - \mathop{\mathbb{P}}_{\mathbf{x}\sim\mathcal{N}(0,I_d)} \left[ z \leq m(\mathbf{x}) \leq R \wedge \mathbf{x} \in A \right] \right| \quad \text{(A.3)}$$

To bound the right side of Equation A.3, consider the class $\mathcal{G}$ of $\{0,1\}$-valued functions of the form $\mathbb{1}_{z\leq p(\mathbf{x}) \leq R \wedge \mathbf{x}\in A}$, where $A$ is a set in $\mathcal{H}$ and $p$ is a polynomial in $d$ dimensions of degree at most $d$. Recall that the VC dimension of degree-$k$ polynomial threshold functions is at most $(d+1)^k$. From the Sauer-Shelah lemma, it follows that the VC dimension of $\mathcal{G}$ is at most $10((d+1)^k + \Delta_{\text{VC}})$. Combining it with the standard VC bound, we see that with probability at least $1 - \delta/4$ for all monomials $m$ of degree at most $k$ we have

$$\left| \mathop{\mathbb{P}}_{\mathbf{x}\sim S} \left[ z \leq m(\mathbf{x}) \leq R \wedge \mathbf{x} \in A \right] - \mathop{\mathbb{P}}_{\mathbf{x}\sim\mathcal{N}} \left[ z \leq m(\mathbf{x}) \leq R \wedge \mathbf{x} \in A \right] \right| \leq \left( \frac{100((d+1)^k + \Delta_{\text{VC}})\log N}{N\delta} \right)^{\frac{1}{2}}$$

Combining this with Equation A.3 we get:

$$\left| \mathop{\mathbb{E}}_{\mathbf{x}\sim S} \left[ m(\mathbf{x})\mathbb{1}_{\mathbf{x}\in A \wedge |m(\mathbf{x})|\leq R \wedge m(\mathbf{x})>0} \right] - \mathop{\mathbb{E}}_{\mathbf{x}\sim\mathcal{N}(0,I_d)} \left[ m(\mathbf{x})\mathbb{1}_{\mathbf{x}\in A \wedge |m(\mathbf{x})|\leq R \wedge m(\mathbf{x})>0} \right] \right| \leq$$

$$10R\sqrt{\frac{((d+1)^k + \Delta_{\text{VC}})\log N}{N\delta}} \quad \text{(A.4)}$$

We now proceed to bounding the second term and the third terms in Equation A.2. If we express $m(\mathbf{x})$ as $\prod_j x_j^{i_j}$, we see that each $i_j$ is at most $k$ and there are at most $k$ values of $j$ for which the power $i_j$ is non-zero. This implies

$$\mathop{\mathbb{E}}_{\mathbf{x}\sim\mathcal{N}(0,I_d)} \left[ (m(\mathbf{x}))^2 \right] \leq 2k \cdot (2k)!! \leq (2k)^{k+2}. \quad \text{(A.5)}$$

Let $\delta'$ be a real number between 0 and 1, value of which will be chosen later. The Markov's inequality implies that with probability at least $1 - \delta'$ we have

$$\mathop{\mathbb{E}}_{\mathbf{x}\sim S} \left[ |m(\mathbf{x})|\mathbb{1}_{|m(\mathbf{x})|>R} \right] \leq \frac{\mathbb{E}_{\mathbf{x}\sim\mathcal{N}(0,I_d)} \left[ |m(\mathbf{x})|\mathbb{1}_{|m(\mathbf{x})|>R} \right]}{\delta'} \leq \frac{\mathbb{E}_{\mathbf{x}\sim\mathcal{N}(0,I_d)} \left[ |m(\mathbf{x})|^2 \right]}{R\delta'} \leq \frac{(2k)^{k+2}}{R\delta'}$$
(A.6)

We also note that

$$\mathop{\mathbb{E}}_{\mathbf{x}\sim\mathcal{N}(0,I_d)} \left[ |m(\mathbf{x})|\mathbb{1}_{|m(\mathbf{x})|>R} \right] \leq \frac{\mathbb{E}_{\mathbf{x}\sim\mathcal{N}(0,I_d)} \left[ |m(\mathbf{x})|^2 \right]}{R} \leq \frac{(2k)^{k+2}}{R} \quad \text{(A.7)}$$

Overall, substituting Equations A.4, A.6 and A.7 into Equation A.2 we get

$$\left| \mathop{\mathbb{E}}_{\mathbf{x}\sim S} \left[ m(\mathbf{x})\mathbb{1}_{\mathbf{x}\in A} \right] - \mathop{\mathbb{E}}_{\mathbf{x}\sim\mathcal{N}(0,I_d)} \left[ m(\mathbf{x})\mathbb{1}_{\mathbf{x}\in A} \right] \right| \leq 10R\sqrt{\frac{((d+1)^k + \Delta_{\text{VC}})\log N}{N\delta}} + 2\frac{(2k)^{k+2}}{R\delta'}.$$

Choosing $R$ to balance the two terms above, we get

$$\left| \mathop{\mathbb{E}}_{\mathbf{x}\sim S} \left[ m(\mathbf{x})\mathbb{1}_{\mathbf{x}\in A} \right] - \mathop{\mathbb{E}}_{\mathbf{x}\sim\mathcal{N}(0,I_d)} \left[ m(\mathbf{x})\mathbb{1}_{\mathbf{x}\in A} \right] \right| \leq \sqrt{80\frac{(2k)^{k+2}}{\delta'}\sqrt{\frac{((d+1)^k + \Delta_{\text{VC}})\log N}{N\delta}}}.$$

Taking $\delta' = \frac{\delta}{4(d+1)^k}$ and taking a union bound to insure that Equation A.6 holds for all monomials $m$ of degree at most $k$, we see that with probability at least $1 - \delta/2$ it is the case that all monomials

$m$ of degree at most $k$ and all $A$ in $\mathcal{H}$ it is the case that

$$\left| \mathop{\mathbb{E}}_{\mathbf{x} \sim S}[m(\mathbf{x}) \mathbb{1}_{\mathbf{x} \in A}] - \mathop{\mathbb{E}}_{\mathbf{x} \sim \mathcal{N}(0, I_d)}[m(\mathbf{x}) \mathbb{1}_{\mathbf{x} \in A}] \right| \leq \sqrt{320 \frac{(2k)^{k+2}(d+1)^k}{\delta^{3/2}}} \sqrt{\frac{((d+1)^k + \Delta_{\mathrm{VC}}) \log N}{N}} \leq$$
$$\frac{60(2k)^{k+2}(d+1)^k \Delta_{\mathrm{VC}}}{\delta} \left( \frac{\log N}{N} \right)^{1/4}. \quad \text{(A.8)}$$

Recall again that there are at most $(d+1)^k$ degree-$k$ monomials $m$. This allows us to combine Equation A.8 with the triangle inequality to conclude that with probability at least $1 - \delta/2$ for every polynomial $p$ of degree at most $k$ with coefficients bounded by $B$ in absolute value and for all $A$ in $\mathcal{H}$ we have

$$\left| \mathop{\mathbb{E}}_{\mathbf{x} \sim S}[p(\mathbf{x}) \mathbb{1}_{\mathbf{x} \in A}] - \mathop{\mathbb{E}}_{\mathbf{x} \sim \mathcal{N}(0, I_d)}[p(\mathbf{x}) \mathbb{1}_{\mathbf{x} \in A}] \right| \leq \frac{60B(2k)^{k+2}(d+1)^{2k} \Delta_{\mathrm{VC}}}{\delta} \left( \frac{\log N}{N} \right)^{1/4}.$$

The polynomial $p^2$ has a degree of at most $2k$ and each coefficient of $p^2$ is bounded by $B^2(d+1)^{2k}$. Therefore, with probability at least $1 - \delta/2$ for every polynomial $p$ of degree at most $k$ with coefficients bounded by $B$ in absolute value and for all $A$ in $\mathcal{H}$ we have

$$\left| \mathop{\mathbb{E}}_{\mathbf{x} \sim S}\left[ (p(\mathbf{x}))^2 \mathbb{1}_{\mathbf{x} \in A} \right] - \mathop{\mathbb{E}}_{\mathbf{x} \sim \mathcal{N}(0, I_d)}\left[ (p(\mathbf{x}))^2 \mathbb{1}_{\mathbf{x} \in A} \right] \right| \leq \frac{60B^2(4k)^{2k+2}(d+1)^{6k} \Delta_{\mathrm{VC}}}{\delta} \left( \frac{\log N}{N} \right)^{1/4},$$

which completes the proof. $\qquad\square$

## B  Testable Learning with respect to Massart Noise Oracles

We now turn to the more challenging task of testable learning with respect to Gaussian Massart oracles and state our main theorem which shows that there is a fully polynomial-time algorithm even in this case. Observe that, informally, for some ground-truth halfspace $f$, the Massart oracle $\mathsf{EX}^{\mathsf{Massart}}_{\mathcal{N}, f, \eta_0}$ can be equivalently viewed as follows: The oracle outputs a Gaussian example $\mathbf{x}$, and with probability $\eta_0$ the adversary is given an *option* to make the accompanying label incorrect (i.e. $-f(\mathbf{x})$), and otherwise the label is correct (i.e. $f(\mathbf{x})$). The Massart noise model is known to be more challenging that the RCN model, because the label noise (in general) does not have symmetry properties that can be harnessed to make error terms coming from different regions cancel each other out.

**Theorem B.1** (Main Result). *Let $c \in (0, 1/2)$ be any constant and $\eta_0 = 1/2 - c$. Then, there is an algorithm that testably learns the class $\mathcal{H}_{\mathsf{hs}}$ with respect to $\mathsf{EX}^{\mathsf{Massart}}_{\mathcal{N}, \mathcal{H}_{\mathsf{hs}}, \eta_0} = \{\mathsf{EX}^{\mathsf{Massart}}_{\mathcal{N}, f, \eta} : f \in \mathcal{H}_{\mathsf{hs}}, \sup_{\mathbf{x} \in \mathbb{R}^d} \eta(\mathbf{x}) \leq \eta_0\}$ with time and sample complexity $\mathrm{poly}(d, 1/\epsilon) \log(1/\delta)$.*

*Moreover, even if the input set $\bar{S}$ is arbitrary (not necessarily i.i.d.), whenever the algorithm accepts, it outputs $h \in \mathcal{H}_{\mathsf{hs}}$ such that $\mathbb{P}_{(\mathbf{x},y) \sim \bar{S}}[y \neq h(\mathbf{x})] \leq \mathsf{opt}_{\bar{S}} + \epsilon$, where $\mathsf{opt}_{\bar{S}} = \min_{f \in \mathcal{H}_{\mathsf{hs}}} \mathbb{P}_{(\mathbf{x},y) \sim \bar{S}}[y \neq f(\mathbf{x})]$.*

The final part of Theorem B.1 states that the guarantee we achieve is actually stronger than the one in Definition 1.3, since the output is near-optimal whenever the algorithm accepts, without requiring that the input $\bar{S}$ consists of independent examples. For small $c$, the runtime of our algorithm scales as $(d/\epsilon)^{\mathrm{poly}(1/c)}$. This gives a polynomial-time algorithm when $c$ is constant, but we leave it as an interesting open question whether the dependence on $1/c$ can be improved. We provide lower bounds for the case $c = 0$ in Appendix C.

The proof of Theorem B.1 follows the same outline we provided for the case of random classification noise. However, there are two differences. First, we need a version of Fact 2.6 that works under Massart noise (and Gaussian marginal) and gives an algorithm that approximately recovers the parameters of the ground truth. Second, the disagreement tester from before does not give a testable bound for the quantity $|\bar{S}_b|/|\bar{S}| = \mathbb{P}_{(\mathbf{x},y) \sim \bar{S}}[y \neq \mathsf{sign}(\mathbf{v} \cdot \mathbf{x})$ and $\mathsf{sign}(\mathbf{v}^* \cdot \mathbf{x}) \neq \mathsf{sign}(\mathbf{v} \cdot \mathbf{x})]$ anymore. Even if we assume once more that under the target assumption we have exact recovery (i.e., $\mathbf{v} = \mathbf{v}^*$), the event $y \neq \mathsf{sign}(\mathbf{v}^* \cdot \mathbf{x})$ is not independent from $\mathbf{x}$ and, if we used the disagreement tester

of Theorem 2.4, the completeness criterion would not necessarily be satisfied under the target assumption.

Fortunately, the first difference is not an issue, since appropriate results are known from prior work on classical learning under Massart noise and Gaussian marginal (see, e.g., [ABL17, DKTZ20a]).

**Fact B.2** ([DKTZ20a]). *For any $\epsilon', \delta \in (0, 1)$ and $\eta_0 = 1/2 - c$, where $c > 0$ is any constant, there is an algorithm with time and sample complexity $\mathrm{poly}(d, 1/\epsilon) \log(1/\delta)$ that has access to an example oracle in $\mathsf{EX}^{\mathsf{Massart}}_{\mathcal{N}, f^*, \eta_0}$ for some unknown $f^*(\mathbf{x}) = \mathrm{sign}(\mathbf{v}^* \cdot \mathbf{x})$, $\mathbf{v}^* \in \mathbb{S}^{d-1}$ and outputs $\mathbf{v} \in \mathbb{S}^{d-1}$ s.t. $\angle(\mathbf{v}^*, \mathbf{v}) \le \epsilon'$, with probability at least $1 - \delta$.*

**The spectral tester and how it is applied.** In order to resolve the second complication and provide a certificate bounding the quantity $|\bar{S}_b|/|\bar{S}|$, we follow a different testing approach, based on ideas from tolerant testable learning [GSSV24], where the testers must accept whenever the input distribution is close to the target (and not necessarily equal). We provide the following tester which is guaranteed to accept subsets of Gaussian samples, since it is monotone under datapoint removal.

**Theorem B.3** (Spectral tester, see Theorem E.1). *Let $\mu \in (0, 1)$ be any constant. There is an algorithm (Algorithm 2) that receives $\epsilon, \delta \in (0, 1)$, $U \in \mathbb{N}$, $\mathbf{v} \in \mathbb{S}^{d-1}$ and a set $S$ of points in $\mathbb{R}^d$, runs in time $\mathrm{poly}(d, 1/\epsilon, |S|)$ and then either outputs* Reject *or* Accept, *satisfying the following specifications.*

1. *(Soundness) If the algorithm accepts and $|S| \le U$, then the following is true for any $\mathbf{v}' \in \mathbb{S}^{d-1}$*

$$\frac{1}{U} \sum_{\mathbf{x} \in S} \mathbb{1}\{\mathrm{sign}(\mathbf{v} \cdot \mathbf{x}) \ne \mathrm{sign}(\mathbf{v}' \cdot \mathbf{x})\} \le (1 + \mu)\angle(\mathbf{v}, \mathbf{v}')/\pi + \epsilon$$

2. *(Completeness) If $S$ consists of at least $(\frac{Cd}{\epsilon\delta})^C$ i.i.d. examples from $\mathcal{N}_d$, where $C \ge 1$ is some sufficiently large constant depending on $\mu$, then the algorithm accepts with probability at least $1 - \delta$.*

3. *(Monotonicity under removal) If the algorithm accepts on input $(\epsilon, \delta, U, \mathbf{v}, S)$ and $S'$ is such that $S' \subseteq S$, then the algorithm also accepts on input $(\epsilon, \delta, U, \mathbf{v}, S')$.*

Given this tool, we are able to obtain a testable bound for $|\bar{S}_b|/|\bar{S}|$. Recall that the set $\bar{S}_b$ is the set of points $(\mathbf{x}, y)$ in $\bar{S}$ such that $y \ne \mathrm{sign}(\mathbf{v} \cdot \mathbf{x})$ and $\mathrm{sign}(\mathbf{v}^* \cdot \mathbf{x}) \ne \mathrm{sign}(\mathbf{v} \cdot \mathbf{x})$. For the soundness, observe that $|\bar{S}_b|/|\bar{S}| = \frac{U}{|\bar{S}|} \cdot \frac{1}{U} \sum_{\mathbf{x} \in S_{\mathsf{False}}} \mathbb{1}\{\mathrm{sign}(\mathbf{v} \cdot \mathbf{x}) \ne \mathrm{sign}(\mathbf{v}^* \cdot \mathbf{x})\}$, where $\bar{S}_{\mathsf{False}} = \{(\mathbf{x}, y) \in \bar{S} : y \ne \mathrm{sign}(\mathbf{v} \cdot \mathbf{x})\}$. The soundness condition of Theorem B.3 gives us that $|\bar{S}_b|/|\bar{S}| \le \frac{U}{|\bar{S}|}(1 + \mu)\angle(\mathbf{v}, \mathbf{v}^*) + \epsilon$, as long as $|S_{\mathsf{False}}| \le U$. The quantity $S_{\mathsf{False}}$ can be testably bounded by $(1/2 - c)|\bar{S}|$, since $S_{\mathsf{False}}$ is defined with respect to $\mathbf{v}$ and we can therefore pick $U = (1/2 - c)|\bar{S}|$. Overall, we obtain the same bound for $|\bar{S}_b|/|\bar{S}|$ as in the RCN case.

In order to show that our test will accept under the target assumption, the main observation is that we can interpret the Massart noise oracle with noise rate $\eta_0$ as follows: To form the input set $\bar{S}$, the oracle first calls the RCN oracle of rate $\eta_0$ to form a set $\bar{S}^{\mathsf{RCN}}$. Let $\bar{S}^{\mathsf{RCN}}_{\mathsf{False}}$ be the subset of $\bar{S}^{\mathsf{RCN}}$ such that $y \ne \mathrm{sign}(\mathbf{v}^* \cdot \mathbf{x})$. The Massart noise oracle then flips the labels of some elements $\bar{S}^{\mathsf{RCN}}_{\mathsf{False}}$ back to match the ground-truth label. In other words, we have that $\bar{S}_{\mathsf{False}} \subseteq \bar{S}^{\mathsf{RCN}}_{\mathsf{False}}$ (assuming $\mathbf{v} = \mathbf{v}^*$). Observe that $S^{\mathsf{RCN}}_{\mathsf{False}}$ is drawn according to the distribution of $\mathbf{x}$ conditioned on $y \ne \mathrm{sign}(\mathbf{v}^* \cdot \mathbf{x})$ and is, therefore, an i.i.d. Gaussian sample.

**Designing the spectral tester.** Theorem B.3 follows from a combination of ideas used to prove Theorem 2.4 and the spectral testing approach of [GSSV24]. In particular, instead of matching the Chow parameters $\mathbb{E}_{\mathbf{x} \sim S}[\mathcal{I}_i(\mathbf{x}) \prod_{j \in [d]} \mathbf{x}^{\alpha_i}]$ of the quantities $\mathcal{I}_i$ as in Algorithm 1, we bound the maximum singular value of the Chow parameter matrices $\mathbb{E}_{\mathbf{x} \sim S}[(\mathbf{x}^{\otimes k})(\mathbf{x}^{\otimes k})^\top \mathcal{I}_i(\mathbf{x})]$, where $\mathbf{x}^{\otimes k}$ denotes the vector of monomials of degree at most $k$. This can be done efficiently via the SVD algorithm and, crucially, satisfies the monotonicity under removal property of Theorem B.3. Moreover, once the Chow parameter matrix is bounded, we have a bound for all quantities of the form $\mathbb{E}_{\mathbf{x} \sim S}[(p(\mathbf{x}))^2 \mathcal{I}_i(\mathbf{x})]$, where $p$ is of degree at most $k$. Combining this observation with an analysis similar to the one for Lemma 2.5 (see Propositions E.4 and E.5) and a stronger version of

**Algorithm 2:** Spectral tester

---

**Input:** $\epsilon, \delta, \mu \in (0, 1)$, $\mathbf{v} \in \mathbb{S}^{d-1}$ and set $S$ of points in $\mathbb{R}^d$

Let $C \geq 1$ be a sufficiently large constant

Set $K = \frac{2}{\epsilon}\sqrt{\log(2/\epsilon)}$, $k = C/\mu^5$ and $\Delta = \frac{\epsilon^2}{CKd^{Ck}}$

**for** $i = -K, -K+1, \ldots, 0, 1, \ldots, K-1$ **do**

> Let $\mathcal{I}_i(\mathbf{x}) = \mathbb{1}\{i\epsilon \leq \mathbf{v} \cdot \mathbf{x} < (i+1)\epsilon\}$ for all $\mathbf{x} \in S$
>
> **if** $\mathbb{E}_{\mathbf{x} \sim S}[(\mathbf{x}^{\otimes k})(\mathbf{x}^{\otimes k})^\top \mathcal{I}_i(\mathbf{x})] \preceq \mathbb{E}_{\mathbf{x} \sim \mathcal{N}}[(\mathbf{x}^{\otimes k})(\mathbf{x}^{\otimes k})^\top \mathcal{I}_i(\mathbf{x})] + \Delta I$ **then** continue
>
> **else** output Reject

**if** $\mathbb{E}_{\mathbf{x} \sim S}[(\mathbf{x}^{\otimes k})(\mathbf{x}^{\otimes k})^\top \mathbb{1}_{\{\mathbf{v} \cdot \mathbf{x} \geq K\epsilon\}}] \preceq \mathbb{E}_{\mathbf{x} \sim \mathcal{N}}[(\mathbf{x}^{\otimes k})(\mathbf{x}^{\otimes k})^\top \mathbb{1}_{\{\mathbf{v} \cdot \mathbf{x} \geq K\epsilon\}}] + \Delta I$ **then** continue

**else** output Reject

**if** $\mathbb{E}_{\mathbf{x} \sim S}[(\mathbf{x}^{\otimes k})(\mathbf{x}^{\otimes k})^\top \mathbb{1}_{\{\mathbf{v} \cdot \mathbf{x} \leq -K\epsilon\}}] \preceq \mathbb{E}_{\mathbf{x} \sim \mathcal{N}}[(\mathbf{x}^{\otimes k})(\mathbf{x}^{\otimes k})^\top \mathbb{1}_{\{\mathbf{v} \cdot \mathbf{x} \leq -K\epsilon\}}] + \Delta I$ **then** continue

**else** output Reject

Output Accept

---

the sandwiching polynomials of [DGJ⁺10, GOWZ10] by [KSV24b], we obtain that Algorithm 2 satisfies Theorem B.3.

**Overall algorithm.** The overall algorithm receives an input set of labeled examples $\bar{S}$ and obtains a candidate $\mathbf{v} \in \mathbb{S}^{d-1}$ by running the algorithm of [DKTZ20a] with parameter $\epsilon' = \epsilon^{3/2}/(C\sqrt{d})$ for some large enough constant $C$ (see Fact B.2). Then, it runs the disagreement tester of Theorem 2.4 with parameters $(S, \mathbf{v}, \epsilon, \delta, \mu)$ (for some small constant $\mu$ depending on the noise rate $\eta_0 = 1/2 - c$). Subsequently, the tester checks whether $\mathbb{P}_{(\mathbf{x},y)\sim\bar{S}}[y \neq \text{sign}(\mathbf{v} \cdot \mathbf{x})]$ is at most $1/2 - c/2$.

Finally, it splits the set $\bar{S}_{\text{False}} = \{(\mathbf{x}, y) \in \bar{S} : y \neq \text{sign}(\mathbf{v} \cdot \mathbf{x})\}$ in two parts as follows.

$$\bar{S}_{\text{False}}^{\text{far}} = \left\{(\mathbf{x}, y) \in \bar{S}_{\text{False}} : |\measuredangle(\mathbf{v}, \mathbf{x}) - \pi/2| > \epsilon^{3/2}/(d-1)^{1/2}\right\} \quad \text{and} \quad \bar{S}_{\text{False}}^{\text{near}} = \bar{S}_{\text{False}} \setminus \bar{S}_{\text{False}}^{\text{far}}$$

For $\bar{S}_{\text{False}}^{\text{near}}$, it checks that it contains at most $O(\epsilon)|\bar{S}|$ elements, while for $\bar{S}_{\text{False}}^{\text{far}}$, it runs the spectral tester of Theorem B.3 with inputs $(U = (1/2 - c/2)|\bar{S}|, S = S_{\text{False}}^{\text{far}}, \mathbf{v}, \epsilon, \delta, \mu)$.

## C  Lower Bounds in the High-Noise Regime

**Notation.** The set $\mathbb{Z}_q$ equals to $\{0, 1, 2, \ldots, q-1\}$. We denote with $\mathcal{N}_d(\mu, \Sigma; S)$ the Gaussian distribution in $d$ dimensions with mean $\mu \in \mathbb{R}^d$ and covariance matrix $\Sigma \in \mathbb{R}^{d \times d}$, truncated on the set $S \subseteq \mathbb{R}^d$.

We show that there is no efficient tester-learner that accepts whenever the input dataset is generated by Gaussian examples with random classification noise (RCN) of rate $1/2$. We give both cryptographic lower bounds, as well as lower bounds in the statistical query (SQ) framework that match the best known bounds for classical (non-testable) learning under adversarial label noise. Since RCN noise is a special type of Massart noise, where all of the labels are flipped with the same rate (i.e., $\eta(\mathbf{x})$ is constant), the lower bounds we give also imply lower bounds for the case of Massart noise of rate $1/2$ (which is also called *strong* Massart noise). Recall that random classification noise is defined in Definition 2.1.

The hard distributions for learning under adversarial label noise proposed by [DKPZ21, Tie23, DKR23] are all indistinguishable from the distribution generated by the oracle $\text{EX}_{\mathcal{N}, 1/2}^{\text{RCN}}$. Using this fact, we obtain our lower bounds by the following simple observation that any tester-learner that accepts $\text{EX}_{\mathcal{N}, 1/2}^{\text{RCN}}$ can distinguish between $\text{EX}_{\mathcal{N}, 1/2}^{\text{RCN}}$ and any distribution where the value of opt is non-trivial.

**Observation C.1.** *Let $\mathcal{H} \subseteq \{\mathbb{R}^d \to \{\pm 1\}\}$ be a concept class, $\tau \in (0, 1/8)$ and suppose that algorithm $\mathcal{A}$ testably learns $\mathcal{H}$ with respect to $\text{EX}_{\mathcal{N}, 1/2}^{\text{RCN}}$ up to excess error $\epsilon \in (0, 1/4)$ and probability of failure $\delta = 1/6$. Let $\mathfrak{D}_g$ be the class of distributions over $\mathbb{R}^d \times \{\pm 1\}$ such that the marginal on $\mathbb{R}^d$ is $\mathcal{N}_d$ and $\min_{f \in \mathcal{H}} \mathbb{P}[y \neq f(\mathbf{x})] \leq \frac{1}{2} - \epsilon - 2\tau$. Then, there is an algorithm $\mathcal{A}'$ that calls $\mathcal{A}$ once and uses additional time $\text{poly}(d, 1/\tau)$ such that $|\mathbb{P}[\mathcal{A}'(\mathcal{N}_d \times \text{Unif}(\{\pm 1\})) = 1] - \mathbb{P}[\mathcal{A}'(\mathcal{D}_{\mathbf{x},y}) = 1]| \geq 1/3$ for any $\mathcal{D}_{\mathbf{x},y} \in \mathfrak{D}_g$.*

*Proof.* Let $\mathcal{D}_{\mathbf{x},y}$ be the input distribution. The algorithm $\mathcal{A}'(\mathcal{D}_{\mathbf{x},y})$ calls $\mathcal{A}(\mathcal{D}_{\mathbf{x},y})$ once and then:

- If $\mathcal{A}$ outputs Reject, then $\mathcal{A}'$ outputs $0$.

- If $\mathcal{A}$ outputs (Accept, $h$), then $\mathcal{A}'$ estimates the quantity $q = \mathbb{P}_{(\mathbf{x},y) \sim \mathcal{D}_{\mathbf{x},y}}[y \neq h(\mathbf{x})]$ up to tolerance $\tau$ and probability of failure $1/6$ and outputs $1$ if the estimate $q$ is at least $1/2 - \tau$ and $0$ otherwise.

We now consider the case that $\mathcal{D}_{\mathbf{x},y} = \mathcal{N}_d \times \mathrm{Unif}(\{\pm 1\})$. According to Definition 1.3, the probability that $\mathcal{A}$ accepts is at least $5/6$. Moreover, we have that regardless of the choice of $h$, $\mathbb{P}_{(\mathbf{x},y) \sim \mathcal{D}_{\mathbf{x},y}}[y \neq h(\mathbf{x})] = 1/2$ and therefore $\mathcal{A}'$ will overall output $1$ with probability at least $2/3$.

In the case that $\mathcal{D}_{\mathbf{x},y} \in \mathfrak{D}_g$, $\mathcal{A}'$ will output $0$ unless the guarantee of the soundness does not hold (which happens with probability at most $1/6$) or the error of estimation of $q$ is more than $\tau$ (which happens with probability at most $1/6$. Hence, overall, $\mathcal{A}'$ will output $1$ with probability at most $1/3$. $\qquad\square$

## C.1 Cryptographic Hardness

We provide cryptographic lower bounds based on the widely-believed hardness of the problem of learning with errors (LWE), which was introduced by [Reg09] and is defined as follows.

**Definition C.2** (Learning with Errors). Let $d, q, m \in \mathbb{N}$ and $\sigma > 0$. The LWE problem with parameters $d, q, m, \sigma$ and advantage $\alpha \in (0, 1)$ is defined as follows. Let $\mathbf{s} \sim \mathrm{Unif}(\mathbb{Z}_q^d)$ and consider the following distributions over $\mathbb{Z}_q^d \times \mathbb{R}$.

- $\mathcal{D}_{\mathsf{null}}$: $\mathbf{x} \sim \mathrm{Unif}(\mathbb{Z}_q^d)$ and $y \sim \mathrm{Unif}(\mathbb{Z}_q)$.

- $\mathcal{D}_{\mathsf{alt}}$: $\mathbf{x} \sim \mathrm{Unif}(\mathbb{Z}_q^d)$, $z \sim \mathcal{N}_1(0, \sigma^2; \mathbb{Z})$, $y = (\mathbf{x} \cdot \mathbf{s} + z) \mod q$

We receive $m$ i.i.d. examples from some distribution $\mathcal{D}_{\mathbf{x},y}$ over $\mathbb{Z}_q^d \times \mathbb{R}$ which is either equal to $\mathcal{D}_{\mathsf{null}}$ or $\mathcal{D}_{\mathsf{alt}}$ and we are asked to output $v \in \{\pm 1\}$ such that $|\mathbb{P}[v = 1 | \mathcal{D}_{\mathbf{x},y} = \mathcal{D}_{\mathsf{null}}] - \mathbb{P}[v = 1 | \mathcal{D}_{\mathbf{x},y} = \mathcal{D}_{\mathsf{alt}}]| \geq \alpha$.

There is strong evidence that the LWE problem cannot be solved in subexponential time, since there are quantum reductions from worst-case lattice problems [Reg09, Pei09].

**Assumption C.3** (Hardness of LWE). *Let $d, q, m \in \mathbb{N}$ and $\sigma > 0$ such that $q \leq d^k$, $\sigma = c\sqrt{d}$ and $m = 2^{O(d^\gamma)}$, where $\gamma \in (0, 1)$, $k \in \mathbb{N}$ are arbitrary constants and $c > 0$ is a sufficiently large constant. Then, any algorithm that solves LWE with parameters $d, q, m, \sigma$ and advantage $2^{-O(d^\gamma)}$ requires time $2^{\Omega(d^\gamma)}$.*

As an immediate corollary of results in [DKR23] (combined with Observation C.1), we obtain the following lower bound under Assumption C.3.

**Theorem C.4** (Cryptographic Hardness in High-Noise Regime, Theorem 3.1 in [DKR23]). *Under Assumption C.3, every algorithm with the guarantees of $\mathcal{A}'$ in Observation C.1 for $\tau = \epsilon \leq 1/\log^{1/2+\beta}(d)$ and $\mathcal{H} = \mathcal{H}_{\mathsf{hs}}$, requires time $\min\{d^{\Omega(1/(\epsilon\sqrt{\log(d)})^\alpha)}, 2^{d^{0.99}}\}$, where $\alpha, \beta \in (0, 2)$ are arbitrary constants.*

*Therefore, the same is true for any testable learning algorithm for $\mathcal{H}_{\mathsf{hs}}$ with respect to the RCN oracle with noise rate $\eta_0 = 1/2$ that has excess error $\epsilon \leq 1/\log^{1/2+\beta}(d)$ and failure probability $\delta \leq 1/6$.*

## C.2 SQ Lower Bounds

We also give lower bounds in the statistical query (SQ) model, which was originally defined by [Kea98]. The SQ framework captures most of the usual algorithmic techniques like moment methods and gradient descent ([FGR+17, FGV17]), and there is a long line of works in computational learning theory giving SQ lower bounds for various learning tasks.

**Definition C.5** (Statistical Query Model). Let $\mathcal{D}_{\mathbf{x},y}$ be a distribution over $\mathbb{R}^d \times \{\pm 1\}$ and $\tau > 0$. A statistical query (SQ) algorithm $\mathcal{A}$ with tolerance $\tau$ has access to $\mathcal{D}_{\mathbf{x},y}$ as follows: The algorithm

(adaptively) makes bounded queries of the form $q : \mathbb{R}^d \times [-1, 1] \to [-1, 1]$. For each query $q$, the algorithm receives a value $v \in \mathbb{R}$ with $|v - \mathbb{E}_{\mathbf{x} \sim \mathcal{D}}[q(\mathbf{x}, y)]| \leq \tau$.

We obtain our lower bound as an immediate corollary of results in [DKR23], combined with Observation C.1, where note that the reduction of the hard distinguishing problem to testable learning also works in the SQ framework, using one statistical query with tolerance $\tau$.

**Theorem C.6** (SQ Lower Bound in High-Noise Regime, Propositions 2.1, 2.8, Corollary B.1 in [DKPZ21]). *Every SQ algorithm with the guarantees of $\mathcal{A}'$ in Observation C.1 for $\tau = \epsilon \geq d^{-c}$ and $\mathcal{H} = \mathcal{H}_{\mathsf{hs}}$, where $c > 0$ is a sufficiently small constant, either requires queries of tolerance $d^{-\Omega(1/\epsilon^2)}$ or makes $2^{d^{\Omega(1)}}$ queries.*

*Therefore, the same is true for any SQ testable learning algorithm for $\mathcal{H}_{\mathsf{hs}}$ with respect to the RCN oracle with noise rate $\eta_0 = 1/2$ that has excess error $\epsilon \geq d^{-c}$ and failure probability $\delta \leq 1/6$.*

# D    Disagreement Tester

In this section we prove the following theorem.

**Theorem D.1.** *For every positive absolute constant $\mu$, there exists a deterministic algorithm $\mathcal{T}_{\text{disagreement}}$ and some absolute constant $C$ that, given*

- *a dataset $S$ of points in $\mathbb{R}^d$ of size $N \geq \left(\frac{Cd}{\epsilon\delta}\right)^C$.*
- *a unit vector $\mathbf{v}$ in $\mathbb{R}^d$,*
- *parameters $\epsilon, \delta$ and $\mu$ in $(0, 1)$.*

*For any absolute constant $\mu$, the algorithm runs in time $poly\left(\frac{dN}{\epsilon\delta}\right)$ and outputs Accept or outputs Reject, subject to the following for all $\epsilon$ and $\delta$ in $(0, 1)$:*

- ***Completeness:*** *if $S$ consists of $N \geq \left(\frac{Cd}{\epsilon\delta}\right)^C$ i.i.d. samples from the standard Gaussian distribution, then with probability at least $1 - O(\delta)$ the set $S$ is such that for all unit vectors $\mathbf{v}$ the algorithm $\mathcal{T}_{\text{disagreement}}$ accepts when given $(S, \mathbf{v}, \epsilon, \delta, \mu)$ as the input.*

- ***Soundness:*** *For any dataset $S$ and unit vector $\mathbf{v}$, if the tester $\mathcal{T}_{\text{disagreement}}$ accepts, then for every unit vector $\mathbf{v}'$ in $\mathbb{R}^d$ the following holds*

$$\mathbb{P}_{\mathbf{x} \sim S}[\mathsf{sign}(\mathbf{x} \cdot \mathbf{v}) \neq \mathsf{sign}(\mathbf{x} \cdot \mathbf{v}')] = (1 \pm \mu)\frac{\angle(\mathbf{v}, \mathbf{v}')}{\pi} \pm O(\epsilon).$$

We argue that the following algorithm (see also Algorithm 1) satisfies the specifications above:

- **Given:** parameter $\epsilon, \delta$ in $(0, 1)$, dataset $S$ of points in $\mathbb{R}^d$ of size $N \geq \left(\frac{Cd}{\epsilon\delta}\right)^C$, a unit vector $\mathbf{v}$ in $\mathbb{R}^d$,

1. $k_1 \leftarrow \frac{2\sqrt{\log 2/\epsilon}}{\epsilon}$
2. $k_2 \leftarrow \frac{C^{0.1}}{\mu^4}$
3. For all $a$ and $b$ in $\{-\infty, -k_1\epsilon, -(k_1 - 1)\epsilon, \cdots, -\epsilon, 0, +\epsilon, \cdots, (k_1 - 1)\epsilon, k_1\epsilon, +\infty\}$
   (a) For all monomials $m$ of degree at most $k_2$ over $\mathbb{R}^d$:
      i. $A_m^{a,b} \leftarrow \mathbb{E}_{\mathbf{x} \sim \mathcal{N}(0, I_d)}[m(\mathbf{x}) \cdot \mathbb{1}_{a \leq \mathbf{x} \cdot \mathbf{v} < b}] \pm \frac{60(2k_2(d+1))^{k_2+2}}{\delta}\left(\frac{\log N}{N}\right)^{1/4}$. (For how to compute this approximation, see Claim 3).
      ii. If $\left|\mathbb{E}_{\mathbf{x} \sim S}[m(\mathbf{x}) \cdot \mathbb{1}_{a \leq \mathbf{x} \cdot \mathbf{v} < b}] - A_m^{a,b}\right| > \frac{200(2k_2(d+1))^{k_2+2}}{\delta}\left(\frac{\log N}{N}\right)^{1/4}$, then output Reject.
4. If did not reject in any previous step, output Accept.

It is immediate that the algorithm indeed runs in time $poly\left(\frac{dN}{\epsilon\delta}\right)$.

### D.1 Completeness

Suppose the set dataset $S$ consists of i.i.d. samples from $\mathcal{N}(0, I_d)$. We observe that the collection $\mathcal{H}$ of sets of the form $\mathbb{1}_{a \leq \mathbf{v} \cdot \mathbf{x} < b}$ has VC dimension at most $(d+1)^2$. This allows us to use Lemma A.1, to conclude that with probability at least $1 - \delta$ for all pairs of $a$ and $b$, for all unit vectors $\mathbf{v}$ and for all monomials $m$ of degree at most $k_2$ we have

$$\left| \underset{\mathbf{x} \sim S}{\mathbb{E}} [m(\mathbf{x}) \cdot \mathbb{1}_{a \leq \mathbf{x} \cdot \mathbf{v} < b}] - \underset{\mathbf{x} \sim \mathcal{N}(0.I_d)}{\mathbb{E}} [m(\mathbf{x}) \cdot \mathbb{1}_{a \leq \mathbf{x} \cdot \mathbf{v} < b}] \right| \leq \frac{60(2k_2)^{k_2+2}(d+1)^{k_2+2}}{\delta} \left( \frac{\log N}{N} \right)^{1/4}$$

and Claim 3 implies that

$$\left| A_m^{a,b} - \underset{\mathbf{x} \sim \mathcal{N}(0.I_d)}{\mathbb{E}} [m(\mathbf{x}) \cdot \mathbb{1}_{a \leq \mathbf{x} \cdot \mathbf{v} < b}] \right| \leq \frac{60(2k_2)^{k_2+2}(d+1)^{k_2+2}}{\delta} \left( \frac{\log N}{N} \right)^{1/4}$$

The two inequalities above together imply the completeness condition.

### D.2 Soundness

In order to deduce the soundness condition, we will need the following notions:

**Definition D.2.** Let $\mathcal{C}$ be a collection of disjoint subsets of $\mathbb{R}^d$. We say that $\mathcal{C}$ is a *partition* of $\mathbb{R}^d$ if $\mathbb{R}^d$ equals to the union $\bigcup_{A \in \mathcal{C}} A$.

**Definition D.3.** We say that a function $f : \mathbb{R}^d \to \{0, 1\}$ is $\epsilon$-sandwiched in $L_1$ norm between a pair of functions $f_{\text{up}} : \mathbb{R}^d \to \mathbb{R}$ and $f_{\text{down}} : \mathbb{R}^d \to \mathbb{R}$ under $\mathcal{N}(0, I_d)$ if:

- For all $\mathbf{x}$ in $\mathbb{R}^d$ we have $f_{\text{down}}(\mathbf{x}) \leq f(\mathbf{x}) \leq f_{\text{up}}(\mathbf{x})$

- $\mathbb{E}_{\mathbf{x} \sim \mathcal{N}(0, I_d)} [f_{\text{up}}(\mathbf{x}) - f_{\text{down}}(\mathbf{x})] \leq \epsilon$.

**Definition D.4.** We say that a function $f : \mathbb{R}^d \to \{0, 1\}$ has $(\epsilon, B)$-sandwiching degree of at most $k$ in $L_1$ norm under $\mathcal{N}(0, I_d)$ with respect to a partition $\mathcal{C}$ of $\mathbb{R}^d$ if the function $f$ is $\epsilon$-sandwiched in $L_1$ norm under $\mathcal{N}(0, I_d)$ between $\sum_{A \in \mathcal{C}} \left( p_{\text{down}}^A \mathbb{1}_A \right)$ and $\sum_{A \in \mathcal{C}} \left( p_{\text{up}}^A \mathbb{1}_A \right)$, where $p_{\text{up}}^A$ and $p_{\text{down}}^A$ are degree$-k$ polynomials over $\mathbb{R}^d$ whose coefficients are bounded by $B$ in absolute value.

Subsection D.3 is dedicated to proving the following bound on the sandwiching degree of a specific family of functions with respect to a specific partition of $\mathbb{R}^d$.

**Proposition D.5.** *For all $\epsilon$ and $k_2$, let $k_1 = \frac{2\sqrt{\log 2/\epsilon}}{\epsilon}$, and let $\mathbf{v}$ be a unit vector in $\mathbb{R}^d$. Then, there exists a partition $\mathcal{C}$ of $\mathbb{R}^d$ consisting of sets of the form $\{ \mathbf{x} \in \mathbb{R}^d : a \leq \mathbf{v} \cdot \mathbf{x} \leq b \}$ for a certain collection of pairs $a, b$ in $\{ -\infty, -k_1\epsilon, -(k_1-1)\epsilon, \cdots, -\epsilon, 0, +\epsilon, \cdots, (k_1-1)\epsilon, k_1\epsilon, +\infty \}$. Then, for every unit vector $\mathbf{v}'$, the function $f(\mathbf{x}) = \mathbb{1}_{\text{sign}(\mathbf{v} \cdot \mathbf{x}) \neq \text{sign}(\mathbf{v}' \cdot \mathbf{x})}$ has $\left( O \left( \frac{\sphericalangle(\mathbf{v}, \mathbf{v}')}{k_2^{1/4}} \right) + 10\epsilon, O \left( d^{10k_2} \right) \right)$-sandwiching degree of at most $k_2$ in $L_1$ norm under $\mathcal{N}(0, I_d)$ with respect to the partition $\mathcal{C}$ of $\mathbb{R}^d$.*

A bound on the sandwiching degree of a class of functions leads to a guarantee for the tester $\mathcal{T}_{\text{disagreement}}$:

**Proposition D.6.** *Let $\mathcal{C}$ be a partition of $\mathbb{R}^d$ and suppose that a set $S$ of points in $\mathbb{R}^d$ satisfies the following condition for all $A$ in $\mathcal{C}$ and degree-$k_2$ monomials $m$ over $\mathbb{R}^d$:*

$$\left| \underset{\mathbf{x} \sim S}{\mathbb{E}} [m(\mathbf{x}) \cdot \mathbb{1}_{\mathbf{x} \in A}] - \underset{\mathbf{x} \sim \mathcal{N}(0.I_d)}{\mathbb{E}} [m(\mathbf{x}) \cdot \mathbb{1}_{\mathbf{x} \in A}] \right| \leq \frac{\epsilon}{(d+1)^k |\mathcal{C}| B} \tag{D.1}$$

*Then, every $\{0, 1\}$-valued function $f$ that has has $(\nu, B)$-sandwiching degree of at most $k$ in $L_1$ norm under $\mathcal{N}(0, I_d)$ with respect to the partition $\mathcal{C}$ we have*

$$\left| \underset{\mathbf{x} \sim S}{\mathbb{P}} [f(\mathbf{x}) = 1] - \underset{\mathbf{x} \sim \mathcal{N}(0, I_d)}{\mathbb{P}} [f(\mathbf{x}) = 1] \right| \leq \nu + O(\epsilon)$$

*Proof.* Since $f$ has $(\nu, B)$-sandwiching degree of at most $k$ in $L_1$ norm under $\mathcal{N}(0, I_d)$ with respect to the partition $\mathcal{C}$, we have a collection of polynomials $\{p_{\text{down}}^A, p_{\text{up}}^A\}$ for all $A$ in $\mathcal{C}$ that have coefficients bounded by $B$, satisfy for all $\mathbf{x}$ the condition

$$f(\mathbf{x}) \in \left[ \sum_{A \in \mathcal{C}} \left( p_{\text{down}}^A(\mathbf{x}) \mathbb{1}_{\mathbf{x} \in A} \right), \sum_{A \in \mathcal{C}} \left( p_{\text{up}}^A(\mathbf{x}) \mathbb{1}_{\mathbf{x} \in A} \right) \right], \tag{D.2}$$

as well as

$$\mathbb{E}_{\mathbf{x} \sim \mathcal{N}(0, I_d)} \left[ \sum_{A \in \mathcal{C}} \left( p_{\text{up}}^A(\mathbf{x}) \mathbb{1}_{\mathbf{x} \in A} \right) - \sum_{A \in \mathcal{C}} \left( p_{\text{down}}^A(\mathbf{x}) \mathbb{1}_{\mathbf{x} \in A} \right) \right] \leq \nu. \tag{D.3}$$

From the bound $B$ on all coefficients of $p_{\text{up}}^A$ and $p_{\text{down}}^A$ and Equation D.1 we see that:

$$\left| \sum_{A \in \mathcal{C}} \mathbb{E}_{\mathbf{x} \sim \mathcal{N}(0, I_d)} \left[ \left( p_{\text{down}}^A(\mathbf{x}) \mathbb{1}_{\mathbf{x} \in A} \right) \right] - \sum_{A \in \mathcal{C}} \mathbb{E}_{\mathbf{x} \sim D} \left[ \left( p_{\text{down}}^A(\mathbf{x}) \mathbb{1}_{\mathbf{x} \in A} \right) \right] \right| \leq \frac{\epsilon (d+1)^k |\mathcal{C}| B}{(d+1)^k |\mathcal{C}| B} = \epsilon, \tag{D.4}$$

$$\left| \sum_{A \in \mathcal{C}} \mathbb{E}_{\mathbf{x} \sim \mathcal{N}(0, I_d)} \left[ \left( p_{\text{up}}^A(\mathbf{x}) \mathbb{1}_{\mathbf{x} \in A} \right) \right] - \sum_{A \in \mathcal{C}} \mathbb{E}_{\mathbf{x} \sim D} \left[ \left( p_{\text{up}}^A(\mathbf{x}) \mathbb{1}_{\mathbf{x} \in A} \right) \right] \right| \leq \frac{\epsilon (d+1)^k |\mathcal{C}| B}{(d+1)^k |\mathcal{C}| B} = \epsilon. \tag{D.5}$$

Equation D.2 implies that

$$\sum_{A \in \mathcal{C}} \mathbb{E}_{\mathbf{x} \sim \mathcal{N}(0, I_d)} \left[ \left( p_{\text{down}}^A(\mathbf{x}) \mathbb{1}_{\mathbf{x} \in A} \right) \right] \leq \mathbb{E}_{\mathbf{x} \sim \mathcal{N}(0, I_d)} [f(\mathbf{x})] \leq \sum_{A \in \mathcal{C}} \mathbb{E}_{\mathbf{x} \sim \mathcal{N}(0, I_d)} \left[ \left( p_{\text{up}}^A(\mathbf{x}) \mathbb{1}_{\mathbf{x} \in A} \right) \right], \tag{D.6}$$

and Equation D.2 together with Equations D.4 and D.5 implies that:

$$\sum_{A \in \mathcal{C}} \mathbb{E}_{\mathbf{x} \sim \mathcal{N}(0, I_d)} \left[ \left( p_{\text{down}}^A(\mathbf{x}) \mathbb{1}_{\mathbf{x} \in A} \right) \right] - \epsilon \leq \sum_{A \in \mathcal{C}} \mathbb{E}_{\mathbf{x} \sim D} \left[ \left( p_{\text{down}}^A(\mathbf{x}) \mathbb{1}_{\mathbf{x} \in A} \right) \right] \leq \mathbb{E}_{\mathbf{x} \sim D} [f(\mathbf{x})] \leq$$

$$\leq \sum_{A \in \mathcal{C}} \mathbb{E}_{\mathbf{x} \sim D} \left[ \left( p_{\text{up}}^A(\mathbf{x}) \mathbb{1}_{\mathbf{x} \in A} \right) \right] \leq \sum_{A \in \mathcal{C}} \mathbb{E}_{\mathbf{x} \sim \mathcal{N}(0, I_d)} \left[ \left( p_{\text{up}}^A(\mathbf{x}) \mathbb{1}_{\mathbf{x} \in A} \right) \right] + \epsilon. \tag{D.7}$$

Together Equations D.7 and D.6 constraint the values of both $\mathbb{E}_{\mathbf{x} \sim D}[f(\mathbf{x})]$ and $\mathbb{E}_{\mathbf{x} \sim \mathcal{N}(0, I_d)}[f(\mathbf{x})]$ to the same interval that via Equation D.3 has a width of at most $\nu + 2\epsilon$. This allows us to conclude

$$\left| \mathbb{P}_{\mathbf{x} \sim D}[f(\mathbf{x}) = 1] - \mathbb{P}_{\mathbf{x} \sim \mathcal{N}(0, I_d)}[f(\mathbf{x}) = 1] \right| = \left| \mathbb{E}_{\mathbf{x} \sim D}[f(\mathbf{x})] - \mathbb{E}_{\mathbf{x} \sim \mathcal{N}(0, I_d)}[f(\mathbf{x})] \right| \leq \nu + 2\epsilon,$$

completing the proof. $\square$

Claim 3 implies that all pairs of $a$ and $b$ in the set

$\{-\infty, -k_1\epsilon, -(k_1 - 1)\epsilon, \cdots, -\epsilon, 0, +\epsilon, \cdots, (k_1 - 1)\epsilon, k_1\epsilon, +\infty\}$ and for all monomials $m$ of degree at most $k_2$ we have

$$\left| A_m^{a,b} - \mathbb{E}_{\mathbf{x} \sim \mathcal{N}(0, I_d)}[m(\mathbf{x}) \cdot \mathbb{1}_{a \leq \mathbf{x} \cdot \mathbf{v} < b}] \right| \leq \frac{60(2k_2)^{k_2+2}(d+1)^{k_2+2}}{\delta} \left( \frac{\log N}{N} \right)^{1/4}.$$

If the algorithm accepts, then we have for all pairs of $a$ and $b$ in

$\{-\infty, -k_1\epsilon, -(k_1 - 1)\epsilon, \cdots, -\epsilon, 0, +\epsilon, \cdots, (k_1 - 1)\epsilon, k_1\epsilon, +\infty\}$ and for all monomials $m$ of degree at most $k_2$ that

$$\left| \mathbb{E}_{\mathbf{x} \sim S}[m(\mathbf{x}) \cdot \mathbb{1}_{a \leq \mathbf{x} \cdot \mathbf{v} < b}] - A_m^{a,b} \right| \leq \frac{200(2k_2)^{k_2+2}(d+1)^{k_2+2}}{\delta} \left( \frac{\log N}{N} \right)^{1/4}.$$

The two inequalities above imply that

$$\left| \mathbb{E}_{\mathbf{x} \sim S}[m(\mathbf{x}) \cdot \mathbb{1}_{a \leq \mathbf{x} \cdot \mathbf{v} < b}] - \mathbb{E}_{\mathbf{x} \sim \mathcal{N}(0, I_d)}[m(\mathbf{x}) \cdot \mathbb{1}_{a \leq \mathbf{x} \cdot \mathbf{v} < b}] \right| \leq \frac{260(2k_2)^{k_2+2}(d+1)^{k_2+2}}{\delta} \left( \frac{\log N}{N} \right)^{1/4}$$

Taking the equation above, together with Proposition D.6 and Proposition D.5 we conclude that

$$\left| \mathop{\mathbb{P}}_{\mathbf{x}\sim S}[\mathsf{sign}(\mathbf{v}\cdot\mathbf{x}) \neq \mathsf{sign}(\mathbf{v}'\cdot\mathbf{x})] - \mathop{\mathbb{P}}_{\mathbf{x}\sim\mathcal{N}(0,I_d)}[\mathsf{sign}(\mathbf{v}\cdot\mathbf{x}) \neq \mathsf{sign}(\mathbf{v}'\cdot\mathbf{x})] \right| \leq O\left(\frac{\angle(\mathbf{v},\mathbf{v}')}{k_2^{1/4}}\right) + $$
$$ + O\left(\frac{(2k_2)^{k_2+2}(d+1)^{k_2+2}}{\delta}\left(\frac{\log N}{N}\right)^{1/4}\right)$$

Substituting $k_2 \leftarrow \frac{C^{0.1}}{\mu^4}$, $N \geq \left(\frac{Cd}{\epsilon\delta}\right)^C$, taking $C$ to be a sufficiently large absolute constant and recalling that $\mathbb{P}_{\mathbf{x}\sim\mathcal{N}(0,I_d)}[\mathsf{sign}(\mathbf{v}\cdot\mathbf{x}) \neq \mathsf{sign}(\mathbf{v}'\cdot\mathbf{x})]$ equals to $\angle(\mathbf{v},\mathbf{v}')/\pi$ we conclude that

$$\mathop{\mathbb{P}}_{\mathbf{x}\sim S}[\mathsf{sign}(\mathbf{x}\cdot\mathbf{v}) \neq \mathsf{sign}(\mathbf{x}\cdot\mathbf{v}')] = (1 \pm \mu)\frac{\angle(\mathbf{v},\mathbf{v}')}{\pi} \pm O(\epsilon).$$

### D.3 Bounding sandwiching degree of the disagreement region

To prove Proposition D.5, we will need the following result by [DGJ$^+$10], [GOWZ10].

**Fact D.7.** *For every positive integer $k$ and a real value $t$, the function $f(z) = \mathbb{1}_{z\leq t}$ has $(O(\frac{\log^3 k}{\sqrt{k}}), O(2^{10k}))$-sandwiching degree in $L_1$ norm of at most $k$ under $\mathcal{N}(0,1)$.*

The following corollary slightly strengthens the fact above:

**Corollary D.8.** *Let $t \in \mathbb{R}$. For every positive integer $k \geq 2$, the function $f : \mathbb{R} \to \{0,1\}$ with $f(z) = \mathbb{1}_{z\leq t}$ is $(O(\min(\frac{\log^3 k}{\sqrt{k}}, \frac{1}{t^2})), 2^{10k})$-sandwiched in $L_1$ norm under $\mathcal{N}(0,1)$ between a pair of polynomials $R_{down}^t$ and $R_{up}^t$ of degree $k$.*

*Proof.* Indeed, if $\frac{\log^3 k}{\sqrt{k}} \leq \frac{1}{t^2}$ then the corollary follows from Fact D.7. So all we need to do is to consider the other case. We see that either $t > 1$ or $t < -1$ (since $k \geq 2$). if $t > 1$ we take $p_{\mathsf{down}}(\mathbf{x}) = 0$ and $p_{\mathsf{up}}(\mathbf{x}) = \left(\frac{x}{t}\right)^2$. If $t < -1$, we take take $p_{\mathsf{up}}(\mathbf{x}) = 1$ and $p_{\mathsf{down}}(\mathbf{x}) = 1 - \left(\frac{x}{t}\right)^2$. In either case, we see that the polynomials $p_{\mathsf{down}}$ and $p_{\mathsf{up}}$ form a pair of $\left(O\left(\min\left(\frac{\log^3 k}{\sqrt{k}}, \frac{1}{t^2}\right)\right), 1\right)$-sandwiching polynomials of degree 2. $\qquad\square$

Let $\mathbf{v}_\perp$ be the unit vector equal up to scaling to the component of $\mathbf{v}'$ perpendicular to $\mathbf{v}$. Then, we have

$$\psi_{\mathsf{down}}(\mathbf{x}) \leq \mathbb{1}_{\mathbf{v}\cdot\mathbf{x}\geq 0 \wedge \mathbf{v}'\cdot\mathbf{x}<0} \leq \psi_{\mathsf{up}}(\mathbf{x}) \tag{D.8}$$
$$\psi_{\mathsf{down}}(\mathbf{x}) \leq \mathbb{1}_{\mathbf{v}\cdot\mathbf{x}>0 \wedge \mathbf{v}'\cdot\mathbf{x}\leq 0} \leq \psi_{\mathsf{up}}(\mathbf{x}) \tag{D.9}$$

where

$$\psi_{\mathsf{up}}(\mathbf{x}) = \begin{cases} 1 & \text{if } \mathbf{v}\cdot\mathbf{x} \geq k_1\epsilon \text{ or } \mathbf{v}\cdot\mathbf{x} = 0 \\ \mathbb{1}_{\mathbf{v}_\perp\cdot\mathbf{x}\tan\theta\leq -j\epsilon} & \text{if } \mathbf{v}\cdot\mathbf{x} \neq 0 \text{ and } \mathbf{v}\cdot\mathbf{x} \in [j\epsilon, (j+1)\epsilon) \text{ for } 0 \leq j \leq k_1 - 1 \\ 0 & \text{if } \mathbf{v}\cdot\mathbf{x} < 0 \end{cases}$$

$$\psi_{\mathsf{down}}(\mathbf{x}) = \begin{cases} 0 & \text{if } \mathbf{v}\cdot\mathbf{x} \geq k_1\epsilon \text{ or } \mathbf{v}\cdot\mathbf{x} = 0 \\ \mathbb{1}_{\mathbf{v}_\perp\cdot\mathbf{x}\tan\theta< -(j+1)\epsilon} & \text{if } \mathbf{v}\cdot\mathbf{x} \neq 0 \text{ and } \mathbf{v}\cdot\mathbf{x} \in [j\epsilon, (j+1)\epsilon) \text{ for } 0 \leq j \leq k_1 - 1 \\ 0 & \text{if } \mathbf{v}\cdot\mathbf{x} < 0 \end{cases}$$

Recall that for every $t \in \mathbb{R}$, Corollary D.8 gives us one-dimensional degree-$k_2$ sandwiching polynomials $R_{\text{down}}^t(z)$ and $R_{\text{up}}^t(z)$ for $\mathbb{1}_{z \leq t}$. Using this notation, we have for all $\mathbf{x}$ in $\mathbb{R}^d$

$$
\overbrace{\sum_{j=0}^{k_1-1} \mathbb{1}_{\mathbf{v} \cdot \mathbf{x} \cdot [j\epsilon, (j+1)\epsilon)} R_{\text{down}}^{-(j+1)\epsilon/\tan\theta}(\mathbf{v}_\perp \cdot \mathbf{x})}^{\text{Denote this } \varphi_{\text{down}}(\mathbf{x})} \leq \psi_{\text{down}}(\mathbf{x}) \leq \mathbb{1}_{\mathbf{v} \cdot \mathbf{x} \geq 0 \wedge \mathbf{v}' \cdot \mathbf{x} < 0} \leq
$$

$$
\leq \psi_{\text{up}}(\mathbf{x}) \leq \underbrace{\mathbb{1}_{\mathbf{v} \cdot \mathbf{x} \geq k_1\epsilon} + \sum_{j=0}^{k_1-1} \mathbb{1}_{\mathbf{v} \cdot \mathbf{x} \cdot [j\epsilon, (j+1)\epsilon)} R_{\text{up}}^{-j\epsilon/\tan\theta}(\mathbf{v}_\perp \cdot \mathbf{x})}_{\text{Denote this } \varphi_{\text{up}}(\mathbf{x})} \quad \text{(D.10)}
$$

In order to conclude Proposition D.5. We show the following two claims:

**Claim 1.** *We have*

$$
\mathbb{E}_{\mathbf{x} \sim \mathcal{N}(0, I_d)} [\varphi_{up}(\mathbf{x}) - \varphi_{down}(\mathbf{x})] \leq O\left(\frac{\log^{1.5} k_2}{k_2^{1/4}} \cdot \measuredangle(\mathbf{v}, \mathbf{v}')\right) + 10\epsilon
$$

**Claim 2.** *For all integers $j$ in $[0, k_1 - 1]]$, every coefficient of $R_{down}^{-(j+1)\epsilon/\tan\theta}(\mathbf{v}_\perp \cdot \mathbf{x})$ and $R_{up}^{-j\epsilon/\tan\theta}(\mathbf{v}_\perp \cdot \mathbf{x})$ is at most $O\left(d^{10k_2}\right)$ in absolute value.*

Proposition D.5 follows from the two claims above for as follows. We first observe that Equations D.9 and D.10 imply that

$$
\varphi_{\text{down}}(-\mathbf{x}) \leq \mathbb{1}_{\mathbf{v} \cdot \mathbf{x} < 0 \wedge \mathbf{v}' \cdot \mathbf{x} \geq 0} \leq \varphi_{\text{up}}(-\mathbf{x}).
$$

Recalling our convention that $\text{sign}(0) = 1$, we see that

$$
\mathbb{1}_{\text{sign}(\mathbf{v} \cdot \mathbf{x}) \neq \text{sign}(\mathbf{v}' \cdot \mathbf{x})} = \mathbb{1}_{\mathbf{v} \cdot \mathbf{x} \geq 0 \wedge \mathbf{v}' \cdot \mathbf{x} < 0} + \mathbb{1}_{\mathbf{v} \cdot \mathbf{x} < 0 \wedge \mathbf{v}' \cdot \mathbf{x} \geq 0}
$$

this, together with D.10 allows us to bound

$$
\varphi_{\text{down}}(\mathbf{x}) + \varphi_{\text{down}}(-\mathbf{x}) \leq \mathbb{1}_{\text{sign}(\mathbf{v} \cdot \mathbf{x}) \neq \text{sign}(\mathbf{v}' \cdot \mathbf{x})} \leq \varphi_{\text{up}}(\mathbf{x}) + \varphi_{\text{up}}(-\mathbf{x}), \quad \text{(D.11)}
$$

Claim D.10 allows us to conclude that

$$
\mathbb{E}_{\mathbf{x} \sim \mathcal{N}(0, I_d)} [\varphi_{\text{up}}(\mathbf{x}) + \varphi_{\text{up}}(-\mathbf{x}) - \varphi_{\text{down}}(\mathbf{x}) - \varphi_{\text{down}}(-\mathbf{x})] \leq O\left(\frac{\log^{1.5} k_2}{k_2^{1/4}} \cdot \measuredangle(\mathbf{v}, \mathbf{v}')\right) + 20\epsilon.
$$

$$\text{(D.12)}$$

Equations D.11 and D.12, together with comparing the definition of $\varphi_{\text{up}}$ and $\varphi_{\text{down}}$ with Definition D.4 and recalling Claim 2, allow us to conclude that there exists a partition $\mathcal{C}$ of $\mathbb{R}^d$ consisting of sets of the form $\{\mathbf{x} \in \mathbb{R}^d : a \leq \mathbf{v} \cdot \mathbf{x} \leq b\}$ for a certain collection of pairs $a, b$ in

$\{-\infty, -k_1\epsilon, -(k_1 - 1)\epsilon, \cdots, -\epsilon, 0, +\epsilon, \cdots, (k_1 - 1)\epsilon, k_1\epsilon, +\infty\}$, such that for every unit vector $\mathbf{v}'$, the function $f(\mathbf{x}) = \mathbb{1}_{\text{sign}(\mathbf{v} \cdot \mathbf{x}) \neq \text{sign}(\mathbf{v}' \cdot \mathbf{x})}$ has $\left(O\left(\frac{\measuredangle(\mathbf{v}, \mathbf{v}')}{k_2^{1/4}}\right) + 10\epsilon, O\left(d^{10k_2}\right)\right)$-sandwiching

degree of at most $k_2$ in $L_1$ norm under $\mathcal{N}(0, I_d)$ with respect to the partition $\mathcal{C}$ of $\mathbb{R}^d$. This implies Proposition D.5.

We now proceed to proving Claim 1

*Proof of Claim 1.* We have the following.

$$
\mathbb{E}_{\mathbf{x} \sim \mathcal{N}(0, I_d)} [\psi_{\text{up}}(\mathbf{x}) - \psi_{\text{down}}(\mathbf{x})] \leq
$$

$$
\mathbb{P}_{\mathbf{x} \sim \mathcal{N}(0, I_d)} [\mathbf{v} \cdot \mathbf{x} > k_1\epsilon] + \sum_{j=0}^{k_1-1} \mathbb{P}_{\mathbf{x} \sim \mathcal{N}(0, I_d)} [\{\mathbf{v} \cdot \mathbf{x} \in [j\epsilon, (j+1)\epsilon)\} \wedge \{\mathbf{v}_\perp \cdot \mathbf{x} \tan\theta \in [-(j+1)\epsilon, -j\epsilon)\}] \leq
$$

$$
e^{-(k_1\epsilon)^2} + \epsilon \underbrace{\sum_{j=0}^{\infty} \mathbb{P}_{\mathbf{x} \sim \mathcal{N}(0, I_d)} [\mathbf{v}_\perp \cdot \mathbf{x} \tan\theta \in [-(j+1)\epsilon, -j\epsilon)]}_{\leq 1} \leq 2\epsilon \quad \text{(D.13)}
$$

Let $\theta$ denote the angle $\measuredangle(\mathbf{v}, \mathbf{v}')$. Given the inequality above, in order to finish the proof of Claim 1, it remains to upper-bound $\mathbb{E}_{\mathbf{x} \sim \mathcal{N}(0, I_d)} [\varphi_{\mathrm{up}}(\mathbf{x}) - \psi_{\mathrm{up}}(\mathbf{x})]$ and $\mathbb{E}_{\mathbf{x} \sim \mathcal{N}(0, I_d)} [\varphi_{\mathrm{up}}(\mathbf{x}) - \psi_{\mathrm{up}}(\mathbf{x})]$ by $O(\theta) + 4\epsilon$.

From Corollary D.8 we know that for any $t$ we have:

$$\mathop{\mathbb{E}}_{z \sim \mathcal{N}(0,1)} \left[ R_{\mathrm{up}}^t(z) - R_{\mathrm{down}}^t(z) \right] \leq O\left( \min\left( \frac{\log^3 k}{\sqrt{k}}, \frac{1}{t^2} \right) \right), \tag{D.14}$$

and for every $z$ in $\mathbb{R}$

$$R_{\mathrm{down}}^t(z) \leq \mathbb{1}_{z \leq t} \leq R_{\mathrm{up}}^t(z). \tag{D.15}$$

Since $\mathbf{v}$ and $\mathbf{v}_\perp$ are orthogonal, the random variables $\mathbf{v}_\perp \mathbf{x}$ and $\mathbf{v} \cdot \mathbf{x}$ are independent standard Gaussians. Using this, together with Equations D.14 and D.15 we obtain the following.

$$\mathop{\mathbb{E}}_{\mathbf{x} \sim \mathcal{N}(0, I_d)} [\varphi_{\mathrm{up}}(\mathbf{x}) - \psi_{\mathrm{up}}(\mathbf{x})]$$

$$= \sum_{j=0}^{k_1-1} \mathop{\mathbb{P}}_{\mathbf{x} \sim \mathcal{N}(0, I_d)} [\mathbf{v} \cdot \mathbf{x} \in [j\epsilon, (j+1)\epsilon)] \mathop{\mathbb{E}}_{\mathbf{x} \sim \mathcal{N}(0, I_d)} \left[ R_{\mathrm{up}}^{-j\epsilon/\tan\theta}(\mathbf{v}_\perp \cdot \mathbf{x}) - \mathbb{1}_{\mathbf{v}_\perp \cdot \mathbf{x} \leq -j\epsilon/\tan\theta} \right]$$

$$\leq \sum_{j=0}^{k_1-1} \mathop{\mathbb{P}}_{z_1 \sim \mathcal{N}(0,1)} [z_1 \in [j\epsilon, (j+1)\epsilon)] \mathop{\mathbb{E}}_{z_2 \sim \mathcal{N}(0,1)} \left[ R_{\mathrm{up}}^{-j\epsilon/\tan\theta}(z_2) - R_{\mathrm{down}}^{-j\epsilon/\tan\theta}(z_2) \right]$$

$$\leq \sum_{j=0}^{k_1-1} \mathop{\mathbb{P}}_{z_1 \sim \mathcal{N}(0,1)} [z_1 \in [j\epsilon, (j+1)\epsilon)] O\left( \min\left( \frac{\log^3 k_2}{\sqrt{k_2}}, \left( \frac{\tan\theta}{j\epsilon} \right)^2 \right) \right)$$

First, consider the case $\theta \geq \pi/4$. The above inequality implies

$$\mathop{\mathbb{E}}_{\mathbf{x} \sim \mathcal{N}(0, I_d)} [\varphi_{\mathrm{up}}(\mathbf{x}) - \psi_{\mathrm{up}}(\mathbf{x})] \leq O\left( \frac{\log^3 k}{\sqrt{k}} \right) \underbrace{\sum_{j=-k_1}^{k_1-1} \mathop{\mathbb{P}}_{z_1 \sim \mathcal{N}(0,1)} [z_1 \in [j\epsilon, (j+1)\epsilon)]}_{= \mathbb{P}_{z_1 \sim \mathcal{N}(0,1)} [-k_1\epsilon \leq z_1 < (k_1-1)\epsilon]}$$

$$= O\left( \frac{\log^3 k_2}{\sqrt{k_2}} \right) = O\left( \frac{\theta \log^{1.5} k_2}{k_2^{1/4}} \right)$$

On the other hand, if $\theta \leq \pi/4$ we have $\tan\theta \leq 2\theta$ and therefore, recalling that for any $j$ it is the case that $\mathbb{P}_{z_1 \sim \mathcal{N}(0,1)} [z_1 \in [j\epsilon, (j+1)\epsilon)] \leq \epsilon$, we have

$$\mathop{\mathbb{E}}_{\mathbf{x} \sim \mathcal{N}(0, I_d)} \left[ \varphi_{\mathrm{up}}(\mathbf{x}) - \psi_{\mathrm{up}}(\mathbf{x}) \right] \leq \sum_{j=0}^{k_1-1} O\left( \min\left( \frac{\log^3 k_2}{\sqrt{k_2}}, \left( \frac{\tan\theta}{j\epsilon} \right)^2 \right) \epsilon \right)$$

$$= \int_0^{k_1\epsilon} O\left( \min\left( \frac{\log^3 k_2}{\sqrt{k_2}}, \left( \frac{\tan\theta}{\lfloor z/\epsilon \rfloor \epsilon} \right)^2 \right) \right) dz$$

$$\leq \int_0^{+\infty} O\left( \min\left( \frac{\log^3 k_2}{\sqrt{k_2}}, \left( \frac{\tan\theta}{z - \epsilon} \right)^2 \right) \right) dz$$

$$= O\left( \frac{\log^3 k_2}{\sqrt{k_2}} \epsilon \right) + \int_0^{+\infty} O\left( \min\left( \frac{\log^3 k_2}{\sqrt{k_2}}, \left( \frac{\tan\theta}{z} \right)^2 \right) \right) dz,$$

which together with a change of variables with a new variable $z' = z/\tan\theta$ allows us to proceed as follows:

$$\mathop{\mathbb{E}}_{\mathbf{x} \sim \mathcal{N}(0, I_d)} \left[ \varphi_{\mathrm{up}}(\mathbf{x}) - \psi_{\mathrm{up}}(\mathbf{x}) \right] = O\left( \frac{\log^3 k_2}{\sqrt{k_2}} \epsilon \right) + \tan\theta \int_0^{+\infty} O\left( \min\left( \frac{\log^3 k_2}{\sqrt{k_2}}, \left( \frac{1}{z'} \right)^2 \right) \right) dz' =$$

$$= O\left( \frac{\log^3 k_2}{\sqrt{k_2}} \epsilon + \tan\theta \left( \frac{\log^3 k_2}{\sqrt{k_2}} \left( \frac{\sqrt{k_2}}{\log^3 k_2} \right)^{0.5} + \left( \frac{\log^3 k_2}{\sqrt{k_2}} \right)^{0.5} \right) \right) = O\left( \frac{\theta \log^{1.5} k_2}{k_2^{1/4}} \right) \tag{D.16}$$

Overall, in either case we have $\mathbb{E}_{\mathbf{x}\sim\mathcal{N}(0,I_d)}\left[\varphi_{\text{up}}(\mathbf{x}) - \psi_{\text{up}}(\mathbf{x})\right] = O\left(\frac{\theta\log^{1.5}k_2}{k_2^{1/4}}\right)$. We now go through

a fully analogous argument to show that also $\mathbb{E}_{\mathbf{x}\sim\mathcal{N}(0,I_d)}\left[\psi_{\text{down}}(\mathbf{x}) - \varphi_{\text{down}}(\mathbf{x})\right] = O\left(\frac{\theta\log^{1.5}k_2}{k_2^{1/4}}\right)$.

Again, from the independence of $\mathbf{v}_\perp\mathbf{x}$ and $\mathbf{v}\cdot\mathbf{x}$, together with Equations D.14 and D.15 we have:

$$
\mathbb{E}_{\mathbf{x}\sim\mathcal{N}(0,I_d)}\left[\psi_{\text{down}}(\mathbf{x}) - \varphi_{\text{down}}(\mathbf{x})\right]
$$

$$
= \sum_{j=0}^{k_1-1} \mathbb{P}_{\mathbf{x}\sim\mathcal{N}(0,I_d)}\left[\mathbf{v}\cdot\mathbf{x}\in[j\epsilon,(j+1)\epsilon)\right]\ \mathbb{E}_{\mathbf{x}\sim\mathcal{N}(0,I_d)}\left[\mathbb{1}_{\mathbf{v}_\perp\cdot\mathbf{x}\leq-(j+1)\epsilon/\tan\theta} - R_{\text{down}}^{-(j+1)\epsilon/\tan\theta}(\mathbf{v}_\perp\cdot\mathbf{x})\right]
$$

$$
\leq \sum_{j=0}^{k_1-1} \mathbb{P}_{z_1\sim\mathcal{N}(0,1)}\left[z_1\in[j\epsilon,(j+1)\epsilon)\right]\ \mathbb{E}_{z_2\sim\mathcal{N}(0,1)}\left[R_{\text{up}}^{-(j+1)\epsilon/\tan\theta}(z_2) - R_{\text{down}}^{-(j+1)\epsilon/\tan\theta}(z_2)\right]
$$

$$
\leq \sum_{j=-k_1}^{k_1-1} \mathbb{P}_{z_1\sim\mathcal{N}(0,1)}\left[z_1\in\left[j\epsilon,(j+1)\epsilon\right)\right]O\left(\min\left(\frac{\log^3 k_2}{\sqrt{k_2}},\left(\frac{\tan\theta}{(j+1)\epsilon}\right)^2\right)\right)
$$

Again, we first consider the case $\theta\geq\pi/4$. The above inequality implies

$$
\mathbb{E}_{\mathbf{x}\sim\mathcal{N}(0,I_d)}\left[\psi_{\text{down}}(\mathbf{x}) - \varphi_{\text{down}}(\mathbf{x})\right] \leq
$$

$$
O\left(\frac{\log^3 k}{\sqrt{k}}\right)\underbrace{\sum_{j=0}^{k_1-1}\mathbb{P}_{z_1\sim\mathcal{N}(0,1)}\left[z_1\in\left[j\epsilon,(j+1)\epsilon\right)\right]}_{=\mathbb{P}_{z_1\sim\mathcal{N}(0,1)}\left[0\leq z_1<(k_1-1)\epsilon\right]} = O\left(\frac{\log^3 k_2}{\sqrt{k_2}}\right) = O\left(\frac{\theta\log^{1.5}k_2}{k_2^{1/4}}\right)
$$

On the other hand, if $\theta\leq\pi/4$ we have $\tan\theta\leq2\theta$ and therefore, recalling that for any $j$ it is the case that $\mathbb{P}_{z_1\sim\mathcal{N}(0,1)}\left[z_1\in[j\epsilon,(j+1)\epsilon)\right]\leq\epsilon$, we have

$$
\mathbb{E}_{\mathbf{x}\sim\mathcal{N}(0,I_d)}\left[\psi_{\text{down}}(\mathbf{x}) - \varphi_{\text{down}}(\mathbf{x})\right] \leq \sum_{j=0}^{k_1-1} O\left(\min\left(\frac{\log^3 k_2}{\sqrt{k_2}},\left(\frac{\tan\theta}{(j+1)\epsilon}\right)^2\right)\epsilon\right)
$$

$$
\leq \int_\infty^{+\infty} O\left(\min\left(\frac{\log^3 k_2}{\sqrt{k_2}},\left(\frac{\tan\theta}{z}\right)^2\right)\right)dz = O\left(\frac{\theta\log^{1.5}k_2}{k_2^{1/4}}\right),
$$

where the last step follows via precisely the same chain of inequalities as in Equation D.16.

In total, combining our bounds on $\mathbb{E}_{\mathbf{x}\sim\mathcal{N}(0,I_d)}\left[\psi_{\text{down}}(\mathbf{x}) - \varphi_{\text{down}}(\mathbf{x})\right], \mathbb{E}_{\mathbf{x}\sim\mathcal{N}(0,I_d)}\left[\varphi_{\text{up}}(\mathbf{x}) - \psi_{\text{up}}(\mathbf{x})\right]$ and $\mathbb{E}_{\mathbf{x}\sim\mathcal{N}(0,I_d)}\left[\psi_{\text{up}}(\mathbf{x}) - \psi_{\text{down}}(\mathbf{x})\right]$ we conclude that the quantity $\mathbb{E}_{\mathbf{x}\sim\mathcal{N}(0,I_d)}\left[\varphi_{\text{up}}(\mathbf{x}) - \varphi_{\text{down}}(\mathbf{x})\right]$ is at most $O\left(\frac{\log^{1.5}k_2}{k_2^{1/4}}\cdot\sphericalangle(\mathbf{v},\mathbf{v}')\right) + 10\epsilon$, as desired. $\qquad\square$

It only remains to prove Claim 2 to conclude the proof of the completeness condition.

*Proof of Claim 2.* Corrollary D.6 says that for any value of $t$, the degree-$k_2$ one-dimensional polynomials $R_{\text{up}}^t(z)$ and $R_{\text{down}}^t(z)$ have all their coefficients bounded by $O\left(2^{10k_2}\right)$. If one substitutes $\mathbf{v}\cdot\mathbf{x}$ in place of $z$ into either of these polynomials and opens the parentheses, the fact that $\mathbf{v}$ is a unit vector allows us to bound the size of the largest coefficients of $R_{\text{down}}^t(\mathbf{v}_\perp\cdot\mathbf{x})$ and $R_{\text{up}}^t(\mathbf{v}_\perp\cdot\mathbf{x})$ by $O((d+1)^{k_2}(k_2+1)2^{10k_2}) = O(d^{10k_2})$, proving the claim. $\qquad\square$

## D.4   Miscellaneous Claims

**Claim 3.** *There is a deterministic algorithm that given a unit vector $\mathbf{v}$ in $\mathbb{R}^d$, scalars $a$ and $b$, a monomial $m$ over $\mathbb{R}^d$ of degree at most $k_2$, an accuracy parameter $\beta\in(0,1]$, runs in time $\text{poly}\left((k_2 d)^{k_2}/\beta\right)$ and computes an approximation of $\mathbb{E}_{\mathbf{x}\sim\mathcal{N}(0,I_d)}\left[m(\mathbf{x})\cdot\mathbb{1}_{a\leq\mathbf{x}\cdot\mathbf{v}<b}\right]$ up to an additive error $\beta$.*

*Proof.* Firstly, we compute an orthonormal basis $\{\mathbf{w}_1, \cdots, \mathbf{w}_{d-1}\}$ for the $(d-1)$-dimensional subspace of $\mathbb{R}^d$ that is orthogonal to $\mathbf{v}$. We express $m(\mathbf{x}) = p(\mathbf{w_1} \cdot \mathbf{x}, \cdots, \mathbf{w_{d-1}} \cdot \mathbf{x}, \mathbf{v} \cdot \mathbf{x})$, and note that the polynomial $p$ has all its coefficients between 0 and $(d+1)^{k_2}$, and $p$ is comprised of at most $(d+1)^{k_2}$ monomials. Thus, to have an additive $\beta$-approximation for $\mathbb{E}_{\mathbf{x} \sim \mathcal{N}(0, I_d)} [m(\mathbf{x}) \cdot \mathbb{1}_{a \leq \mathbf{x} \cdot \mathbf{v} < b}]$, it suffices to compute for every monomial $m'$ of degree at most $k_2$ an additive $\frac{\beta}{d^{2k_2}}$-approximation to the quantity

$$\mathbb{E}_{\mathbf{x} \sim \mathcal{N}(0, I_d)} [m'(\mathbf{w_1} \cdot \mathbf{x}, \cdots, \mathbf{w_{d-1}} \cdot \mathbf{x}, \mathbf{v} \cdot \mathbf{x}) \cdot \mathbb{1}_{a \leq \mathbf{v} \cdot \mathbf{x} < b}],$$

which via the spherical symmetry of $\mathcal{N}(0, I_d)$ equals to $\mathbb{E}_{\mathbf{x} \sim \mathcal{N}(0, I_d)} [m'(\mathbf{x}) \cdot \mathbb{1}_{a \leq x_1 < b}]$.

Secondly, for every monomial $m'$ of degree at most $k_2$, we compute an approximation of the quantity $\mathbb{E}_{\mathbf{x} \sim \mathcal{N}(0, I_d)} [m'(\mathbf{x}) \cdot \mathbb{1}_{a \leq x_1 < b}]$ up to an additive error of $\frac{\beta}{10 d^{k_2}}$. To this end, we write $m'(\mathbf{x}) = \prod_i (x_i)^{\alpha_i}$ where $\sum_i \alpha_i \leq k_2$ and see that

$$\mathbb{E}_{\mathbf{x} \sim \mathcal{N}(0, I_d)} [m'(\mathbf{x}) \cdot \mathbb{1}_{a \leq x_1 < b}] = \underbrace{\left( \prod_{i > 1} (\alpha_i - 1)!! \mathbb{1}_{\alpha_i \text{ is even}} \right)}_{\leq k_2^{10 k_2}} \frac{1}{\sqrt{2\pi}} \int_a^b e^{-z^2/2} z^{\alpha_1} \, dz.$$

Note that $\alpha_1$ is an integer between 0 and $k_2$. Since we were seeking to compute a $\frac{\beta}{d^{2k_2}}$-approximation to $\mathbb{E}_{\mathbf{x} \sim \mathcal{N}(0, I_d)} [m'(\mathbf{x}) \cdot \mathbb{1}_{a \leq x_1 < b}]$, we see that this approximation can be obtained from the equaiton above together with an additive $\frac{\beta}{(d+1)^{2k_2} k^{10 k_2}}$-approximation to $\frac{1}{\sqrt{2\pi}} \int_a^b e^{-z^2/2} z^{\alpha_1} \, dz$. We denote $\rho(z) = e^{-z^2/2} z^{\alpha_1}$, and let $\beta' = \frac{\beta}{(d+1)^{2k_2} k_2^{10 k_2}}$. We see that the function $\rho$ has the following key properties:

1. For all $z$ in $\mathbb{R}^d$, the derivative $\rho'(z) = \alpha_1 e^{-z^2/2} z^{\alpha_1 - 1} \mathbb{1}_{\alpha_1 \geq 1} - e^{-z^2/2} z^{\alpha_1 + 1}$ we have $|\rho'(z)| \leq (k_2 + 1)^{k_2 + 1}$

2. For all $z_0$ in $\mathbb{R}^d$ satisfying $z_0 > 4k_2 + 2$ the value $\int_{|z| > z_0} \left| e^{-z^2/2} z^{\alpha_1} \right| dz$ is at most $\int_{|z| > z_0} e^{-z^2/4} \, dz$ which in turn is at most $e^{-z_0^2/4}$.

The three properties above imply that one can approximate the value of $\int_a^b \rho(z) \, dz$ up to an additive error of $\beta'$ via discretization, i.e., by splitting the interval $[a, b] \cap [-\sqrt{2 \ln(\beta')}, \sqrt{2 \ln(\beta')}]$ into intervals of size at most $\Delta$ and for each of these intervals $[a_j', b_j']$ use the inequality

$$\int_{a_j'}^{b_j'} \rho(z) \, dz = \rho(a_j')(a_j' - b_j') \pm \left( \sup_{z \in \mathbb{R}} |\rho'(z)| \right) (a_j' - b_j')^2,$$

which implies that

$$\int_{z \in [a,b]} \rho(z) \, dz = \overbrace{\int_{z \in [a,b] \cap \left[ -\sqrt{2 \ln(\beta')}, \sqrt{2 \ln(\beta')} \right]} \rho(z) \, dz \pm \frac{\beta'}{2}}^{\text{by property (2) of } \rho}$$

$$= \sum_j \left( \rho(a_j')(a_j' - b_j') \pm \left( \sup_{z \in \mathbb{R}} |\rho'(z)| \right) (a_j' - b_j')^2 \right) \pm \frac{\beta'}{2}$$

$$= \sum_j \rho(a_j')(a_j' - b_j') \pm \left( \underbrace{\left( \sup_{z \in \mathbb{R}} |\rho'(z)| \right)}_{\substack{\leq (k_2 + 1)^{k_2 + 1} \\ \text{by property (1) of } \rho}} \sqrt{8 \ln(\beta')} \Delta + \frac{\beta'}{2} \right),$$

which implies that if we take $\Delta$ to be $\frac{\beta'}{\sqrt{8 \ln(\beta')} (k_2 + 1)^{k_2 + 1}}$, then

$$\sum_j \rho(a_j')(a_j' - b_j') = \int_{z \in [a,b]} \rho(z) \, dz \pm \beta'.$$

Overall, evaluating the sum above requires one to compute $\rho(a'_j)$ on $\text{poly}((k_2)^{k_2}/\beta')$ values of $a'_j$. Therefore, substituting $\beta' = \frac{\beta}{(d+1)^{2k_2} k_2^{10k_2}}$ so it can be computed in time $\text{poly}((k_2 d)^{k_2}/\beta)$. $\qquad\square$

## E   Spectral Tester

In this section we prove the following theorem.

**Theorem E.1.** *There exists some absolute constant $C$ and a deterministic algorithm $\mathcal{T}_{spectral}$ that, given*

- *A positive integer $U \geq \left(\frac{Cd}{\epsilon\delta}\right)^C$.*

- *a dataset $S$ of points in $\mathbb{R}^d$ of size $M \leq U$.*

- *a unit vector $\mathbf{v}$ in $\mathbb{R}^d$,*

- *parameters $\epsilon, \delta$ and $\mu$ in $(0,1)$.*

*For every positive absolute constant $\mu$, the algorithm $\mathcal{T}_{spectral}$ runs in time $\text{poly}\left(\frac{dU}{\epsilon\delta}\right)$ and outputs Accept or output Reject. For all $\epsilon, \delta$ and $U \geq \left(\frac{Cd}{\epsilon\delta}\right)^C$ the algorithm $\mathcal{T}_{spectral}$ satisfies the following:*

- ***Completeness:** If $S$ consists of $M \leq U$ i.i.d. samples from the standard Gaussian distribution, then with probability at least $1 - O(\delta)$ the set $S$ is such that for all unit vectors $\mathbf{v}$ the algorithm $\mathcal{T}_{spectral}$ accepts when given $(U, S, \mathbf{v}, \epsilon, \delta, \mu)$ as the input.*

- ***Monotonicity under Datapoint Removal:** If the algorithm $\mathcal{T}_{spectral}$ outputs Accept for some specific input $(U, S, \mathbf{v}, \epsilon, \delta, \mu)$, then for all subsets $S' \subset S$ the tester $\mathcal{T}_{spectral}$ will also accept the input $(U, S', \mathbf{v}, \epsilon, \delta, \mu)$.*

- ***Soundness:** For any dataset $S$ and unit vector $\mathbf{v}$, if the tester $\mathcal{T}_{spectral}$ accepts the input $(U, S, \mathbf{v}, \epsilon, \delta, \mu)$ then for every unit vector $\mathbf{v}'$ in $\mathbb{R}^d$ we have*

$$\frac{1}{U} \sum_{\mathbf{x} \in S} \left[ \mathbb{1}_{\text{sign}(\mathbf{x}\cdot\mathbf{v}) \neq \text{sign}(\mathbf{x}\cdot\mathbf{v}')} \right] \leq (1+\mu)\frac{\angle(\mathbf{v}, \mathbf{v}')}{\pi} + O(\epsilon).$$

We argue that the following algorithm (which is essentially a restatement of Algorithm 2) satisfies the specifications above:

- **Given:** parameter $\epsilon, \delta, \mu$ in $(0,1)$, dataset $S$ of points in $\mathbb{R}^d$ of size $M \leq U$, a unit vector $\mathbf{v}$ in $\mathbb{R}^d$,

1. $k_1 \leftarrow \frac{2\sqrt{\log 2/\epsilon}}{\epsilon}$, $k_2 \leftarrow \frac{C^{0.1}}{\mu^5}$

2. $\Delta \leftarrow \frac{60(4k_2)^{2k_2+2}(d+1)^{6k_2+2}}{\delta}\left(\frac{\log U}{U}\right)^{1/4}$

3. For all $a$ and $b$ in $\{-\infty, -k_1\epsilon, -(k_1-1)\epsilon, \cdots, -\epsilon, 0, +\epsilon, \cdots, (k_1-1)\epsilon, k_1\epsilon, +\infty\}$

   (a) Compute $W^{a,b}$ such that

   $$W^{a,b} - \Delta I_{\binom{d+1}{k_2} \times \binom{d+1}{k_2}} \preceq \mathbb{E}_{\mathbf{x}\sim\mathcal{N}}\left[\left(\mathbf{x}^{\otimes k_2}\right)\left(\mathbf{x}^{\otimes k_2}\right)^\top \cdot \mathbb{1}_{a \leq \mathbf{x}\cdot\mathbf{v} < b}\right] \preceq W^{a,b} + \Delta I_{\binom{d+1}{k_2} \times \binom{d+1}{k_2}}$$
   (E.1)

   (For how to compute this approximation, see Claim 6).

   (b) If the following does not hold:

   $$\frac{1}{U} \sum_{\mathbf{x} \in S} \left(\mathbf{x}^{\otimes k_2}\right)\left(\mathbf{x}^{\otimes k_2}\right)^\top \mathbb{1}_{a \leq \mathbf{x}\cdot\mathbf{v} < b} \preceq W^{a,b} + 3\Delta I_{\binom{d+1}{k_2} \times \binom{d+1}{k_2}}, \qquad \text{(E.2)}$$

   then output Reject.

4. If did not reject in any previous step, output Accept.

It is immediate that the algorithm indeed runs in time $poly\left(\frac{dU}{\epsilon\delta}\right)$, because step (2b) can be performed by computing the largest eigenvalue of a $\binom{d+1}{k_2} \times \binom{d+1}{k_2}$-sized matrix. Monotonicity over datapoint removal also follows immediately since if $S' \subset S$ then

$$\frac{1}{U} \sum_{\mathbf{x} \in S'} \left(\mathbf{x}^{\otimes k_2}\right) \left(\mathbf{x}^{\otimes k_2}\right)^\top \preceq \frac{1}{U} \sum_{\mathbf{x} \in S} \left(\mathbf{x}^{\otimes k_2}\right) \left(\mathbf{x}^{\otimes k_2}\right)^\top,$$

and therefore if the condition in step (4) holds for $S$ then it will also hold for $S'$.

### E.1 Completeness

Since we have already proven the property of monotonicity under datapoint removal, we can assume without loss of generality that $M = U$. If not, the set $S$ can be obtained by first taking $U$ samples from $\mathcal{N}(0, I_d)$ and then removing the last $U - M$ of them. If the $\mathcal{T}_{\text{spectral}}$ accepted the dataset before removing these points, then it will also accept it after these datapoints are removed.

Suppose the set dataset $S$ consists of $U$ i.i.d. samples from $\mathcal{N}(0, I_d)$. Similar to Section D.1, we again note that the collection $\mathcal{H}$ of sets of the form $\mathbb{1}_{a \leq \mathbf{v} \cdot \mathbf{x} < b}$ has VC dimension at most $(d+1)^2$. Lemma A.1 then implies that with probability at least $1 - \delta$ for all pairs of $a$ and $b$, for every unit vector $\mathbf{v}$ and for every polynomial $p$ of degree at most $k_2$, if $B_p$ denotes the largest coefficient of $p$ (in absolute value) then we have

$$\left| \frac{1}{U} \sum_{\mathbf{x} \in S} \left[ (p(\mathbf{x}))^2 \mathbb{1}_{\mathbf{x} \in A} \right] - \mathop{\mathbb{E}}_{\mathbf{x} \sim \mathcal{N}(0, I_d)} \left[ (p(\mathbf{x}))^2 \mathbb{1}_{\mathbf{x} \in A} \right] \right| \leq 2B_p^2 (d+1)^{5k_2} \sqrt{\frac{(4k_2)^{2k_2+2}}{\delta U}},$$

$$\left| \frac{1}{U} \sum_{\mathbf{x} \in S} \left[ (p(\mathbf{x}))^2 \mathbb{1}_{\mathbf{x} \in A} \right] - \mathop{\mathbb{E}}_{\mathbf{x} \sim \mathcal{N}(0, I_d)} \left[ (p(\mathbf{x}))^2 \mathbb{1}_{\mathbf{x} \in A} \right] \right| \leq \frac{60 B_p^2 (4k_2)^{2k_2+2} (d+1)^{6k_2+2}}{\delta} \left( \frac{\log U}{U} \right)^{1/4},$$

Combining this with Equation A.1 we get

$$\frac{1}{U} \sum_{\mathbf{x} \in S} \left[ (p(\mathbf{x}))^2 \mathbb{1}_{\mathbf{x} \in A} \right] \leq \mathop{\mathbb{E}}_{\mathbf{x} \sim \mathcal{N}(0, I_d)} \left[ (p(\mathbf{x}))^2 \mathbb{1}_{\mathbf{x} \in A} \right] + (\|p\|_{\text{coeff}})^2 \frac{60 (4k_2)^{2k_2+2} (d+1)^{6k_2+2}}{\delta} \left( \frac{\log U}{U} \right)^{1/4}$$

and Claim 6 implies that Equation E.17 holds which implies that

$$p^\top W^{a,b} p \geq \mathop{\mathbb{E}}_{\mathbf{x} \sim \mathcal{N}(0, I_d)} \left[ (p(\mathbf{x}))^2 \mathbb{1}_{\mathbf{x} \in A} \right] + (\|p\|_{\text{coeff}})^2 \frac{60 B_p^2 (4k_2)^{2k_2+2} (d+1)^{6k_2+2}}{\delta} \left( \frac{\log U}{U} \right)^{1/4}.$$

Combining the last two equations above, we get:

$$\frac{1}{U} \sum_{\mathbf{x} \in S} \left[ (p(\mathbf{x}))^2 \mathbb{1}_{\mathbf{x} \in A} \right] \leq p^\top W^{a,b} p + (\|p\|_{\text{coeff}})^2 \frac{120 B_p^2 (4k_2)^{2k_2+2} (d+1)^{6k_2+2}}{\delta} \left( \frac{\log U}{U} \right)^{1/4}.$$

Recalling the notation in A.1, we see that the assertion that the inequality above holds for every $p$, is equivalent to the following matrix inequality

$$\frac{1}{U} \sum_{\mathbf{x} \in S} \left[ \mathbf{x}^{\otimes k_2} \left( \mathbf{x}^{\otimes k_2} \right)^\top \mathbb{1}_{\mathbf{x} \in A} \right] \preceq W^{a,b} + \frac{120 B_p^2 (4k_2)^{2k_2+2} (d+1)^{6k_2+2}}{\delta} \left( \frac{\log U}{U} \right)^{1/4} I_{\binom{d+1}{k_2} \times \binom{d+1}{k_2}}.$$

Finally, substituting $k_2 = \frac{C^{0.1}}{\mu^5}$ and $U \geq \left( \frac{Cd}{\epsilon\delta} \right)^C$, we see that for a sufficiently large absolute constant $C$, the inequality above implies Equation A.1, and thus $\mathcal{T}_{\text{spectral}}$ accepts.

### E.2 Soundness

In order to deduce the soundness condition, expand upon definitions introduced in D.2. We emphasize that unlike the $L_1$-sandwiching degree used to analyze the disagreement tester, here we use the notion of $L_2$-sandwiching polynomials.

**Definition E.2.** We say that a function $f : \mathbb{R}^d \to \{0, 1\}$ is $\epsilon$-sandwiched in $L_2$ norm between a pair of functions $f_{\text{up}} : \mathbb{R}^d \to \mathbb{R}$ and $f_{\text{down}} : \mathbb{R}^d \to \mathbb{R}$ under $\mathcal{N}(0, I_d)$ if:

- For all $\mathbf{x}$ in $\mathbb{R}^d$ we have $f_{\text{down}}(\mathbf{x}) \le f(\mathbf{x}) \le f_{\text{up}}(\mathbf{x})$

- $\mathbb{E}_{\mathbf{x} \sim \mathcal{N}(0, I_d)} \left[ (f_{\text{up}}(\mathbf{x}) - f_{\text{down}}(\mathbf{x}))^2 \right] \le \epsilon.$

**Definition E.3.** We say that a function $f : \mathbb{R}^d \to \{0, 1\}$ has $(\epsilon, B)$-sandwiching degree of at most $k$ in $L_2$ norm under $\mathcal{N}(0, I_d)$ with respect to a partition $\mathcal{C}$ of $\mathbb{R}^d$ if the function $f$ is $\epsilon$-sandwiched in $L_2$ norm under $\mathcal{N}(0, I_d)$ between $\sum_{A \in \mathcal{C}} \left( p_{\text{down}}^A \mathbb{1}_A \right)$ and $\sum_{A \in \mathcal{C}} \left( p_{\text{up}}^A \mathbb{1}_A \right)$, where $p_{\text{up}}^A$ and $p_{\text{down}}^A$ are degree$-k$ polynomials over $\mathbb{R}^d$ whose coefficients are bounded by $B$ in absolute value.

Subsection E.3 is dedicated to proving the following bound on the $L_2$-sandwiching degree of a specific family of functions with respect to a specific partition of $\mathbb{R}^d$.

**Proposition E.4.** *For all $\epsilon$ and $k_2$, let $k_1 = \frac{2\sqrt{\log 2/\epsilon}}{\epsilon}$, and let $\mathbf{v}$ be a unit vector in $\mathbb{R}^d$. Then, there exists a partition $\mathcal{C}$ of $\mathbb{R}^d$ consisting of sets of the form $\{\mathbf{x} \in \mathbb{R}^d : a \le \mathbf{v} \cdot \mathbf{x} \le b\}$ for a certain collection of pairs $a, b$ in $\{-\infty, -k_1\epsilon, -(k_1-1)\epsilon, \cdots, -\epsilon, 0, +\epsilon, \cdots, (k_1-1)\epsilon, k_1\epsilon, +\infty\}$. Then, for every unit vector $\mathbf{v}'$, the function $f(\mathbf{x}) = \mathbb{1}_{\text{sign}(\mathbf{v} \cdot \mathbf{x}) \ne \text{sign}(\mathbf{v}' \cdot \mathbf{x})}$ has $(O(\measuredangle(\mathbf{v}, \mathbf{v}') \frac{\log^5 k_2}{k_2^{1/4}} \cdot) + 10\epsilon, O(d^{10k_2}))$-sandwiching degree of at most $k_2$ in $L_2$ norm under $\mathcal{N}(0, I_d)$ with respect to the partition $\mathcal{C}$ of $\mathbb{R}^d$.*

A bound on the sandwiching degree of a class of functions leads to a guarantee for the tester $\mathcal{T}_{\text{disagreement}}$:

**Proposition E.5.** *Let $\mathcal{C}$ be a partition of $\mathbb{R}^d$ and $f$ a $\{0, 1\}$-valued function that has $(\nu, B)$-sandwiching degree of at most $k_2$ in $L_2$ norm under $\mathcal{N}(0, I_d)$ with respect to the partition $\mathcal{C}$. If a set $S$ of points in $\mathbb{R}^d$ satisfies the following condition for all $A$ in $\mathcal{C}$ :*

$$\frac{1}{U} \sum_{\mathbf{x} \in S} (\mathbf{x}^{\otimes k_2})(\mathbf{x}^{\otimes k_2})^\top \mathbb{1}_{\mathbf{x} \in A} \preceq \mathbb{E}_{\mathbf{x} \sim \mathcal{N}(0, I_d)} [(\mathbf{x}^{\otimes k_2})(\mathbf{x}^{\otimes k_2})^\top \mathbb{1}_{\mathbf{x} \in A}] + \frac{\epsilon^2}{|\mathcal{C}| B^2 (d+1)^{k_2}} I_{\binom{d+1}{k_2} \times \binom{d+1}{k_2}} \tag{E.3}$$

*then we have*

$$\sqrt{\frac{1}{U} \sum_{\mathbf{x} \sim S} \left[ \mathbb{1}_{f(\mathbf{x}) = 1} \right]} \le \sqrt{\mathbb{P}_{\mathbf{x} \sim \mathcal{N}(0, I_d)} [f(\mathbf{x}) = 1]} + \sqrt{\nu} + \epsilon.$$

*Proof.* Since $f$ has $(\nu, B)$-sandwiching degree of at most $k_2$ in $L_2$ norm under $\mathcal{N}(0, I_d)$ with respect to the partition $\mathcal{C}$, we have a collection of polynomials $\{p_{\text{down}}^A, p_{\text{up}}^A\}$ for all $A$ in $\mathcal{C}$ that have coefficients bounded by $B$, satisfy for all $\mathbf{x}$ the condition

$$f(\mathbf{x}) \in \Big[ \sum_{A \in \mathcal{C}} \left( p_{\text{down}}^A(\mathbf{x}) \mathbb{1}_{\mathbf{x} \in A} \right), \sum_{A \in \mathcal{C}} \left( p_{\text{up}}^A(\mathbf{x}) \mathbb{1}_{\mathbf{x} \in A} \right) \Big], \tag{E.4}$$

as well as

$$\mathbb{E}_{\mathbf{x} \sim \mathcal{N}(0, I_d)} \Big[ \Big( \sum_{A \in \mathcal{C}} \left( p_{\text{up}}^A(\mathbf{x}) \mathbb{1}_{\mathbf{x} \in A} \right) - \sum_{A \in \mathcal{C}} \left( p_{\text{down}}^A(\mathbf{x}) \mathbb{1}_{\mathbf{x} \in A} \right) \Big)^2 \Big] \le \nu. \tag{E.5}$$

For all $\mathbf{x}$ in $\mathbb{R}^d$ we have $f(\mathbf{x}) \le (f(\mathbf{x}))^2 \le (\sum_{A \in \mathcal{C}} (p_{\text{up}}^A(\mathbf{x}) \mathbb{1}_{\mathbf{x} \in A}))^2$. Since all distinct pairs $A_1, A_2$ in $\mathcal{C}$ are disjoint, we have $(\sum_{A \in \mathcal{C}} (p_{\text{up}}^A(\mathbf{x}) \mathbb{1}_{\mathbf{x} \in A}))^2 = \sum_{A \in \mathcal{C}} (p_{\text{up}}^A(\mathbf{x}) \mathbb{1}_{\mathbf{x} \in A})^2$. Therefore, we have

$$\frac{1}{U} \sum_{\mathbf{x} \sim S} [f(\mathbf{x})] \le \sum_{A \in \mathcal{C}} \left( \frac{1}{U} \sum_{\mathbf{x} \sim S} \left[ \left( p_{\text{up}}^A(\mathbf{x}) \mathbb{1}_{\mathbf{x} \in A} \right)^2 \right] \right) \tag{E.6}$$

Referring to definitions in Subsection A.1, we see that Equation E.3 is equivalent to the assertion that for every $A$ in $\mathcal{C}$ and every degree-$k_2$ polynomial $p$ we have

$$\frac{1}{U} \sum_{\mathbf{x} \in S} (p(\mathbf{x}))^2 \, \mathbb{1}_{\mathbf{x} \in A} \le \mathbb{E}_{\mathbf{x} \sim \mathcal{N}(0, I_d)} \left[ (p(\mathbf{x}))^2 \, \mathbb{1}_{\mathbf{x} \in A} \right] + \frac{\epsilon^2}{|\mathcal{C}| B^2 (d+1)^{k_2}} \left( \|p\|_{\text{coeff}} \right)^2.$$

Choosing $p = p_{\text{up}}^A$ in the inequality above and combining with Equation E.6 we get:

$$\frac{1}{U}\sum_{\mathbf{x}\sim S}\Big[f(\mathbf{x})\Big] \leq \sum_{A\in\mathcal{C}}\Big(\mathop{\mathbb{E}}_{\mathbf{x}\sim\mathcal{N}(0,I_d)}[(p_{\text{up}}^A(\mathbf{x})\mathbb{1}_{\mathbf{x}\in A})^2] + \frac{\epsilon^2}{|\mathcal{C}|B^2(d+1)^{k_2}}(\|p_{\text{up}}^A\|_{\text{coeff}})^2\Big).$$

By Equation A.1, we have $\|p_{\text{up}}^A\|_{\text{coeff}} \leq B^2 d^{k_2}$. Substituting this and again recalling that all distinct pairs $A_1, A_2$ in $\mathcal{C}$ are disjoint, we obtain

$$\frac{1}{U}\sum_{\mathbf{x}\sim S}[f(\mathbf{x})] \leq \mathop{\mathbb{E}}_{\mathbf{x}\sim\mathcal{N}(0,I_d)}\Big[\Big(\sum_{A\in\mathcal{C}}p_{\text{up}}^A(\mathbf{x})\mathbb{1}_{\mathbf{x}\in A}\Big)^2\Big] + \epsilon^2.$$

Taking square roots of both sides gives us

$$\sqrt{\frac{1}{U}\sum_{\mathbf{x}\sim S}[f(\mathbf{x})]} \leq \sqrt{\mathop{\mathbb{E}}_{\mathbf{x}\sim\mathcal{N}_d}\Big[\Big(\sum_{A\in\mathcal{C}}p_{\text{up}}^A(\mathbf{x})\mathbb{1}_{\mathbf{x}\in A}\Big)^2\Big] + \epsilon^2}$$

$$\leq \sqrt{\mathop{\mathbb{E}}_{\mathbf{x}\sim\mathcal{N}_d}\Big[\Big(\sum_{A\in\mathcal{C}}p_{\text{up}}^A(\mathbf{x})\mathbb{1}_{\mathbf{x}\in A}\Big)^2\Big]} + \epsilon. \tag{E.7}$$

Equation E.4, together with the triangle inequality and the fact that $(f(\mathbf{x}))^2 = f(\mathbf{x})$, implies that

$$\sqrt{\mathop{\mathbb{E}}_{\mathbf{x}\sim\mathcal{N}(0,I_d)}\Big[\Big(\sum_{A\in\mathcal{C}}p_{\text{up}}^A(\mathbf{x})\mathbb{1}_{\mathbf{x}\in A}\Big)^2\Big]} \leq$$

$$\leq \sqrt{\mathop{\mathbb{E}}_{\mathbf{x}\sim\mathcal{N}(0,I_d)}\Big[f(\mathbf{x})\Big]} + \sqrt{\mathop{\mathbb{E}}_{\mathbf{x}\sim\mathcal{N}(0,I_d)}\Big[\Big(\sum_{A\in\mathcal{C}}p_{\text{up}}^A(\mathbf{x})\mathbb{1}_{\mathbf{x}\in A} - f(\mathbf{x})\Big)^2\Big]}$$

$$\leq \sqrt{\mathop{\mathbb{E}}_{\mathbf{x}\sim\mathcal{N}(0,I_d)}\Big[f(\mathbf{x})\Big]} + \sqrt{\mathop{\mathbb{E}}_{\mathbf{x}\sim\mathcal{N}(0,I_d)}\Big[\Big(\sum_{A\in\mathcal{C}}p_{\text{up}}^A(\mathbf{x})\mathbb{1}_{\mathbf{x}\in A} - \sum_{A\in\mathcal{C}}p_{\text{down}}^A(\mathbf{x})\mathbb{1}_{\mathbf{x}\in A}\Big)^2\Big]}$$

Substituting Equation E.5, we get

$$\sqrt{\mathop{\mathbb{E}}_{\mathbf{x}\sim\mathcal{N}(0,I_d)}\Big[\Big(\sum_{A\in\mathcal{C}}p_{\text{up}}^A(\mathbf{x})\mathbb{1}_{\mathbf{x}\in A}\Big)^2\Big]} \leq \sqrt{\mathop{\mathbb{E}}_{\mathbf{x}\sim\mathcal{N}(0,I_d)}[f(\mathbf{x})]} + \sqrt{\nu},$$

which combined with Equation E.7 finishes the proof. $\qquad\square$

Claim 6 implies that matrices $W^{a,b}$ satisfy Equation E.1, which implies that for all monomials $p$ of degree at most $k_2$ we have

$$W^{a,b} \preceq \mathop{\mathbb{E}}_{\mathbf{x}\sim\mathcal{N}(0.I_d)}\Big[\big(\mathbf{x}^{\otimes k_2}\big)\big(\mathbf{x}^{\otimes k_2}\big)^\top \mathbb{1}_{a\leq\mathbf{x}\cdot\mathbf{v}<b}\Big] + \Delta I_{\binom{d+1}{k_2}\times\binom{d+1}{k_2}}$$

If the above is the case, and the algorithm accepts, then we have for all pairs of $a$ and $b$ in $\{-\infty, -k_1\epsilon, -(k_1-1)\epsilon, \cdots, -\epsilon, 0, +\epsilon, \cdots, (k_1-1)\epsilon, k_1\epsilon, +\infty\}$ that

$$\frac{1}{U}\sum_{\mathbf{x}\in S}\big(\mathbf{x}^{\otimes k_2}\big)\big(\mathbf{x}^{\otimes k_2}\big)^\top \mathbb{1}_{a\leq\mathbf{x}\cdot\mathbf{v}<b} \preceq$$

$$\mathop{\mathbb{E}}_{\mathbf{x}\sim\mathcal{N}(0.I_d)}\Big[\big(\mathbf{x}^{\otimes k_2}\big)\big(\mathbf{x}^{\otimes k_2}\big)^\top \mathbb{1}_{a\leq\mathbf{x}\cdot\mathbf{v}<b}\Big] + \frac{210(4k_2)^{2k_2+2}(d+1)^{6k_2+2}}{\delta}\Big(\frac{\log U}{U}\Big)^{1/4} I_{\binom{d+1}{k_2}\times\binom{d+1}{k_2}}.$$

Taking the equation above, together with Proposition E.5 and Proposition E.4 we conclude that for all unit vectors $\mathbf{v}'$:

$$\sqrt{\frac{1}{U}\sum_{\mathbf{x}\sim S}\Big[\mathbb{1}_{\text{sign}(\mathbf{x}\cdot\mathbf{v})\neq\text{sign}(\mathbf{x}\cdot\mathbf{v}')}\Big]} \leq \sqrt{\mathop{\mathbb{P}}_{\mathbf{x}\sim\mathcal{N}(0,I_d)}[\text{sign}(\mathbf{v}\cdot\mathbf{x})\neq\text{sign}(\mathbf{v}'\cdot\mathbf{x})]} + \sqrt{O\Big(\frac{\log^5 k_2}{k_2^{1/4}}\measuredangle(\mathbf{v},\mathbf{v}')\Big)} +$$

$$+ O\Big(\sqrt{(2k_1+2)\frac{(4k_2)^{2k_2+2}(d+1)^{7k_2+2}}{\delta}\Big(\frac{\log U}{U}\Big)^{1/4}d^{10k_2}}\Big)$$

Substituting $k_1 \leftarrow \frac{2\sqrt{\log 2/\epsilon}}{\epsilon}$, $k_2 \leftarrow \frac{C^{0.1}}{\mu^5}$, $Y \leftarrow C\left(\frac{(kd)^{k_2}}{\delta}\right)^C$, taking $C$ to be a sufficiently large absolute constant and recalling that $\mathbb{P}_{\mathbf{x} \sim \mathcal{N}(0, I_d)}[\text{sign}(\mathbf{v} \cdot \mathbf{x}) \neq \text{sign}(\mathbf{v}' \cdot \mathbf{x})]$ equals to $\angle(\mathbf{v}, \mathbf{v}')/\pi$ we conclude that

$$\frac{1}{U} \sum_{\mathbf{x} \sim S} \left[ \mathbb{1}_{\text{sign}(\mathbf{x} \cdot \mathbf{v}) \neq \text{sign}(\mathbf{x} \cdot \mathbf{v}')} \right] \leq (1 + \mu) \frac{\angle(\mathbf{v}, \mathbf{v}')}{\pi} + O(\epsilon).$$

### E.3 Bounding the $L_2$ sandwiching degree of the disagreement region

To prove Proposition E.4, we follow an exactly analogous approach as the one for Proposition D.5. We will need the following result from [KSV24b]:

**Fact E.6** ([KSV24b]). *For every positive integer $k$ and a real value $t$, the function $f(z) = \mathbb{1}_{z \leq t}$ has $(O(\frac{\log^{10} k}{\sqrt{k}}), O(2^{10k}))$-sandwiching degree in $L_2$ norm of at most $k$ under $\mathcal{N}(0, 1)$.*

The following corollary slightly strengthens the fact above:

**Corollary E.7.** *For every positive integer $k \geq 4$ and a real value $t$, the function $f(z) = \mathbb{1}_{z \leq t}$ is $(O(\min(\frac{\log^{10} k}{\sqrt{k}}, \frac{1}{t^2})), 2^{10k})$-sandwiched in $L_2$ norm under $\mathcal{N}(0, 1)$ between a pair of polynomials $J_{up}^t$ and $J_{down}^t$ of degree of at most $k$.*

*Proof.* Indeed, if $\frac{\log^{10} k}{\sqrt{k}} \leq \frac{1024}{t^2}$ then the corollary follows from Fact E.6. So all we need to do is to consider the other case. We see that either $t > 1$ or $t < -1$ (since $k \geq 4$). if $t > 1$ we take $p_{\text{down}}(\mathbf{x}) = 0$ and $p_{\text{up}}(\mathbf{x}) = \left(\frac{x}{t}\right)^2$. If $t < -1$, we take take $p_{\text{up}}(\mathbf{x}) = 1$ and $p_{\text{down}}(\mathbf{x}) = 1 - \left(\frac{x}{t}\right)^2$. In either case, the polynomials $p_{\text{down}}, p_{\text{up}}$ form a pair of $(O(\min(\frac{\log^{10} k}{\sqrt{k}}, \frac{1}{t^2})), 1)$-sandwiching polynomials of degree 2. $\square$

Let $\mathbf{v}_\perp$ be the unit vector equal up to scaling to the component of $\mathbf{v}'$ perpendicular to $\mathbf{v}$. Then, we can write

$$\psi_{\text{down}}(\mathbf{x}) \leq \psi_{\text{up}}(\mathbf{x}) \leq \mathbb{1}_{\mathbf{v} \cdot \mathbf{x} \geq 0 \wedge \mathbf{v}' \cdot \mathbf{x} < 0} \tag{E.8}$$

$$\psi_{\text{down}}(\mathbf{x}) \leq \psi_{\text{up}}(\mathbf{x}) \leq \mathbb{1}_{\mathbf{v} \cdot \mathbf{x} > 0 \wedge \mathbf{v}' \cdot \mathbf{x} \leq 0} \tag{E.9}$$

where

$$\psi_{\text{up}}(\mathbf{x}) = \begin{cases} 1 & \text{if } \mathbf{v} \cdot \mathbf{x} \geq k_1 \epsilon \text{ or } \mathbf{v} \cdot \mathbf{x} = 0, \\ \mathbb{1}_{\mathbf{v}_\perp \cdot \mathbf{x} \tan \theta \leq -j\epsilon} & \text{if } \mathbf{v} \cdot \mathbf{x} \neq 0 \text{ and } \mathbf{v} \cdot \mathbf{x} \in [j\epsilon, (j+1)\epsilon) \text{ for } 0 \leq j \leq k_1 - 1, \\ 0 & \text{if } \mathbf{v} \cdot \mathbf{x} < 0 \end{cases}$$

$$\psi_{\text{down}}(\mathbf{x}) = \begin{cases} 0 & \text{if } \mathbf{v} \cdot \mathbf{x} \geq k_1 \epsilon \text{ or } \mathbf{v} \cdot \mathbf{x} = 0 \\ \mathbb{1}_{\mathbf{v}_\perp \cdot \mathbf{x} \tan \theta < -(j+1)\epsilon} & \text{if } \mathbf{v} \cdot \mathbf{x} \neq 0 \text{ and } \mathbf{v} \cdot \mathbf{x} \in [j\epsilon, (j+1)\epsilon) \text{ for } 0 \leq j \leq k_1 - 1 \\ 0 & \text{if } \mathbf{v} \cdot \mathbf{x} < 0 \end{cases}$$

Recall that for every $t \in \mathbb{R}$, Corollary E.7 gives us one-dimensional degree-$k_2$ sandwiching polynomials $J_{\text{down}}^t(z)$ and $J_{\text{up}}^t(z)$ for $\mathbb{1}_{z \leq t}$ under $L_2$ norm. Using this notation, we have for all $\mathbf{x}$ in

$\mathbb{R}^d$

$$\overbrace{\sum_{j=0}^{k_1-1} \mathbb{1}_{\mathbf{v}\cdot\mathbf{x}\cdot[j\epsilon,(j+1)\epsilon)} J_{down}^{-(j+1)\epsilon/\tan\theta}(\mathbf{v}_\perp\cdot\mathbf{x})}^{\text{Denote this } \varphi_{down}^{L_2}(\mathbf{x})} \le \psi_{down}(\mathbf{x}) \le \mathbb{1}_{\mathbf{v}\cdot\mathbf{x}\ge0\wedge\mathbf{v}'\cdot\mathbf{x}<0} \le$$

$$\le \psi_{up}(\mathbf{x}) \le \underbrace{\mathbb{1}_{\mathbf{v}\cdot\mathbf{x}\ge k_1\epsilon} + \sum_{j=0}^{k_1-1} \mathbb{1}_{\mathbf{v}\cdot\mathbf{x}\cdot[j\epsilon,(j+1)\epsilon)} J_{up}^{-j\epsilon/\tan\theta}(\mathbf{v}_\perp\cdot\mathbf{x})}_{\text{Denote this } \varphi_{up}^{L_2}(\mathbf{x})} \quad \text{(E.10)}$$

In order to conclude Proposition E.4. We show the following two claims:

**Claim 4.** *We have*

$$\mathop{\mathbb{E}}_{\mathbf{x}\sim\mathcal{N}(0,I_d)}\left[\left(\varphi_{up}^{L_2}(\mathbf{x}) - \varphi_{down}^{L_2}(\mathbf{x})\right)^2\right] \le O\left(\frac{\log^5 k_2}{k_2^{1/4}}\cdot\measuredangle(\mathbf{v},\mathbf{v}')\right) + 10\epsilon$$

**Claim 5.** *For all integers $j$ in $[0, k_1 - 1]$, every coefficient of $J_{down}^{-(j+1)\epsilon/\tan\theta}(\mathbf{v}_\perp\cdot\mathbf{x})$ and $J_{up}^{-j\epsilon/\tan\theta}(\mathbf{v}_\perp\cdot\mathbf{x})$ is at most $O\left(d^{10k_2}\right)$ in absolute value.*

Proposition E.4 follows from the two claims above for as follows. We first observe that Equations E.9 and E.10 imply that

$$\varphi_{down}^{L_2}(-\mathbf{x}) \le \mathbb{1}_{\mathbf{v}\cdot\mathbf{x}<0\wedge\mathbf{v}'\cdot\mathbf{x}\ge0} \le \varphi_{up}^{L_2}(-\mathbf{x}).$$

Recalling our convention that $\text{sign}(0) = 1$, we see that

$$\mathbb{1}_{\text{sign}(\mathbf{v}\cdot\mathbf{x})\ne\text{sign}(\mathbf{v}'\cdot\mathbf{x})} = \mathbb{1}_{\mathbf{v}\cdot\mathbf{x}\ge0\wedge\mathbf{v}'\cdot\mathbf{x}<0} + \mathbb{1}_{\mathbf{v}\cdot\mathbf{x}<0\wedge\mathbf{v}'\cdot\mathbf{x}\ge0}$$

this, together with E.10 allows us to bound

$$\varphi_{down}^{L_2}(\mathbf{x}) + \varphi_{down}^{L_2}(-\mathbf{x}) \le \mathbb{1}_{\text{sign}(\mathbf{v}\cdot\mathbf{x})\ne\text{sign}(\mathbf{v}'\cdot\mathbf{x})} \le \varphi_{up}^{L_2}(\mathbf{x}) + \varphi_{up}^{L_2}(-\mathbf{x}), \quad \text{(E.11)}$$

Claim E.10 allows us to conclude that

$$\mathop{\mathbb{E}}_{\mathbf{x}\sim\mathcal{N}(0,I_d)}\left[\left(\varphi_{up}^{L_2}(\mathbf{x}) + \varphi_{up}^{L_2}(-\mathbf{x}) - \varphi_{down}^{L_2}(\mathbf{x}) - \varphi_{down}^{L_2}(-\mathbf{x})\right)^2\right] \le$$

$$2\left(\mathop{\mathbb{E}}_{\mathbf{x}\sim\mathcal{N}(0,I_d)}\left[\left(\varphi_{up}^{L_2}(\mathbf{x}) - \varphi_{down}^{L_2}(\mathbf{x})\right)^2\right] + \mathop{\mathbb{E}}_{\mathbf{x}\sim\mathcal{N}(0,I_d)}\left[\left(\varphi_{up}^{L_2}(-\mathbf{x}) - \varphi_{down}^{L_2}(-\mathbf{x})\right)^2\right]\right) \le$$

$$O\left(\frac{\log^{1.5} k_2}{k_2^{1/4}}\cdot\measuredangle(\mathbf{v},\mathbf{v}')\right) + 20\epsilon. \quad \text{(E.12)}$$

Equations E.11 and E.12, together with comparing the definition of $\varphi_{up}$ and $\varphi_{down}$ with Definition E.3 and recalling Claim 5, allow us to conclude that there exists a partition $\mathcal{C}$ of $\mathbb{R}^d$ consisting of sets of the form $\left\{\mathbf{x}\in\mathbb{R}^d : a \le \mathbf{v}\cdot\mathbf{x} \le b\right\}$ for a certain collection of pairs $a, b$ in

$\{-\infty, -k_1\epsilon, -(k_1-1)\epsilon, \cdots, -\epsilon, 0, +\epsilon, \cdots, (k_1-1)\epsilon, k_1\epsilon, +\infty\}$, such that for every unit vector $\mathbf{v}'$, the function $f(\mathbf{x}) = \mathbb{1}_{\text{sign}(\mathbf{v}\cdot\mathbf{x})\ne\text{sign}(\mathbf{v}'\cdot\mathbf{x})}$ has $\left(O\left(\frac{\measuredangle(\mathbf{v},\mathbf{v}')}{k_2^{1/4}}\right) + 10\epsilon, O\left(d^{10k_2}\right)\right)$-sandwiching degree of at most $k_2$ in $L_1$ norm under $\mathcal{N}(0, I_d)$ with respect to the partition $\mathcal{C}$ of $\mathbb{R}^d$. This implies Proposition E.4.

We now proceed to proving Claim 4

*Proof of Claim 4.* We have:

$$\mathop{\mathbb{E}}_{\mathbf{x}\sim\mathcal{N}(0,I_d)}\left[(\psi_{\mathrm{up}}(\mathbf{x})-\psi_{\mathrm{down}}(\mathbf{x}))^2\right] = \mathop{\mathbb{E}}_{\mathbf{x}\sim\mathcal{N}(0,I_d)}\left[\psi_{\mathrm{up}}(\mathbf{x})-\psi_{\mathrm{down}}(\mathbf{x})\right] \le$$

$$\mathop{\mathbb{P}}_{\mathbf{x}\sim\mathcal{N}(0,I_d)}\left[\mathbf{v}\cdot\mathbf{x} > k_1\epsilon\right] + \sum_{j=0}^{k_1-1}\mathop{\mathbb{P}}_{\mathbf{x}\sim\mathcal{N}(0,I_d)}\left[\{\mathbf{v}\cdot\mathbf{x}\in[j\epsilon,(j+1)\epsilon)\}\wedge\{\mathbf{v}_\perp\cdot\mathbf{x}\tan\theta\in[-(j+1)\epsilon,-j\epsilon)\}\right] \le$$

$$e^{-(k_1\epsilon)^2} + \epsilon\underbrace{\sum_{j=0}^{\infty}\mathop{\mathbb{P}}_{\mathbf{x}\sim\mathcal{N}(0,I_d)}\left[\mathbf{v}_\perp\cdot\mathbf{x}\tan\theta\in[-(j+1)\epsilon,-j\epsilon)\right]}_{\le 1} \le 2\epsilon \quad \text{(E.13)}$$

Let $\theta$ denote the angle $\measuredangle(\mathbf{v},\mathbf{v}')$. Given the inequality above, in order to finish the proof of Claim 4, it remains to upper-bound $\mathbb{E}_{\mathbf{x}\sim\mathcal{N}(0,I_d)}\left[\varphi_{\mathrm{up}}(\mathbf{x})-\psi_{\mathrm{up}}(\mathbf{x})\right]$ and $\mathbb{E}_{\mathbf{x}\sim\mathcal{N}(0,I_d)}\left[\varphi_{\mathrm{up}}(\mathbf{x})-\psi_{\mathrm{up}}(\mathbf{x})\right]$ by $O(\theta)+4\epsilon$.

From Corollary E.7 we know that for any $t$ we have:

$$\mathop{\mathbb{E}}_{z\sim\mathcal{N}(0,1)}\left[\left(J_{\mathrm{up}}^t(z)-J_{\mathrm{down}}^t(z)\right)^2\right] \le O\left(\min\left(\frac{\log^{10}k}{\sqrt{k}},\frac{1}{t^2}\right)\right), \quad \text{(E.14)}$$

and for every $z$ in $\mathbb{R}$

$$J_{\mathrm{down}}^t(z) \le \mathbb{1}_{z\le t} \le J_{\mathrm{up}}^t(z). \quad \text{(E.15)}$$

Since $\mathbf{v}$ and $\mathbf{v}_\perp$ are orthogonal, the random variables $\mathbf{v}_\perp\mathbf{x}$ and $\mathbf{v}\cdot\mathbf{x}$ are independent standard Gaussians. Using this, together with Equations E.14 and E.15 we get:

$$\mathop{\mathbb{E}}_{\mathbf{x}\sim\mathcal{N}(0,I_d)}\left[\left(\varphi_{\mathrm{up}}^{L_2}(\mathbf{x})-\psi_{\mathrm{up}}(\mathbf{x})\right)^2\right] = \sum_{j=0}^{k_1-1}\mathop{\mathbb{E}}_{\mathbf{x}\sim\mathcal{N}(0,I_d)}\left[\mathbb{1}_{\mathbf{v}\cdot\mathbf{x}\in[j\epsilon,(j+1)\epsilon)}\left(\varphi_{\mathrm{up}}^{L_2}(\mathbf{x})-\psi_{\mathrm{up}}^{L_2}(\mathbf{x})\right)^2\right]$$

$$\le \sum_{j=0}^{k_1-1}\mathop{\mathbb{P}}_{\mathbf{x}\sim\mathcal{N}(0,I_d)}\left[\mathbf{v}\cdot\mathbf{x}\in[j\epsilon,(j+1)\epsilon)\right]\mathop{\mathbb{E}}_{\mathbf{x}\sim\mathcal{N}(0,I_d)}\left[\left(J_{\mathrm{up}}^{-j\epsilon/\tan\theta}(\mathbf{v}_\perp\cdot\mathbf{x})-\mathbb{1}_{\mathbf{v}_\perp\cdot\mathbf{x}\le-j\epsilon/\tan\theta}\right)^2\right] =$$

$$\sum_{j=0}^{k_1-1}\mathop{\mathbb{P}}_{z_1\sim\mathcal{N}(0,1)}\left[z_1\in[j\epsilon,(j+1)\epsilon)\right]\mathop{\mathbb{E}}_{z_2\sim\mathcal{N}(0,1)}\left[\left(J_{\mathrm{up}}^{-j\epsilon/\tan\theta}(z_2)-J_{\mathrm{down}}^{-j\epsilon/\tan\theta}(z_2)\right)^2\right] \le$$

$$\sum_{j=0}^{k_1-1}\mathop{\mathbb{P}}_{z_1\sim\mathcal{N}(0,1)}\left[z_1\in[j\epsilon,(j+1)\epsilon)\right]O\left(\min\left(\frac{\log^{10}k_2}{\sqrt{k_2}},\left(\frac{\tan\theta}{j\epsilon}\right)^2\right)\right)$$

First, consider the case $\theta\ge\pi/4$. The above inequality implies

$$\mathop{\mathbb{E}}_{\mathbf{x}\sim\mathcal{N}(0,I_d)}\left[\left(\varphi_{\mathrm{up}}^{L_2}(\mathbf{x})-\psi_{\mathrm{up}}(\mathbf{x})\right)^2\right] \le O\left(\frac{\log^{10}k}{\sqrt{k}}\right)\underbrace{\sum_{j=0}^{k_1-1}\mathop{\mathbb{P}}_{z_1\sim\mathcal{N}(0,1)}\left[z_1\in[j\epsilon,(j+1)\epsilon)\right]}_{=\mathbb{P}_{z_1\sim\mathcal{N}(0,1)}[0\le z_1<(k_1)\epsilon]} =$$

$$O\left(\frac{\log^{10}k_2}{\sqrt{k_2}}\right) = O\left(\frac{\theta\log^5 k_2}{k_2^{1/4}}\right).$$

On the other hand, if $\theta \le \pi/4$ we have $\tan \theta \le 2\theta$ and therefore, recalling that for any $j$ it is the case that $\mathbb{P}_{z_1 \sim \mathcal{N}(0,1)}\left[z_1 \in [j\epsilon, (j+1)\epsilon)\right] \le \epsilon$, we have

$$
\begin{aligned}
\mathop{\mathbb{E}}_{\mathbf{x} \sim \mathcal{N}(0,I_d)}\left[\left(\varphi_{\text{up}}^{L_2}(\mathbf{x}) - \psi_{\text{up}}(\mathbf{x})\right)^2\right] &\le \sum_{j=0}^{k_1-1} O\left(\min\left(\frac{\log^{10} k_2}{\sqrt{k_2}}, \left(\frac{\tan\theta}{j\epsilon}\right)^2\right)\epsilon\right) \\
&\le \int_0^{+\infty} O\left(\min\left(\frac{\log^{10} k_2}{\sqrt{k_2}}, \left(\frac{\tan\theta}{z-\epsilon}\right)^2\right)\right) dz \\
&\le O\left(\frac{\log^{10} k_2}{\sqrt{k_2}}\epsilon\right) + \int_0^{+\infty} O\left(\min\left(\frac{\log^{10} k_2}{\sqrt{k_2}}, \left(\frac{\tan\theta}{z}\right)^2\right)\right) dz \\
&= O\left(\frac{\log^{10} k_2}{\sqrt{k_2}}\epsilon\right) + \tan\theta \int_0^{+\infty} O\left(\min\left(\frac{\log^{10} k_2}{\sqrt{k_2}}, \left(\frac{1}{z}\right)^2\right)\right) dz \\
&= O\left(\frac{\log^{10} k_2}{\sqrt{k_2}}\epsilon + \tan\theta\left(\frac{\log^{10} k_2}{\sqrt{k_2}}\left(\frac{\sqrt{k_2}}{\log^{10} k_2}\right)^{0.5} + \left(\frac{\log^{10} k_2}{\sqrt{k_2}}\right)^{0.5}\right)\right) \\
&= O\left(\frac{\theta \log^5 k_2}{k_2^{1/4}}\right) \tag{E.16}
\end{aligned}
$$

Overall, in either case we have $\mathbb{E}_{\mathbf{x} \sim \mathcal{N}(0,I_d)}\left[\left(\varphi_{\text{up}}^{L_2}(\mathbf{x}) - \psi_{\text{up}}(\mathbf{x})\right)^2\right] = O\left(\frac{\theta \log^{1.5} k_2}{k_2^{1/4}}\right)$. We now go through a fully analogous argument to show that also $\mathbb{E}_{\mathbf{x} \sim \mathcal{N}(0,I_d)}\left[\left(\psi_{\text{down}}(\mathbf{x}) - \varphi_{\text{down}}^{L_2}(\mathbf{x})\right)^2\right] = O\left(\frac{\theta \log^{1.5} k_2}{k_2^{1/4}}\right)$. Again, from the independence of $\mathbf{v}_\perp \mathbf{x}$ and $\mathbf{v} \cdot \mathbf{x}$, together with Equations E.14 and E.15 we have:

$$
\begin{aligned}
&\mathop{\mathbb{E}}_{\mathbf{x} \sim \mathcal{N}(0,I_d)}\left[\left(\psi_{\text{down}}(\mathbf{x}) - \varphi_{\text{down}}^{L_2}(\mathbf{x})\right)^2\right] = \\
&= \sum_{j=0}^{k_1-1} \mathop{\mathbb{P}}_{\mathbf{x} \sim \mathcal{N}(0,I_d)}\left[\mathbf{v} \cdot \mathbf{x} \in [j\epsilon, (j+1)\epsilon)\right] \mathop{\mathbb{E}}_{\mathbf{x} \sim \mathcal{N}(0,I_d)}\left[\left(\mathbb{1}_{\mathbf{v}_\perp \cdot \mathbf{x} \le -(j+1)\epsilon/\tan\theta} - J_{\text{down}}^{-(j+1)\epsilon/\tan\theta}(\mathbf{v}_\perp \cdot \mathbf{x})\right)^2\right] \le \\
&\sum_{j=0}^{k_1-1} \mathop{\mathbb{P}}_{z_1 \sim \mathcal{N}(0,1)}\left[z_1 \in [j\epsilon, (j+1)\epsilon)\right] \mathop{\mathbb{E}}_{z_2 \sim \mathcal{N}(0,1)}\left[\left(R_{\text{up}}^{-(j+1)\epsilon/\tan\theta}(z_2) - R_{\text{down}}^{-(j+1)\epsilon/\tan\theta}(z_2)\right)^2\right] \le \\
&\sum_{j=0}^{k_1-1} \mathop{\mathbb{P}}_{z_1 \sim \mathcal{N}(0,1)}\left[z_1 \in [j\epsilon, (j+1)\epsilon)\right] O\left(\min\left(\frac{\log^{10} k_2}{\sqrt{k_2}}, \left(\frac{\tan\theta}{(j+1)\epsilon}\right)^2\right)\right) \le \\
&\sum_{j=0}^{k_1-1} \mathop{\mathbb{P}}_{z_1 \sim \mathcal{N}(0,1)}\left[z_1 \in [j\epsilon, (j+1)\epsilon)\right] O\left(\min\left(\frac{\log^{10} k_2}{\sqrt{k_2}}, \left(\frac{\tan\theta}{j\epsilon}\right)^2\right)\right)
\end{aligned}
$$

As it was shown previously, the expression above is at most $O\left(\frac{\theta \log^5 k_2}{k_2^{1/4}}\right)$.

In total, combining our bounds on $\mathbb{E}_{\mathbf{x} \sim \mathcal{N}(0,I_d)}[(\psi_{\text{down}}(\mathbf{x}) - \varphi_{\text{down}}^{L_2}(\mathbf{x}))^2]$, $\mathbb{E}_{\mathbf{x} \sim \mathcal{N}(0,I_d)}[(\varphi_{\text{up}}^{L_2}(\mathbf{x}) - \psi_{\text{up}}(\mathbf{x}))^2]$ and $\mathbb{E}_{\mathbf{x} \sim \mathcal{N}(0,I_d)}[(\psi_{\text{up}}(\mathbf{x}) - \psi_{\text{down}}(\mathbf{x}))^2]$ we conclude that

$$
\mathop{\mathbb{E}}_{\mathbf{x} \sim \mathcal{N}(0,I_d)}\left[\left(\varphi_{\text{up}}^{L_2}(\mathbf{x}) - \varphi_{\text{down}}^{L_2}(\mathbf{x})\right)^2\right] \le O\left(\frac{\log^5 k_2}{k_2^{1/4}} \cdot \angle(\mathbf{v}, \mathbf{v}')\right) + 10\epsilon.
$$

$\square$

It only remains to prove Claim 5.

*Proof of Claim 5.* Corollary E.5 says that for any value of $t$, the degree-$k_2$ one-dimensional polynomials $J_{\text{up}}^t(z)$ and $J_{\text{down}}^t(z)$ have all their coefficients bounded by $O\left(2^{10k_2}\right)$. If one substitutes $\mathbf{v} \cdot \mathbf{x}$

in place of $z$ into either of these polynomials and opens the parentheses, the fact that $\mathbf{v}$ is a unit vector allows us to bound the size of the larges coefficients of $R_{\text{down}}^t(\mathbf{v}_\perp \cdot \mathbf{x})$ and $R_{\text{up}}^t(\mathbf{v}_\perp \cdot \mathbf{x})$ by $O\left(d^{k_2}(k_2+1)2^{10k_2}\right) = O(d^{10k_2})$, proving the claim. $\qquad\square$

### E.4 Miscellaneous Claims

**Claim 6.** *There is a deterministic algorithm that given a unit vector $\mathbf{v}$ in $\mathbb{R}^d$, scalars $a$ and $b$, a monomial $m$ over $\mathbb{R}^d$ of degree at most $k_2$, an accuracy parameter $\beta \in (0,1]$, runs in time* $\text{poly}\left((k_2 d)^{k_2}/\beta\right)$ *and computes a $\binom{d+1}{k_2} \times \binom{d+1}{k_2}$-matrix $W^{a,b}$ such that*

$$W^{a,b} - \beta I_{\binom{d+1}{k_2} \times \binom{d+1}{k_2}} \preceq \mathop{\mathbb{E}}_{\mathbf{x} \sim \mathcal{N}(0, I_d)} \left[ \left(\mathbf{x}^{\otimes k_2}\right)\left(\mathbf{x}^{\otimes k_2}\right)^\top \cdot \mathbb{1}_{a \leq \mathbf{x} \cdot \mathbf{v} < b} \right] \preceq W^{a,b} + \beta I_{\binom{d+1}{k_2} \times \binom{d+1}{k_2}} \tag{E.17}$$

*Proof.* From Section 3, we recall that $\mathbf{x}^{\otimes k_2}$ is the vector one gets by evaluating all multidimensional monomials of degree at most $k_2$ on input $\mathbf{x}$, and therefore the $\binom{d+1}{k_2} \times \binom{d+1}{k_2}$-matrix $(\mathbf{x}^{\otimes k_2})(\mathbf{x}^{\otimes k_2})^\top$, viewed as a bilinear form, for degree-$k$ polynomials $p_1$ and $p_2$ we have $p_1^\top (\mathbf{x}^{\otimes k_2})(\mathbf{x}^{\otimes k_2})^\top p_2 = p_1(\mathbf{x})p_2(\mathbf{x})$. Thus, the entries of $(\mathbf{x}^{\otimes k_2})(\mathbf{x}^{\otimes k_2})^\top$ are indexed by pairs of monomials $m_1$ and $m_2$ over $\mathbb{R}^d$ of degree at most $k$, and we have $m_1^\top (\mathbf{x}^{\otimes k_2})(\mathbf{x}^{\otimes k_2})^\top m_2 = m_1(\mathbf{x})m_2(\mathbf{x})$. And the entries of $\mathbb{E}_{\mathbf{x} \sim \mathcal{N}}[(\mathbf{x}^{\otimes k_2})(\mathbf{x}^{\otimes k_2})^\top \cdot \mathbb{1}_{a \leq \mathbf{x} \cdot \mathbf{v} < b}]$ are also indexed by pairs of monomials $m_1$ and $m_2$ over $\mathbb{R}^d$ of degree at most $k$ and equal to
$\mathbb{E}_{\mathbf{x} \sim \mathcal{N}}[m_1(\mathbf{x})m_2(\mathbf{x}) \cdot \mathbb{1}_{a \leq \mathbf{x} \cdot \mathbf{v} < b}]$. Since the product $m_1 m_2$ is a monomial of degree at most $2k_2$, Claim 3 tells us that this value can be approximated up to error $\beta'$ in time $\text{poly}((k_2 d)^{k_2}/\beta')$ (we will set the value of $\beta'$ later).

Thus, we take the $\binom{d+1}{k_2} \times \binom{d+1}{k_2}$-matrix $W^{a,b}$ to have entries pairs of monomials $m_1$ and $m_2$ over $\mathbb{R}^d$ of degree at most $k$ and equal to additive $\beta'$-approximations to $\mathbb{E}_{\mathbf{x} \sim \mathcal{N}}[m_1(\mathbf{x})m_2(\mathbf{x}) \cdot \mathbb{1}_{a \leq \mathbf{x} \cdot \mathbf{v} < b}]$. Thus, the difference between $\mathbb{E}_{\mathbf{x} \sim \mathcal{N}}[(\mathbf{x}^{\otimes k_2})(\mathbf{x}^{\otimes k_2})^\top \cdot \mathbb{1}_{a \leq \mathbf{x} \cdot \mathbf{v} < b}]$ and $W^{a,b}$ is a $\binom{d+1}{k_2} \times \binom{d+1}{k_2}$-matrix whose entries are bounded by $\beta'$ in absolute value. Thus, the Frobenius norm of this difference matrix is at most $(d+1)^{k_2/2}\beta'$, and therefore all eigenvalues of the matrix $\mathbb{E}_{\mathbf{x} \sim \mathcal{N}}[m_1(\mathbf{x})m_2(\mathbf{x}) \cdot \mathbb{1}_{a \leq \mathbf{x} \cdot \mathbf{v} < b}] - W^{a,b}$ are in $[-(d+1)^{k_2/2}\beta', (d+1)^{k_2/2}\beta']$. Taking $\beta' = \beta(d+1)^{-k_2/2}$, we can conclude that Equation E.17 holds.

Overall, the run-time is $\text{poly}((k_2 d)^{k_2}/\beta')$, which we see equals to $\text{poly}((k_2 d)^{k_2}/\beta)$ since we have $\beta' = \beta(d+1)^{-k_2/2}$. This completes the proof. $\qquad\square$

## F  Testing Massart Noise

We give here the full proof of our main theorem (Theorem B.1), which we restate for convenience.

**Theorem F.1.** *There exists a deterministic algorithm $\mathcal{A}_{\text{Massart}}$ that runs in time $\text{poly}\left(\frac{Nd}{\epsilon\delta}\right)$ and for a sufficiently large absolute constant $C$ satisfies the following. Given parameters $\epsilon, \delta$ in $(0,1)$ and a dataset $\bar{S}$ of size $N \geq \left(\frac{Cd}{\epsilon\delta}\right)^C$ consisting of elements in $\mathbb{R}^d \times \{\pm 1\}$, the algorithm $\mathcal{A}_{\text{Massart}}$ outputs either $(\mathsf{Accept}, \mathbf{v})$ for some unit vector $\mathbf{v}$ in $\mathbb{R}^d$, or outputs $\mathsf{Reject}$ (in the former case we say $\mathcal{A}$ accepts, while in the latter case we say $\mathcal{A}$ rejects). The algorithm $\mathcal{A}_{\text{Massart}}$ satisfies the following conditions:*

1. **Completeness:** *The algorithm $\mathcal{A}_{\text{Massart}}$ accepts with probability at least $1 - O(\delta)$ if $\bar{S}$ is generated by $\mathsf{EX}_{\mathcal{N}, f, \eta_0}^{\text{Massart}}$ where $f$ is an origin-centered halfspace and $\eta_0 \leq 1/3$.*

2. **Soundness:** *For any dataset $\bar{S}$ of size $N \geq \left(\frac{Cd}{\epsilon\delta}\right)^C$, if $\mathcal{A}_{\text{Massart}}$ accepts then the vector $\mathbf{v}$ given by $\mathcal{A}_{\text{Massart}}$ satisfies*

$$\mathop{\mathbb{P}}_{(\mathbf{x},y) \sim \bar{S}} [\mathsf{sign}(\mathbf{x} \cdot \mathbf{v}) \neq y] \leq \mathsf{opt} + O(\epsilon), \tag{F.1}$$

   *where opt is defined to be $\min_{\mathbf{v}' \in \mathbb{R}^d} \left( \mathbb{P}_{(\mathbf{x},y) \sim \bar{S}} [\mathsf{sign}(\mathbf{x} \cdot \mathbf{v}') \neq y] \right)$.*

The rest of this section proves the above theorem. The algorithm $\mathcal{A}_{\text{Massart}}$ does the following (where $C$ is a sufficiently large absolute constant):

- **Given:** parameters $\epsilon, \delta$ in $(0, 1)$ and a dataset $\bar{S}$ of size $N \geq \left(\frac{Cd}{\epsilon\delta}\right)^C$ consisting of elements in $\mathbb{R}^d \times \{\pm 1\}$.

  1. Let $\mathbf{v}$ be the output of the algorithm of Fact B.2 run on the dataset $\bar{S}$ with accuracy parameter $\epsilon' = \frac{\epsilon^{3/2}}{100\sqrt{d-1}}$ and failure probability $\delta$. Without loss of generality we can assume that the algorithm is deterministic, because we can use some of the points in $\bar{S}$ for random seeds.
  2. $S \leftarrow \{\mathbf{x} : (\mathbf{x}, y) \in \bar{S}\}$ and $N \leftarrow |\bar{S}|$.
  3. Run the tester $\mathcal{T}_{\text{disagreement}}$ from Theorem D.1, on input $(S, \mathbf{v}, \epsilon, \delta, 0.1)$.
  4. If $\mathcal{T}_{\text{disagreement}}$ rejects in the previous step, output output Reject.
  5. If $|S_{\text{False}}| > \frac{2}{5}N$, then output Reject.
  6. $S_{\text{False}}^{\text{far}} \leftarrow S_{\text{False}} \cap \left\{\mathbf{x} \in \mathbb{R}^d : \left|\measuredangle(\mathbf{x}, \mathbf{v}) - \frac{\pi}{2}\right| > \frac{\epsilon^{3/2}}{\sqrt{d-1}}\right\}$; $S_{\text{False}}^{\text{near}} \leftarrow S_{\text{False}} \setminus S_{\text{False}}^{\text{far}}$.
  7. If $S_{\text{False}}^{\text{near}} > 4\epsilon N$, then output Reject.
  8. Take $U = \frac{2}{5}N$ and then run the spectral tester $\mathcal{T}_{\text{spectral}}$ from Theorem E.1 with the input parameters $(U, S_{\text{False}}^{\text{far}}, \mathbf{v}, \epsilon, \delta, 0.1)$.
  9. If $\mathcal{T}_{\text{spectral}}$ rejects in the previous step, output output Reject.
  10. Otherwise, output (Accept, $\mathbf{v}$).

From the run-time guarantees given in Theorem D.1 and Theorem E.1, we see immideately that the run-time of the algorithm $\mathcal{A}_{\text{Massart}}$ is $\text{poly}\left(\frac{d}{\epsilon}\log\frac{1}{\delta}\right)$.

## F.1 Soundness

We first show the soundness condition. For any dataset $\bar{S}$ of size $N \geq \left(\frac{Cd}{\epsilon\delta}\right)^C$, we need to show that if $\mathcal{A}_{\text{Massart}}$ accepts then the vector $\mathbf{v}$ given by $\mathcal{A}_{\text{Massart}}$ satisfies Equation F.1. Theorems D.1 and E.1 imply that if the algorithm $\mathcal{A}_{\text{Massart}}$ accepts then

$$\mathbb{P}_{(\mathbf{x},y)\sim\bar{S}}[\text{sign}(\mathbf{x}\cdot\mathbf{v}) \neq \text{sign}(\mathbf{x}\cdot\mathbf{v}')] = (1 \pm 0.1)\frac{\measuredangle(\mathbf{v},\mathbf{v}')}{\pi} \pm O(\epsilon). \tag{F.2}$$

$$\frac{1}{U}\sum_{\mathbf{x}\in S_{\text{False}}^{\text{far}}}\left[\mathbb{1}_{\text{sign}(\mathbf{x}\cdot\mathbf{v})\neq\text{sign}(\mathbf{x}\cdot\mathbf{v}')}\right] \leq 1.1\frac{\measuredangle(\mathbf{v},\mathbf{v}')}{\pi} + O(\epsilon). \tag{F.3}$$

Rearranging, we get

$$\mathbb{P}_{(\mathbf{x},y)\sim\bar{S}}[\text{sign}(\mathbf{x}\cdot\mathbf{v}') \neq y] - \mathbb{P}_{(\mathbf{x},y)\sim\bar{S}}[y \neq \text{sign}(\mathbf{v}\cdot\mathbf{x})] =$$

$$\mathbb{P}_{(\mathbf{x},y)\sim\bar{S}}[\text{sign}(\mathbf{x}\cdot\mathbf{v}) \neq \text{sign}(\mathbf{x}\cdot\mathbf{v}') \wedge y = \text{sign}(\mathbf{v}\cdot\mathbf{x})] -$$

$$- \mathbb{P}_{(\mathbf{x},y)\sim\bar{S}}[\text{sign}(\mathbf{x}\cdot\mathbf{v}) \neq \text{sign}(\mathbf{x}\cdot\mathbf{v}') \wedge y \neq \text{sign}(\mathbf{v}\cdot\mathbf{x})] =$$

$$\mathbb{P}_{(\mathbf{x},y)\sim\bar{S}}[\text{sign}(\mathbf{x}\cdot\mathbf{v}) \neq \text{sign}(\mathbf{x}\cdot\mathbf{v}')] - 2\mathbb{P}_{(\mathbf{x},y)\sim\bar{S}}[\text{sign}(\mathbf{x}\cdot\mathbf{v}) \neq \text{sign}(\mathbf{x}\cdot\mathbf{v}') \wedge y \neq \text{sign}(\mathbf{v}\cdot\mathbf{x})] =$$

$$\mathbb{P}_{\mathbf{x}\sim S}[\text{sign}(\mathbf{x}\cdot\mathbf{v}) \neq \text{sign}(\mathbf{x}\cdot\mathbf{v}')] - \frac{2}{N}\sum_{\mathbf{x}\in S_{\text{False}}}\left[\mathbb{1}_{\text{sign}(\mathbf{x}\cdot\mathbf{v})\neq\text{sign}(\mathbf{x}\cdot\mathbf{v}')}\right]$$

By Equation F.2, the first term above is lower-bounded by $0.9\frac{\measuredangle(\mathbf{v},\mathbf{v}')}{\pi} - O(\epsilon)$. The second term can broken into two components: $\frac{2}{N}\sum_{\mathbf{x}\in S_{\text{False}}^{\text{near}}}\left[\mathbb{1}_{\text{sign}(\mathbf{x}\cdot\mathbf{v})\neq\text{sign}(\mathbf{x}\cdot\mathbf{v}')}\right]$ and $\frac{2}{N}\sum_{\mathbf{x}\in S_{\text{False}}^{\text{far}}}\left[\mathbb{1}_{\text{sign}(\mathbf{x}\cdot\mathbf{v})\neq\text{sign}(\mathbf{x}\cdot\mathbf{v}')}\right]$. If the algorithm does not reject in step (8), the former term is upper-bounded by $O(\epsilon)$, while Equation F.2 tells us that the latter term is upper-bounded by $\frac{2U}{N}\left(1.1\frac{\measuredangle(\mathbf{v},\mathbf{v}')}{\pi} + O(\epsilon)\right)$. Overall, substituting

these bounds and recalling that $U/N = 2/5$, we get

$$\mathbb{P}_{(\mathbf{x},y)\sim\bar{S}}\left[\mathsf{sign}(\mathbf{x}\cdot\mathbf{v}') \neq y\right] - \mathbb{P}_{(\mathbf{x},y)\sim\bar{S}}\left[y \neq \mathsf{sign}(\mathbf{v}\cdot\mathbf{x})\right] \geq$$

$$0.9\frac{\measuredangle(\mathbf{v},\mathbf{v}')}{\pi} - O(\epsilon) - \frac{2U}{N}\left(1.1\frac{\measuredangle(\mathbf{v},\mathbf{v}')}{\pi} + O(\epsilon)\right) =$$

$$0.9\frac{\measuredangle(\mathbf{v},\mathbf{v}')}{\pi} - \frac{4}{5}\left(1.1\frac{\measuredangle(\mathbf{v},\mathbf{v}')}{\pi}\right) - O(\epsilon) = 0.02\frac{\measuredangle(\mathbf{v},\mathbf{v}')}{\pi} - O(\epsilon) \geq -O(\epsilon).$$

Thus, choosing $\mathbf{v}'$ to be $\arg\min_{\mathbf{u}}\left(\mathbb{P}_{(\mathbf{x},y)\sim\mathcal{D}_{\mathrm{pairs}}}\left[\mathsf{sign}(\mathbf{x}\cdot\mathbf{u}) \neq y\right]\right)$, we get

$$\mathbb{P}_{(\mathbf{x},y)\sim\bar{S}}\left[\mathsf{sign}(\mathbf{x}\cdot\mathbf{v}) \neq y\right] \leq \mathsf{opt} + O(\epsilon),$$

finishing the proof of soundess.

## F.2 Completeness

We now argue that for a sufficiently large absolute constant $C$, the algorithm $\mathcal{A}_{\mathrm{Massart}}$ satisfies the completeness condition. In this subsection we assume that $\bar{S}$ is generated by $\mathsf{EX}^{\mathsf{Massart}}_{\mathcal{N},f,\eta_0}$ where $f(\mathbf{x}) = \mathsf{sign}(\mathbf{v}^*\cdot\mathbf{x})$ is an origin-centered halfspace and $\eta_0 \leq 1/3$. We remind the reader that, for some function $\eta : \mathbb{R}^d \to [0,\eta_0]$, every time $\mathsf{EX}^{\mathsf{Massart}}_{\mathcal{N},f,\eta_0}$ is invoked it generates an i.i.d. pair $(\mathbf{x},y) \in \mathbb{R}^d \times \{\pm1\}$ where $\mathbf{x}$ is drawn from $\mathcal{N}(0,I_d)$ and $y = f(\mathbf{x})$ with probability $\eta(\mathbf{x})$ and $-f(\mathbf{x})$ with probability $1 - \eta(\mathbf{x})$. We would like to show that $\mathcal{A}_{\mathrm{Massart}}$ accepts with probability at least $1 - O(\delta)$.

For the purposes of completeness analysis, we define the set $S_{\mathrm{augmented}}$ to be a set of points in $\mathbb{R}^d$ generated through the following random process:

- If a datapoint $\mathbf{x}$ in $S$ has label $y = -f(\mathbf{x})$, then $\mathbf{x}$ in included into $S_{\mathrm{augmented}}$.
- If a datapoint $\mathbf{x}$ in $S$ has label $y = f(\mathbf{x})$, then $\mathbf{x}$ in included into $S_{\mathrm{augmented}}$ with probability $\frac{\eta_0 - \eta(\mathbf{x})}{1 - \eta(\mathbf{x})}$ (and this choice is made independently for different $\mathbf{x}$ in $S$).

With the definition above in hand, we claim the following:

**Claim 7.** *If the absolute constant $C$ is large enough, then with probability at least $1 - \delta$ it is the case that $|S_{\mathrm{augmented}}| \leq \frac{2}{5}N$. Furthermore, conditioned on any particular value of the size $|S_{\mathrm{augmented}}|$ of this set, the individual elements of $S_{\mathrm{augmented}}$ are distributed i.i.d. from the standard Gaussian distribution $\mathcal{N}(0,I_d)$.*

*Proof.* Overall, we know that $y = -f(\mathbf{x})$ with probability $\eta(\mathbf{x})$, so overall each $\mathbf{x}$ in $S$ gets included into $S_{\mathrm{augmented}}$ independently with probability $\eta(\mathbf{x}) + (1 - \eta(\mathbf{x}))\frac{\eta_0 - \eta(\mathbf{x})}{1 - \eta(\mathbf{x})} = \eta_0$. Overall every element $S$ is included into $S_{\mathrm{augmented}}$ with independently probability $\eta_0$. Since $\eta_0$ is at most $1/3$ and $N \geq \left(\frac{Cd}{\epsilon\delta}\right)^C$, we see that the standard Hoeffding bound tells us that for a sufficiently large absolute constant $C$ with probability at least $1 - \delta$ it is the case that $|S_{\mathrm{augmented}}| \leq \frac{2}{5}|S| = \frac{2}{5}N$. This proves the first part of the claim.

Additionally, recall that in this subsection we are assuming that $\bar{S}$ is generated by $\mathsf{EX}^{\mathsf{Massart}}_{\mathcal{N},f,\eta_0}$. This implies that the elements of $S$ are generated i.i.d. from $\mathcal{N}(0,I_d)$. Since the decision wheather each datapoint $\mathbf{x}$ in $S$ is included into $S_{\mathrm{augmented}}$ is made with probability $\eta_0$ independently from the actual value of $\mathbf{x}$, this implies the element of $S_{\mathrm{augmented}}$ are distributed i.i.d. as $\mathcal{N}(0,I_d)$ even conditioned on any specific value of $|S_{\mathrm{augmented}}|$. This finishes the proof of the claim. $\square$

The following claim lists a number of desirable events for algorithm $\mathcal{A}_{\mathrm{Massart}}$ and shows that they are likely to hold.

**Claim 8.** *If $C$ is a sufficiently large absolute constant, the following events take place with probability at least $1 - O(\delta)$:*

1. *The set $S$ is such that for all unit vectors $\mathbf{v}'$ the algorithm $\mathcal{T}_{disagreement}$ accepts when given the input $(S, \mathbf{v}', \epsilon, \delta, 0.1)$.*

2. *For all vectors $\mathbf{u}$ in $\mathbb{R}^d$, we have*

$$\left| \Pr_{(\mathbf{x},y) \sim \bar{S}} [\mathsf{sign}(\mathbf{x} \cdot \mathbf{u}) \neq y] - \Pr_{(\mathbf{x},y) \sim \mathsf{EX}^{\mathsf{Massart}}_{\mathcal{N}, f, \eta_0}} [\mathsf{sign}(\mathbf{x} \cdot \mathbf{v}) \neq y] \right| \leq 2d \sqrt{\frac{\log N}{N}} \log \frac{1}{\delta}.$$

3. *For all vectors $\mathbf{u}$ in $\mathbb{R}^d$ and scalars $\theta$, we have*

$$\left| \Pr_{\mathbf{x} \sim S} [\measuredangle(\mathbf{x}, \mathbf{u}) \leq \theta] - \Pr_{\mathbf{x} \sim \mathcal{N}(0, I_d)} [\measuredangle(\mathbf{x}, \mathbf{u}) \leq \theta] \right| \leq 2d \sqrt{\frac{\log N}{N}} \log \frac{1}{\delta}.$$

4. *It is the case that $\measuredangle(\mathbf{v}, \mathbf{v}^*) \leq \frac{\epsilon^{3/2}}{10\sqrt{d-1}}$.*

5. *It is the case that $|S_{\mathsf{False}}| \leq \frac{2}{5} N$ and $|S_{augmented}| \leq \frac{2}{5} N$.*

6. *$S_{augmented}$ is such that for all unit vectors $\mathbf{v}'$ the algorithm $\mathcal{T}_{spectral}$ accepts when given as input on the input $(U, S_{augmented}, \mathbf{v}', \epsilon, \delta, 0.1)$ (we remind the reader that $U = \frac{2}{5} N$).*

*Proof.* Event 1 holds with probability at least $1 - O(\delta)$ by Theorem D.1. The Event (2) holds with probability at least $1 - O(\delta)$ by the standard VC bound, together with the fact that the VC dimension of the class of halfspaces in $\mathbb{R}^d$ is at most $d + 1$. Analogously, Event (2) holds with probability at least $1 - O(\delta)$ by the standard VC bound, together with the fact that the VC dimension of the class of origin-centric cones in $\mathbb{R}^d$ is most $O(d)$.

Recall that in step (1) of $A_{\mathsf{Massart}}$ we used the algorithm of [DKTZ20a] (see Fact B.2) which implies that with probability at least $1 - \delta$ we have $\measuredangle(\mathbf{v}, \mathbf{v}^*) \leq \frac{\epsilon^{3/2}}{10\sqrt{d-1}}$.

If Event (2) holds, we have

$$\Pr_{(\mathbf{x},y) \sim \bar{S}} [\mathsf{sign}(\mathbf{x} \cdot \mathbf{v}) \neq y] \leq \Pr_{(\mathbf{x},y) \sim \mathsf{EX}^{\mathsf{Massart}}_{\mathcal{N}, f, \eta_0}} [\mathsf{sign}(\mathbf{x} \cdot \mathbf{v}) \neq y] + 2d \sqrt{\frac{\log N}{N}} \log \frac{1}{\delta},$$

and if Equation D.1 also holds we have

$$\frac{|S_{\mathsf{False}}|}{N} = \Pr_{(\mathbf{x},y) \sim \bar{S}} [\mathsf{sign}(\mathbf{x} \cdot \mathbf{v}) \neq y] \leq$$

$$\Pr_{(\mathbf{x},y) \sim \mathsf{EX}^{\mathsf{Massart}}_{\mathcal{N}, f, \eta_0}} [\mathsf{sign}(\mathbf{x} \cdot \mathbf{v}^*) \neq y] + \frac{\epsilon^{3/2}}{100\sqrt{d-1}} + 2d \sqrt{\frac{\log N}{N}} \log \frac{1}{\delta} \leq$$

$$\eta_0 + \frac{1}{100} + 2d \sqrt{\frac{\log N}{N}} \log \frac{1}{\delta} \leq \frac{1}{3} + \frac{1}{100} + 2d \sqrt{\frac{\log N}{N}} \log \frac{1}{\delta}.$$

Substituting $N \geq \left( \frac{Cd}{\epsilon\delta} \right)^C$, we see that the above is at most $\frac{2}{5}$ if $C$ is a sufficiently large absolute constant. Thus, with probability at least $1 - O(\delta)$ we have $|S_{\mathsf{False}}| \leq \frac{2N}{5}$. At the same time, Claim D.1 tells us that with probability at least $1 - \delta$ we have $|S_{augmented}| \leq U = \frac{2N}{5}$. Overall, we see that Event (5) holds with probability at least $1 - O(\delta)$.

Finally, Claim E.1 tells us that with probability at least $1 - O(\delta)$ it is the case that $|S_{augmented}| \leq \frac{2}{5} N$. Furthermore, Claim E.1 also tells us that, even conditioned on this event, the set $S_{augmented}$ consists of i.i.d. samples from $\mathcal{N}(0, I_d)$. Then, the Completeness condition in Theorem E.1 tells us that with probability at least $1 - O(\delta)$ it is the case that $\mathcal{T}_{spectral}$ accepts if and is given $\mu = 0.1$, $U = \frac{2}{5} N$ and input dataset $S_{augmented}$. $\qquad\square$

Now, we first note that if Event 1 takes place, then $\mathcal{T}_{disagreement}$ accepts in step (3) of the algorithm $A_{\mathsf{Massart}}$.

If Event 3 in Claim 8 takes place, then from the triangle inequality it follows that

$$\left| \underset{\mathbf{x}\sim S}{\mathbb{P}}\left[\left|\measuredangle(\mathbf{x},\mathbf{v})-\frac{\pi}{2}\right|\le\frac{\epsilon^{3/2}}{\sqrt{d-1}}\right]-\underset{\mathbf{x}\sim\mathcal{N}(0,I_d)}{\mathbb{P}}\left[\left|\measuredangle(\mathbf{x},\mathbf{v})-\frac{\pi}{2}\right|\le\frac{\epsilon^{3/2}}{\sqrt{d-1}}\right]\right|\le 4d\sqrt{\frac{\log N}{N}}\log\frac{1}{\delta}. \tag{F.4}$$

It is also the case that

$$\underset{\mathbf{x}\sim\mathcal{N}(0,I_d)}{\mathbb{P}}\left[\left|\measuredangle(\mathbf{x},\mathbf{v})-\frac{\pi}{2}\right|\le\frac{\epsilon^{3/2}}{\sqrt{d-1}}\right]$$

$$\le\underset{\mathbf{x}\sim\mathcal{N}(0,I_d)}{\mathbb{P}}\left[|\mathbf{x}\cdot\mathbf{v}|\le\epsilon\right]+\underset{\mathbf{x}\sim\mathcal{N}(0,I_d)}{\mathbb{P}}\left[\|\mathbf{x}-\mathbf{v}\,(\mathbf{x}\cdot\mathbf{v})\|\tan\left(\frac{\epsilon^{3/2}}{\sqrt{d-1}}\right)\le\epsilon\right]$$

$$\le\underset{\mathbf{x}\sim\mathcal{N}(0,I_d)}{\mathbb{P}}\left[|\mathbf{x}\cdot\mathbf{v}|\le\epsilon\right]+\underset{\mathbf{x}\sim\mathcal{N}(0,I_d)}{\mathbb{P}}\left[\|\mathbf{x}-\mathbf{v}\,(\mathbf{x}\cdot\mathbf{v})\|\le\sqrt{\frac{d-1}{\epsilon}}\right]$$

$$=\underset{x\sim\mathcal{N}(0,1)}{\mathbb{P}}\left[|x|\le\epsilon\right]+\underset{\mathbf{x}\sim\mathcal{N}(0,I_{d-1})}{\mathbb{P}}\left[\|\mathbf{x}\|\le\sqrt{\frac{d-1}{\epsilon}}\right]\le 3\epsilon \tag{F.5}$$

Combining Equations F.4 and F.5 we get

$$\underset{\mathbf{x}\sim S}{\mathbb{P}}\left[\left|\measuredangle(\mathbf{x},\mathbf{v})-\frac{\pi}{2}\right|\le\frac{\epsilon^{3/2}}{\sqrt{d-1}}\right]\le 3\epsilon+4d\sqrt{\frac{\log N}{N}}\log\frac{1}{\delta}\le 4\epsilon,$$

where the last inequality holds if $C$ is a sufficiently large absolute constant. Since every element $\mathbf{x}$ in $S_{\text{False}}^{\text{near}}$ is in $S$ and also satisfies $\left|\measuredangle(\mathbf{x},\mathbf{v})-\frac{\pi}{2}\right|\le\frac{\epsilon^{3/2}}{10\sqrt{d}}$, we see that $S_{\text{False}}^{\text{near}}$ has a size of at most $4\epsilon N$ and therefore the algorithm $\mathcal{A}_{\text{Massart}}$ does not reject in step 8.

If Event 4 in Claim 8 takes place, then it is the case that $\measuredangle(\mathbf{v},\mathbf{v}^*)\le\frac{\epsilon^{3/2}}{2\sqrt{d-1}}$. If this is the case, the halfspaces $\text{sign}(\mathbf{v}\cdot\mathbf{x})$ and $\text{sign}(\mathbf{v}^*\cdot\mathbf{x})$ will agree for all vectors $\mathbf{x}$ satisfying $\left|\measuredangle(\mathbf{x},\mathbf{v})-\frac{\pi}{2}\right|>\frac{\epsilon^{3/2}}{\sqrt{d-1}}$, which holds for all $\mathbf{x}$ in $S_{\text{False}}^{\text{far}}$. Overall, for every $\mathbf{x}$ in $S_{\text{False}}^{\text{far}}$ the corresponding label $y$ satisfies $y\ne\text{sign}(\mathbf{v}\cdot\mathbf{x})=\text{sign}(\mathbf{v}^*\cdot\mathbf{x})$. Recalling the definition of the set $S_{\text{augmented}}$, we see that $S_{\text{False}}^{\text{far}}\subseteq S_{\text{augmented}}$. If Event 6 in Claim 8 holds then the set $S_{\text{augmented}}$ is such that if the algorithm $\mathcal{T}_{\text{spectral}}$ accepts when given as input $(U,S_{\text{augmented}},\mathbf{v},\epsilon,\delta,0.1)$. But Theorem E.1 shows that $\mathcal{T}_{\text{spectral}}$ satisfies Monotonicity under Datapoint Removal, which together with the inclusion $S_{\text{False}}^{\text{far}}\subseteq S_{\text{augmented}}$ implies that $\mathcal{T}_{\text{spectral}}$ accepts if it is given $\left(U,S_{\text{False}}^{\text{far}},\mathbf{v},\epsilon,\delta,0.1\right)$. Thus, the tester $\mathcal{T}_{\text{spectral}}$ does not reject in step 10.

We conclude that with probability at least $1-O(\delta)$ the algorithm $A_{\text{Massart}}$ will not reject in any of the four steps in which it could potentially reject. If this is the case, the algorithm $A_{\text{Massart}}$ will accept.

