# OpenReview forum: "Testing Noise Assumptions of Learning Algorithms"
_NeurIPS.cc/2025/Workshop/Reliable_ML — NeurIPS 2025 - Reliable ML Workshop_

### Official Review · Reviewer_KeiQ · 2025-09-15
**Novel and rigorous theoretic paper on testable learning**

**Rating:** 10
**Confidence:** 4

**Review:**

# Summary

The paper is on testable learning, where the goal is to design a tester and a learner for a specific problem such that (1) whenever the tester accepts, the learner outputs a near-optimal classifier, and (2) the tester must accept if the given dataset is drawn according to a specified modeling assumption. While the previous work only concerns assumptions on the covariate distribution (e.g., Gaussianity), this paper also addresses the question of testing the assumptions on the noise model of the label.

The paper considers the problem of learning halfspaces under Gaussian inputs with different noise assumptions. For Massart noise with noise rate bounded away from $1/2$, the paper shows that a polynomial-time tester-learner exists. And for Massart noise with noise rate $1/2$, the paper gives a cryptographic and a SQ lower bound of $d^{\Omega(1/ \varepsilon ^2)}$, from the lower bounds from the previous works in agnostic learning.

For the upper bound, the paper designed a black-box result that can certify the optimality of the classifier given by any learner that is guaranteed to output a near-optimal half-space under Massart noise, by decompose the error into several terms with bounds certifiable by some testers. The paper provides a disagreement tester and a spectral tester that improve the previous works, by making use of sandwiching approximators.

# Strengths

The paper is the first to ask the question of testing assumptions on the noise model, and gives optimal upper and lower bounds, in the sense that they match the bounds in the previous results. Meanwhile, the results show a separation between the classical learning and testable learning when the Massart noise rate is $1/2$, which imply a sharper transition for testable learning when the noise rate is around $1/2$, compared to classical learning.

Also, the design of the tester is novel. Instead of testing specific properties of a certain learner or input distribution, the tester here works for any near-optimal learner. In this case, the learner is executed first, before the tester, which is interesting.

The writing is clear and easy to follow. The description of the techniques is intuitive. The paper also provides the proof for the random classification noise model, which is a simplification of the Massart noise model. This makes the first-time readers easy to follow the arguments.

# Weaknesses

The paper only considers Gaussians as the covariate distribution. It would be better if the authors could explain how to extend their results to other distributions like uniform over $\\{ \pm 1 \\}^n$ or sub-Gaussian, or the difficulties to do that.

# Suggestions

While the upper bound is well explained in the main text, it might be good if the sketch proof for the lower bound could be described also.

---

### Official Review · Reviewer_o5b1 · 2025-09-16
**A strong paper on testable learning with noise**

**Rating:** 9
**Confidence:** 2

**Review:**

Summary: This paper poses a fundamental question on efficiently testing whether a training set satisfies the assumptions of a given noise model. Formally, they extend the recently proposed testable learning framework and frame the question as the learner with an associated test should (1) if the test accepts, the learner outputs a classifier with a certificate of optimality, and (2) the test should accept with high probability when the data set is generated from a model with certain assumptions on both the marginal distribution and the noise model. This paper considers the concrete problem of learning halfspaces over Gaussian marginals with Massart noise. This paper gives (1) a full polynomial-time testable algorithm for this setting; (2) a lower bound for the case of random andom classification noise showing that testable learning is harder than classical learning.

Comment: This paper poses a very interesting question regarding testable learning, considering both assumptions on the marginal distribution and noise model, which extends recent models proposed for testable learning. This paper gives strong theoretical results on an efficient testable learning algorithm for halfspaces with Gaussian marginals as well as a time-complexity lower bound, illustrating the limits of testable learning. I think this paper is relevant to the work and a solid contribution to reliable machine learning in the presence of noise. I do not spot any major weakness in the paper.

Typo: Line 176 "This works" -> "These works".

---

### Official Review · Reviewer_KmSL · 2025-09-22

**Rating:** 10
**Confidence:** 4

**Review:**

### Summary

This paper introduces a framework for testable learning under noise assumptions. The goal is to certify, from a given sample, that a specific label-noise assumption holds so that we can safely use learning algorithms tailored to that assumption.

The authors provide algorithms to testably learn halfspaces under Gaussian marginals with Massart noise. In this setting, labels may be flipped with probability $\eta(x)< 1/2$ that can depend on feature $x$. The guarantee is: if a subroutine (the *tester*) outputs “Accept,” then the *learner* outputs a classifier with small generalization error; moreover, if the data are generated under the stated noise model, the tester accepts with high probability.

The algorithm runs in time polynomial in the dimension and target accuracy, improving over prior testable agnostic learning results that have  exponential in $1/\varepsilon$ dependence. The paper also proves a hardness at the extreme noise level: at flip rate $1/2$, testable learning requires super-polynomial complexity (shown via Statistical Query lower bounds), even though classical learning there is trivial because labels carry no information.

---

### Strengths

- The authors provide a general definition for testable learning under noise assumptions that can apply to a plethora of settings.
- The authors depart from previous work and get polynomial time guarantees for the setting of learning under Massart noise. The previous state-of-the-art for testable learning under label-noise assumed worst-case noise (agnostic setting) and suffered an exponential in $1/\varepsilon$ dependence. As the authors suggest, their method can precede the more general agnostic tester and potentially improve on runtime.
- In contrast to previous works, the testing algorithm of this paper does not require a specific learner. That is, any learning algorithm that works for the problem of learning halfspaces under Gaussian marginals with respect to Massart noise can be used. Then the tester assesses some optimality property of the output classifier.
- In addition to the above, the tester certifies an optimality property of the specific classifier over the given sample. Even though, the testable learning definition requires to bound the generalization error of the classifier, in a setting where the data is not i.i.d. this might be the best we can hope for. The algorithm given in this work provides this guarantee.
- The paper uses sandwiching polynomials to approximate the "disagreement regions" between halfspaces under the Gaussian marginal. This allows the tester to replace costly exhaustive search with low-degree moment computations. This polynomial approximation step is key to reducing runtime while preserving the acceptance/rejection guarantees.

---

### Weaknesses

In the proof for the RCN tester, the authors bound the empirical error when the tester accepts, while the definition requires a bound on the population error. While translating the empirical error to the population error for arbitrary distributions is standard for halfspaces (due to uniform convergence), I believe this step should be explicitly stated. It is crucial for possible future extensions beyond halfspaces to fully clarify this argument.

---

### Suggestions

It would be nice to include a more detailed explanation of the lower bound provided for the case of noise level $1/2$. It establishes an interesting separation between the classical and testable frameworks.

---

### Overall

This paper is an important addition to the literature of testable learning. The testability of noise assumptions is of fundamental importance for both theory and practice. I also think that the potential future extensions to non-i.i.d. data is particularly interesting.